# Ensemble-based statistical interpolation with Gaussian anamorphosis for the spatial analysis of precipitation

Cristian Lussana[1], Thomas N. Nipen[1], Ivar A. Seierstad[1], and Christoffer A. Elo[1]

[1]Norwegian Meteorological Institute, Oslo, Norway

**Correspondence:** Cristian Lussana (cristianl@met.no)

**Abstract.** Hourly precipitation over a region is often simultaneously simulated by numerical models and observed by multiple data sources. An accurate precipitation representation based on all available information is a valuable result for numerous applications and a critical aspect of climate monitoring. Inverse problem theory offers an ideal framework for the combination of observations with a numerical model background. In particular, we have considered a modified ensemble optimal interpolation scheme. The deviations between background and observations are used to adjust for deficiencies of the ensemble. A data transformation based on Gaussian anamorphosis has been used to optimally exploit the potential of the spatial analysis, given that precipitation is approximated with a gamma distribution and the spatial analysis requires normally distributed variables. For each point, the spatial analysis returns the shape and rate parameters of its gamma distribution. The Ensemble-based Statistical Interpolation scheme with Gaussian AnamorPhosis (EnSI-GAP) is implemented in a way that the covariance matrices are locally stationary and the background error covariance matrix undergoes a localization process. Concepts and methods that are usually found in data assimilation are here applied to spatial analysis, where they have been adapted in an original way to represent precipitation at finer spatial scales than those resolved by the background, at least where the observational network is dense enough. The EnSI-GAP setup requires the specification of a restricted number of parameters and specifically the explicit values of the error variances are not needed, since they are inferred from the available data. The examples of applications presented over Norway provide a better understanding of EnSI-GAP. The data sources considered are those typically used at national meteorological services, such as local area models, weather radars and in-situ observations. For this last data source, measurements from both traditional and opportunistic sensors have been considered.

## 1 Introduction

Precipitation amounts are measured or estimated simultaneously by multiple observing systems, such as networks of automated weather stations and remote sensing instruments. At the same time, sophisticated numerical models simulating the evolution of the atmospheric state provide a realistic precipitation representation over regular grids with spacing of a few kilometers. An

unprecedented amount of rainfall data is nowadays available at very short sampling rates of one hour or less. Nevertheless, it
is common experience within national meteorological services that the exact amount of precipitation, to some extent, eludes
our knowledge. There may be numerous reasons for this uncertainty. For example, a thunderstorm triggering a landslide may
have occurred in a region of complex topography where in-situ observations are available but not exactly on the landslide spot,
weather radars cover the region in a patchy way because of obstacles blocking the beam, and numerical weather prediction
forecasts are likely misplacing precipitation maxima. Another typical situation is when an intense and localized summer thun-
derstorm hits a city. In this case, several observation systems are measuring the event and more than one numerical model may
provide precipitation totals. From this plurality of data, a detailed reconstruction of the event is possible, provided that the data
agrees both in terms of the event intensity and on its spatial features. This is not always the case and sometimes meteorologists
and hydrologists are left with a number of slightly different but plausible scenarios.

The objective of our study is the precipitation reconstruction through the combination of numerical model output with
observations from multiple data sources. The aim is that the combined fields will provide a more skillful representation than
any of the original data sources. As remarked above, any improvement in the accuracy and precision of precipitation can be
of great help for monitoring the weather, but not only that. Snow- and hydrological- modeling will benefit from improvements
in the quality of precipitation, which is one of the atmospheric forcing variables (Magnusson et al., 2019; Huang et al., 2019).
Climate applications that make use of reanalysis (e.g. Hersbach et al., 2020; Jermey and Renshaw, 2016) or observational
gridded datasets (e.g. Lussana et al., 2018), as for instance the evaluation of regional climate model (Kotlarski et al., 2017)
or the calculation of climate indices (Vicente-Serrano et al., 2015), may also benefit from datasets combining model output
and observations, as shown by Fortin et al. (2018). Besides, the intensity-duration-frequency curve (IDF curve) derived from
precipitation datasets are widely used in civil engineering for determining design values and the quality of the reconstruction
of extremes has a strong influence on IDF curves (Dyrrdal et al., 2015).

The data source considered in our study are precipitation ensemble forecasts, observations from in-situ measurement stations
and estimates derived from weather radars. Numerical model fields are available everywhere and the quality of their output
is constantly increasing over the years. The weather-dependent uncertainty is often delivered in the form of an ensemble. At
present, assessments using hydrological models have shown that input from numerical models "may be comparable or prefer-
able compared to gauge observations to drive a hydrologic and/or snow model in complex terrain" as stated by (Lundquist
et al., 2019) based on their review of recent research. One of the key messages by Lundquist et al. (2019) is that numerical
models represent precipitation fields at ungauged sites in a realistic and convincing way, as it is demonstrated by the accuracy
of their total annual rain and snowfall estimates, notwithstanding that daily or sub-daily aggregated precipitation fields may
misrepresents individual precipitation events, such as storms. In the work by Crespi et al. (2019), it has been demonstrated
that the combination of numerical model output and in-situ observations do improve the representation of monthly precipi-
tation climatologies over Norway, if compared to similar products based on in-situ observations only. Lussana et al. (2019b)
have successfully used monthly precipitation climatologies to improve the performances of statistical interpolation methods
in complex terrain over Norway. However, because model fields represent areal averages, the characteristics of simulated
precipitation depend significantly on the model resolution, as remarked for global and regional reanalyses over the Alps by

Isotta et al. (2019). In particular, Jermey and Renshaw (2016) demonstrates that increasing resolution via downscaling improves precipitation representation, though they also point out that assimilating observations at high resolution in numerical models is important for reconstructing high-threshold/small-scale events. The sources of model errors and their treatments in data assimilation (DA) schemes have been studied extensively. For instance, in the introduction of the paper by Raanes et al. (2015), a list of model errors is reported together with several references to other studies addressing them. Regarding precipitation forecasts, model errors often encountered in applications are (Müller et al., 2017): systematic under- or overestimation of amounts; spatial errors in the placement of events; underestimation of uncertainty. With reference to spatial analysis, we consider observed precipitation data to be more accurate than model estimates. In fact, model outputs are evaluated in their ability to reconstruct observed values. The most important disadvantage of observational networks is that often they do not cover the region under consideration, moreover observations may be irregularly distributed in space and present missing data over time. Each observational data source has its own characteristics that have been extensively studied in literature and that we will address here only superficially, since our objective is the combination of information. For example, rain gauges are possibly the most accurate precipitation measurement available at present (CIMO, 2014), apart from when the observations are affected by gross measurement errors, as they have been defined by Gandin (1988). There are multiple sources of uncertainty for gauge measurements (Zahumensky, 2004), such as catching and counting (Pollock et al., 2014). The undercatch of solid precipitation due to wind (Wolff et al., 2015) is a significant problem in cold climates. Radar-derived estimates are affected by several issues, such as blocking and non-uniform attenuation of the radar beam due to obstacles along the path, especially in complex terrain. A statement in the Introduction of the book by Germann and Joss (2004) is illuminating in this sense "To put a weather radar in a mountainous region is like pitching a tent in a snowstorm: the practical use is obvious and large — but so are the problems". In addition, weather radars do not directly measure precipitation, instead they measure reflectivity, which is then transformed into precipitation rate. The transformation itself contributes to increasing the uncertainty of the final estimates. Another important aspect of observational data that will be treated only marginally here is data quality control, in this work we will consider only quality controlled observations. To sum up, in-situ data are the more accurate observations of precipitation we will consider. Then, radar estimates, which are calibrated using gauges as references, are less accurate than in-situ data. They are spatially correlated with the actual precipitation and they are affected by less uncertainty than the simulations carried out by numerical models. Numerical model output is the basic information available everywhere and the one we consider more uncertain.

Inverse problem theory (Tarantola, 2005) provides the ideal framework for the combination of observations with a numerical model background. The marginal distribution of the precipitation analysis is assumed to be a gamma distribution and we aim at estimating its shape and rate parameters for each grid point. The gamma distribution is appropriate for representing precipitation data, as reported e.g. by Wilks (2019). The formulation of the statistical interpolation method presented is similar to the analysis step of the ensemble Kalman filter Evensen (2006) or the ensemble optimal interpolation (EnOI Evensen, 2003) with the important difference that EnOI uses a time-lagged ensemble, while the ensemble considered in our method is made of members of a single NWP model run. The hourly precipitation over the grid is regarded as the realization of a transformed Gaussian random field (Frei and Isotta, 2019). The Gaussian anamorphosis (Bertino et al., 2003) transforms data such that

precipitation better complies with the assumptions of normality that are required by the analysis procedure. The non-stationary covariance matrices are approximated with locally stationary matrices, as in the paper by Kuusela and Stein (2018). In addition, the background error covariance matrix includes a static (i.e. not flow-dependent) scale matrix that accounts for deficiencies in the background ensemble as in hybrid ensemble optimal interpolation (Carrassi et al., 2018). The term scale matrix has been used by Bocquet et al. (2015). In the following, the ensemble-based statistical interpolation with Gaussian anamorphosis for the spatial analysis of precipitation is referred to as EnSI-GAP. From the point of view of geostatistics, EnSI-GAP can be thought of as performing a Kriging (Wackernagel, 2003) of the Gaussian transformed ensemble mean, then retrieving the probability distribution of precipitation at every location using a predefined Gamma distribution.

The innovative part of the presented approach to statistical interpolation is in the application to spatial analysis of concepts that are usually encountered in DA. The formulation of the problem is adapted to our aim, which is improving precipitation representation instead of providing initial conditions for a physical model, as it is for DA. In the literature, there are a number of articles describing similar approaches applied to precipitation analysis, such as Mahfouf et al. (2007); Soci et al. (2016); Lespinas et al. (2015). However, our statistical interpolation is the first one, to our knowledge, where the background error covariance matrix is derived from numerical model ensemble and where Gaussian anamorphosis is applied directly to precipitation data. An additional innovative part of the method is that EnSI-GAP does not require the explicit specification of error variances for the background or observations, as most of the other methods (Soci et al., 2016). In fact, those error variances are often difficult to estimate in a way that is general enough to cover a wide range of cases. Our approach is to specify the reliability of the background with respect to observations, in such a way that error variances can vary both in time and space. An additional innovative part of our research is that we consider opportunistic sensing networks of the type described by de Vos et al. (2020) within the examples of applications proposed. Citizen weather stations are rapidly increasing in prevalence and are becoming an emerging source of weather information, as described by Nipen et al. (2020). Thanks to those networks, for some regions we can rely on an extremely dense spatial distribution of in-situ observations.

The remaining of the paper is organized as follows. Sec. 2 describes the EnSI-GAP method in a general way, without references to specific data sources. Sec. 3 presents the results of EnSI-GAP applied to three different problems: an idealized experiment, then two examples where the method is applied to real data.

## 2  Methods: EnSI-GAP, Ensemble-based statistical interpolation with Gaussian anamorphosis for precipitation

We assume that the marginal probability density function (PDF) for the hourly precipitation at a point in time follows a gamma distribution (Wilks, 2019). This marginal PDF is characterized through the estimation of the gamma shape and rate for each point and hour.

Precipitation fields are regarded as realizations of locally-stationary, transformed Gaussian random fields, where each hour is considered independently from the others. The time sequence of EnSI-GAP simulated precipitation fields do show temporal continuity because this is present in both observations and background fields. Transformed Gaussian random fields are used for the production of precipitation observational gridded datasets by Frei and Isotta (2019). A random field is said to be stationary

if the covariance between a pair of points depends only on how far apart they are located from each other. Precipitation totals are nonstationary random fields because of the nonstationarity of weather phenomena or simply the influence of topography. In our method, precipitation is locally modeled as a stationary random field. The covariance parameter estimation and spatial analysis are carried out in a moving-window fashion around each grid point. A similar approach is described by Kuusela and Stein (2018) and the elaboration over the grid can be carried out in parallel for several grid points simultaneously.

An implementation of EnSI-GAP is reported in Algorithm 1. The mathematical notation and the symbols used are described in two tables: Tab. 1 for global variables and Tab. 2 for local variables, which are those variables that vary from point to point. As in the paper by Sakov and Bertino (2011), upper accents have been used to denote local variables so, for example, $\overset{i}{\mathbf{X}}$ is the local version of matrix $\mathbf{X}$. If $\mathbf{X}$ is a matrix, $\mathbf{X}_i$ is its $i$th column (column vector) and $\mathbf{X}_{i,:}$ is its $i$th row (row vector). The Bayesian statistical method used in our spatial analysis is optimal for Gaussian random fields. Then, a data transformation is applied as a pre-processing step before the spatial analysis. The introduction of a data transformation compels us to inverse transform the predictions of the spatial analysis back into the original space of precipitation values.

The data transformation chosen is a Gaussian anamorphosis (Bertino et al., 2003), that transforms a random variable following a gamma distribution into a standard Gaussian. In the implementation presented, constant values of the gamma parameters shape and rate are used in the data transformation over the whole domain. The same values are used for the inverse transformation as well. The constant (in space) values are re-estimated every hour. It is worth remarking that the gamma parameters used in the data transformations must not be confused with those defining the gamma distribution of the hourly precipitation at each grid point and that are the objective of our spatial analysis. The analysis procedure returns a different Gaussian PDF for each grid point, which is transformed into a gamma distribution by means of the constant shape and rate estimated for the data transformation. However, since the inverse transformation at each grid point is applied to a Gaussian PDF that differs from those of the surroundings points, then the gamma distribution of hourly precipitation will also vary from one grid point to the other. The gamma shape and rate parameters used in the data transformation are denoted as the scalar values $\alpha_D$ and $\beta_D$, respectively, while the spatially dependent gamma analysis parameters are denoted with the $m$ column vectors $\boldsymbol{\alpha}^{\mathrm{a}}$ and $\boldsymbol{\beta}^{\mathrm{a}}$.

Algorithm 1 can be divided into three parts, that are described in the next sections: the data transformation in Sec. 2.1, the Bayesian spatial analysis in Sec. 2.2 and the inverse transformation in Sec. 2.3.

## 2.1 Data transformation via Gaussian anamorphosis

The Gaussian anamorphosis maps a gamma distribution into a standard Gaussian. Bertino et al. (2003) introduced the concept of Gaussian anamorphosis from geostatistics to data assimilation. A general reference on Gaussian anamorphosis in geostatistics is the book by Chiles and Delfiner (2012), Chapter 6. This pre-processing strategy has been used in several studies in the past, e.g. Amezcua and Leeuwen (2014); Lien et al. (2013). A visual representation of the transformation process can be found in Fig. 1 of the paper by Lien et al. (2013) and in this article in Sec. 3.2.2.

The hourly precipitation background and observations, $\tilde{\mathbf{X}}^{\mathrm{f}}$ and $\tilde{\mathbf{y}}^{\mathrm{o}}$ respectively, are transformed into those used in the spatial analysis by means of the Gaussian anamorphosis $g()$:

$$\mathbf{X}^{\mathrm{f}} = g(\tilde{\mathbf{X}}^{\mathrm{f}}) \tag{1}$$

$$\mathbf{y}^{\mathrm{o}} = g(\tilde{\mathbf{y}}^{\mathrm{o}}) \tag{2}$$

As indicated in Tab. 1, the Gaussian variables are $\mathbf{X}^{\mathrm{f}}$ and $\mathbf{y}^{\mathrm{o}}$, while the variables with the original hourly precipitation values, $\tilde{\mathbf{X}}^{\mathrm{f}}$ and $\tilde{\mathbf{y}}^{\mathrm{o}}$, follow a gamma distribution. The gamma shape and rate $\alpha_D$ and $\beta_D$, respectively, of this gamma distribution are derived from the background precipitation values by a fitting procedure based on maximum likelihood.

In this paragraph, the procedure used in Sec. 3 is described. For an arbitrary hour, two different solutions are adopted, depending on the weather conditions. We are in the presence of dry weather conditions when at least one of the ensemble members reports precipitation in less than 10% of the grid points, otherwise we have wet weather. In case of wet conditions, ensemble members are considered separately and for each of them we derive a single value of shape and a single value of rate, both are kept constants over the whole domain. The values of shape and rate are the maximum likelihood estimators calculated iteratively by means of a Newton-Raphson method as described by Wilks (2019), Sec. 4.6.2. Then, $\alpha_D$ and $\beta_D$ are the averages of all the values of shape (one value for each ensemble member) and rate (one value for each ensemble member). In case of dry weather, $\alpha_D$ and $\beta_D$ are set to typical values obtained as the averages of all the available cases.

In Gaussian anamorphosis, zero precipitation values must be treated as special cases, as explained by Lien et al. (2013). The solution we adopted is to add a very small amount to zero precipitation values, $\xi = 0.0001\mathrm{mm}$, then to apply the transformation $g()$ to all values. The same small amount is then subtracted after the inverse transformation. This is a simple but effective solution for spatial analysis, as shown in the example of Sec. 3.1. In principle, the statistical interpolation is sensitive to the small amount $\xi$ chosen, such that using 0.01 mm instead of 0.0001 mm will return slightly different analysis values in the transition between precipitation and no-precipitation. In practice, we have tested it and we found negligible differences when values smaller than e.g. 0.05 mm (half of the precision of a standard rain gauge measurement) have been used.

The transformation function $g(x)$, applied to the generic scalar value $x$, used in Eqs. (1)-(2) is:

$$g(x) = Q_{\mathrm{Norm}}\left(\mathrm{Gamma}(x + \xi; \alpha_D, \beta_D)\right) \tag{3}$$

where $\mathrm{Gamma}(x + \xi; \alpha_D, \beta_D)$ is the gamma cumulative distribution function when the shape is equal to $\alpha_D$ and the rate is equal to $\beta_D$. $Q_{\mathrm{Norm}}$ is the quantile function (or inverse cumulative distribution function) for the standard Gaussian distribution. An example of application of the procedure described above is given in Sec. 3.2.2.

For the presented implementation of EnSI-GAP, the Gaussian anamorphosis is based on the constant parameters $\alpha_D$ and $\beta_D$ over the whole domain. This assumption might be too restrictive for very large domains, such as for all Europe for instance. In this case, different solutions may be explored such as slowly varying the gamma parameters in space, or time, based on the climatology.

## 2.2 Spatial analysis

The spatial analysis inside Algorithm 1 has been divided into three parts. In Sec. 2.2.1, global variables have been defined. Then, as stated in the introduction of Sec. 2, the analysis procedure is performed on a gridpoint-by-gridpoint basis. In Sections 2.2.2-2.2.3, the procedure applied at the generic $i$th gridpoint is described. In Sec. 2.2.2, the specification of the local error covariance matrices is described. In Sec. 2.2.3, the standard analysis procedure is presented together with the treatment of a special case.

### 2.2.1 Definitions

In Bayesian statistics, according to Savage (1972), a state is "a description of the world, which is the object which we are concerned, leaving no relevant aspect undescribed" and "the true state is the state that does in fact obtain". The mathematical notation used is reported in Tab. 1- 2 and it is similar to that suggested by Ide et al. (1997). The object of our study is the hourly precipitation field $x()$, that is the hourly total precipitation amount over a continuous surface covering a spatial domain in terrain-following coordinates $\mathbf{r}$. Our state is the discretization over a regular grid of this continuous field. The true state (our "truth", $\mathbf{x}^{\mathrm{t}}$) at the $i$th grid point is the areal average:

$$\mathbf{x}_i^{\mathrm{t}} = \int_{V_i} x(\mathbf{r}) \, \mathrm{d}\mathbf{r} \tag{4}$$

where $V_i$ is a region surrounding the $i$th grid point. The size of $V_i$ determines the effective resolution of $\mathbf{x}^{\mathrm{t}}$ at the $i$th grid point. Our aim is to represent the truth with the smallest possible $V_i$. The effective resolution of the truth will inevitably vary across the domain. In observation-void regions, the effective resolution will be the same as that of the numerical model used as the background, then approximately $o(10 - 100 \, km^2)$ for high-resolution local area models (Müller et al., 2017). In observation-dense regions, the effective resolution should be comparable to the average distance between observation locations, with the model resolution as the upper bound.

The analysis is the best estimate of the truth, in the sense that it is the linear, unbiased estimator with the minimum error variance. The analysis is defined as $\mathbf{x}^{\mathrm{a}} = \mathbf{x}^{\mathrm{t}} + \boldsymbol{\eta}^{\mathrm{a}}$, where the column vector of the analysis error at grid points is a random variable following a multivariate normal distribution $\boldsymbol{\eta}^{\mathrm{a}} \sim \mathcal{N}(\mathbf{0}, \mathbf{P}^{\mathrm{a}})$. The marginal distribution of the analysis at the $i$th grid point is a normal random variable and our statistical interpolation scheme returns its mean value $\mathbf{x}_i^{\mathrm{a}}$ and its standard deviation $\sigma_i^{\mathrm{a}} = \sqrt{\mathbf{P}_{ii}^{\mathrm{a}}}$.

As for linear filtering theory (Jazwinski, 2007), the analysis is obtained as a linear combination of the background (a priori information) and the observations. The background is written as $\mathbf{x}^{\mathrm{b}} = \mathbf{x}^{\mathrm{t}} + \boldsymbol{\eta}^{\mathrm{b}}$, where the background error is a random variable $\boldsymbol{\eta}^{\mathrm{b}} \sim N(\mathbf{0}, \mathbf{P}^{\mathrm{b}})$. The background PDF is determined mostly, but not exclusively, by the forecast ensemble, as described in Sec. 2.2.1. The forecast ensemble mean is $\mathbf{x}^{\mathrm{f}} = k^{-1} \mathbf{X}^{\mathrm{f}} \mathbf{1}$, where $\mathbf{1}$ is the $m$-vector with all elements equal to 1. The background expected value is set to the forecast ensemble mean, $\mathbf{x}^{\mathrm{b}} = \mathbf{x}^{\mathrm{f}}$. The forecast perturbations are $\mathbf{A}^{\mathrm{f}}$, where the $i$th perturbation is $\mathbf{A}_i^{\mathrm{f}} = \mathbf{X}_i^{\mathrm{f}} - \mathbf{x}^{\mathrm{f}}$. The covariance matrix:

$$\mathbf{P}^{\mathrm{f}} = (k-1)^{-1} \mathbf{A}^{\mathrm{f}} \mathbf{A}^{\mathrm{fT}} \tag{5}$$

plays a role in the determination of $\mathbf{P}^b$, as defined in Sec. 2.2.2.

The $p$ observations are written as $\mathbf{y}^o = \mathbf{H}\mathbf{x}^t + \boldsymbol{\varepsilon}^o$, where the observation error is $\boldsymbol{\varepsilon}^o \sim N(\mathbf{0}, \mathbf{R})$ and $\mathbf{H}$ is the observation operator, that we consider as a linear function mapping $\mathbb{R}^m$ onto $\mathbb{R}^p$.

### 2.2.2 Specification of the observation and background error covariance matrices

Our definitions of the error covariance matrices follow from a few general principles that we have formulated. P1 (i.e. general
principle 1), background and observation uncertainties are weather- and location- dependent. P2, the background is more uncertain where either the forecast is more uncertain or observations and forecasts disagree the most. P3, observations are a more accurate estimate of the true state than the background. We want to specify how much more we trust the observations than the background in a simple way, such as e.g. "we trust the observations twice as much as the background". P4, the local observation density must be used optimally to ensure a higher effective resolution, as it has been defined in Sec. 2.2.1,
where more observations are available. P5, the spatial analysis at a particular hour does not require the explicit knowledge of observations and forecasts at any other hour. However, constants in the covariance matrices can be set depending on the history of deviations between observations and forecasts. P5 makes the procedure more robust and easier to implement in real-time operational applications.

P1 and P4 led to our choice of implementing Algorithm 1 by means of a loop over grid points. P2 will lead us to the
identification of the regions where the uncertainty on the input data is greatest. P3 will be used to define the observational uncertainty with respect to that of the background.

A distinctive feature of our spatial analysis method is that the background error covariance matrix $\overset{i}{\mathbf{P}}{}^b$ is specified as the sum of two parts: a dynamical component and a static component. This choice is consistent with P1 and P2. The dynamical part introduces nonstationarity, while the static part describes covariance stationary random variables. This choice follows
from P1 and it has been inspired by hybrid data assimilation methods (Carrassi et al., 2018). The dynamical component of the background error covariance matrix is obtained from the forecast ensemble. Because the ensemble has a limited size, and often the number of members is quite small (order of tens of members), a straightforward calculation of the background covariance matrix will include spurious correlations between distant points. Localization is a technique applied in DA to fix this issue (Greybush et al., 2011). The static component has also been introduced to remedy the shortcomings of using numerical weather
prediction as the background. There are deviations between observations and forecasts that cannot be explained by the forecast ensemble. A typical example is when all the ensemble members predict no precipitation but rainfall is observed. In those cases, we trust observations, as stated through P3. Then, the static component adds noise to the model-derived background error, as in the paper by Raanes et al. (2015). In Bocquet et al. (2015), the static component is referred to as a scale matrix, since it is used to scale the noise component of the model error, and we adopt the same term here. In scale matrix, the term "scale" is not
associated with the concept of "spatial scales", instead it refers to a scaling (amplification or reduction) of the uncertainty. We will also refer to this matrix, and its related quantities, with the letter $u$ to emphasize that this component of the background error is "unexplained" by the forecast.

$\overset{i}{\mathbf{P}}{}^{\mathrm{b}}$ is written as:

$$\overset{i}{\mathbf{P}}{}^{\mathrm{b}} = \overset{i}{\mathbf{\Gamma}} \circ \mathbf{P}^{\mathrm{f}} + \overset{i}{\sigma}{}_u^2 \overset{i}{\mathbf{\Gamma}}{}^{\mathrm{u}} \tag{6}$$

The first component on the right-hand side of Eq. (6) is the dynamical part. $\mathbf{P}^{\mathrm{f}}$ is the forecast uncertainty of Eq. (5), $\overset{i}{\mathbf{\Gamma}}$ is the localization matrix and $\circ$ is the Schur product symbol. The localization technique we apply is a combination of local analysis and covariance localization, as they have been defined by Sakov and Bertino (2011). In the local analysis, only the closest observations are used and we have implemented it by considering only observations within a predefined spatial window surrounding each grid point, up to a pre-set maximum number of $p_{mx}$. The covariance localization is implemented through the

element-wise multiplication of $\mathbf{P}^{\mathrm{f}}$ by $\overset{i}{\mathbf{\Gamma}}$, which has the form of a correlation matrix that depends on distances and that is used to suppress long-range correlations. The second component on the right-hand side of Eq. (6) is the static part. The scale matrix is expressed through a constant variance $\overset{i}{\sigma}{}_u^2$, that modulates the noise, and the correlation matrix $\overset{i}{\mathbf{\Gamma}}{}^{\mathrm{u}}$ defining the spatial structure of that noise. In the examples of applications presented in Sec. 3, both $\overset{i}{\mathbf{\Gamma}}$ and $\overset{i}{\mathbf{\Gamma}}{}^{\mathrm{u}}$ are obtained as analytical functions of the spatial coordinates. In Algorithm 1, $\overset{i}{\mathbf{\Gamma}}$ and $\overset{i}{\mathbf{\Gamma}}{}^{\mathrm{u}}$ have been specified through Gaussian functions, other possibilities for correlation

functions have been described for instance by Gaspari and Cohn (1999). We have chosen not to inflate or deflate $\mathbf{P}^{\mathrm{f}}$ directly and to modulate the amplitude of background covariances only through the terms of Eq. (6), this way we reduce the number of parameters that need to be specified. As a matter of fact, for the combination of observations and background in the analysis procedure, the $m$ by $m$ covariance matrices are never directly used. Instead, the matrices used are: the covariances between grid points and observation locations, $\overset{i}{\mathbf{G}}{}^{\mathrm{b}} = \overset{i}{\mathbf{P}}{}^{\mathrm{b}} \overset{i}{\mathbf{H}}{}^{\mathrm{T}}$, specifically only the $i$th row of this matrix is used; and the covariances

between observation locations $\overset{i}{\mathbf{S}}{}^{\mathrm{b}} = \overset{i}{\mathbf{H}} \overset{i}{\mathbf{P}}{}^{\mathrm{b}} \overset{i}{\mathbf{H}}{}^{\mathrm{T}}$. $\overset{i}{\mathbf{H}}$ is the local observation operator, that is a linear function: $\mathbb{R}^m \to \mathbb{R}^{p_i}$.

   The local observation error covariance matrix $\overset{i}{\mathbf{R}}$ is written as the constant observation error variance $\overset{i}{\sigma}{}_o^2$ multiplying the correlation matrix $\overset{i}{\mathbf{\Gamma}}{}^{\mathrm{o}}$:

$$\overset{i}{\mathbf{R}} = \overset{i}{\sigma}{}_o^2 \overset{i}{\mathbf{\Gamma}}{}^{\mathrm{o}} \tag{7}$$

$\overset{i}{\mathbf{\Gamma}}{}^{\mathrm{o}}$ often is the identity but other choices are possible. For instance, if some observations are know to be more accurate than

the average of the others, then the corresponding diagonal elements of $\overset{i}{\mathbf{\Gamma}}{}^{\mathrm{o}}$ can be set to values smaller than 1. The observation uncertainty can vary in time and space, accordingly to P1, however its spatial structure is fixed and depends on the analytical function chosen for $\overset{i}{\mathbf{\Gamma}}{}^{\mathrm{o}}$. Note that the observation error is not determined by the instrumental error only but it includes the representativeness error (Lussana et al., 2010; Lorenc, 1986), which is often the largest component of the observation error. The representative error is a consequence of the mismatch between the spatial supports of the areal averages reconstructed by

the background and the almost point-like observations.

   The spatial structures of the error covariance matrices are determined through the matrices in Eqs. (6)- (7). At this point, we need to set $\overset{i}{\sigma}{}_u^2$ and $\overset{i}{\sigma}{}_o^2$ to scale the magnitude of the covariances. In the process described below we will see that the two variances are completely determined by two scalars $\varepsilon^2$ and $\nu$, also defined below, that we assume to be known before running the spatial analysis. This prior knowledge defines the constraints that the solution has to satisfy and allows us to choose one

particular solution among all the possibilities. $\overset{i}{\sigma}_u^2$ and $\overset{i}{\sigma}_o^2$ characterize the region around the $i$th grid point as a whole, without distinguishing between the individual observations. We introduce two relationships linking $\overset{i}{\sigma}_u^2$ and $\overset{i}{\sigma}_o^2$ through two additional variances, both expressing uncertainty of a quantity over the same region around the $i$th grid point: $\overset{i}{\sigma}_b^2$ is the average background error variance; $\overset{i}{\sigma}_f^2$ is the average forecast error variance. The two relationships are:

$$\varepsilon^2 = \overset{i}{\sigma}_o^2 / \overset{i}{\sigma}_b^2 \tag{8}$$

$$\overset{i}{\sigma}_b^2 = \overset{i}{\sigma}_f^2 + \overset{i}{\sigma}_u^2 \tag{9}$$

$\varepsilon^2$ is a global variable and it is the relative precision of the observations with respect to the background. Eq. (8) implements P3 and $\varepsilon^2$ should be set to a value smaller than 1. For example, $\varepsilon^2 = 0.1$ means that we believe the observations to be ten times more precise an estimate of the true value than the background. Eq. (9) is an adaptation from Eq. (6). The next two relationships we introduce have the objective to estimate $\overset{i}{\sigma}_f^2$ and the empirical (i.e. based on data, not on theories) estimate of $\overset{i}{\sigma}_{ob}^2$. which is the sum of $\overset{i}{\sigma}_o^2$ plus $\overset{i}{\sigma}_b^2$, directly from the forecasts and the observed values. $\overset{i}{\sigma}_{ob}^2$ is used to get a reference value to judge if the ensemble spread is adequate. The equations are (the averaging operator $\langle \ldots \rangle$ is defined as in Algorithm 1):

$$\overset{i}{\sigma}_f^2 = \nu \, \langle \mathrm{diag}\left( \overset{i}{\mathbf{S}}^{\mathrm{f}} \right) \rangle \tag{10}$$

$$\overset{i}{\sigma}_{ob}^2 = \nu \, \langle \left( \overset{i}{\mathbf{y}}^{\mathrm{o}} - \overset{i}{\mathbf{y}}^{\mathrm{b}} \right)^2 \rangle \tag{11}$$

$\nu$ is an inflation factor that can be used to get better results (e.g. via optimization of cross-validation scores or other verification metrics). In addition to that, $\nu$ is introduced because Eq. (11) is sensitive to misbehaviour in the data when applied using data from one single timestep. Proper estimates of $\overset{i}{\sigma}_f^2$ and $\overset{i}{\sigma}_{ob}^2$ would require more than just one case, the ideal situation would be to consider numerous situations characterized by similar weather conditions. Instead, we prefer to stick to P5. The estimation of $\overset{i}{\sigma}_{ob}^2$ is not resistant, in the sense defined by Lanzante (1996). A few outliers in Eq. (11) may have a significant impact on $\overset{i}{\sigma}_{ob}^2$. The introduction of $\nu$ makes the estimation procedure more resilient in the presence of outliers and other non-standard behaviour. Eq. (11) is used for diagnostics in data assimilation (Desroziers et al., 2005) and it is consistent with P2. The combination of Eq. (8) and Eq. (11) returns a rough empirical estimate of $\overset{i}{\sigma}_b^2$ that is:

$$\overset{i}{\sigma}_{b\prime}^2 = \nu \, \frac{\langle \left( \overset{i}{\mathbf{y}}^{\mathrm{o}} - \overset{i}{\mathbf{y}}^{\mathrm{b}} \right)^2 \rangle}{1 + \varepsilon^2} \tag{12}$$

As a final step, to set $\overset{i}{\sigma}_u^2$ and $\overset{i}{\sigma}_o^2$ we distinguish between three situations. The first situation is when the ensemble spread is likely to underestimate the actual uncertainty because the background is missing an event or the spread is too narrow. The test condition is $\overset{i}{\sigma}_f^2 < \overset{i}{\sigma}_{b\prime}^2$. We will refer to this situation as the ensemble being overconfident or underdispersive. This is the case when a positive $\overset{i}{\sigma}_u^2$ is needed in Eq. (6) and we set its value such that $\overset{i}{\sigma}_b^2$ in Eq. (9) is equal to $\overset{i}{\sigma}_{b\prime}^2$ in Eq. (12):

$$\overset{i}{\sigma}_u^2 = \overset{i}{\sigma}_{b\prime}^2 - \overset{i}{\sigma}_f^2 = \nu \left[ \langle (\overset{i}{\mathbf{y}}^{\mathrm{o}} - \overset{i}{\mathbf{y}}^{\mathrm{b}})^2 \rangle / (1 + \varepsilon^2) - \mathrm{diag}(\overset{i}{\mathbf{S}}^{\mathrm{f}}) \right] \tag{13}$$

$$\overset{i}{\sigma}_b^2 = \nu \, \langle (\overset{i}{\mathbf{y}}^{\mathrm{o}} - \overset{i}{\mathbf{y}}^{\mathrm{b}})^2 \rangle / (1 + \varepsilon^2) \tag{14}$$

$$\overset{i}{\sigma}_o^2 = \varepsilon^2 \, \nu \, \langle (\overset{i}{\mathbf{y}}^{\mathrm{o}} - \overset{i}{\mathbf{y}}^{\mathrm{b}})^2 \rangle / (1 + \varepsilon^2) \tag{15}$$

The second situation is when the ensemble spread is consistent with the empirical estimate of $\overset{i}{\sigma}_b^2$. The test condition is $\overset{i}{\sigma}_f^2 \geq \overset{i}{\sigma}_{b\prime}^2$ and $\overset{i}{\sigma}_f^2 > 0$. We will refer to this situation as the ensemble spread being adequate. In this case the background information is given by the ensemble, without adjustments, and:

$$\overset{i}{\sigma}_u^2 \;=\; 0 \tag{16}$$

$$\overset{i}{\sigma}_b^2 \;=\; \overset{i}{\sigma}_f^2 = \nu \; \langle \operatorname{diag}(\overset{i}{\mathbf{S}}{}^{\mathrm{f}}) \rangle \tag{17}$$

$$\overset{i}{\sigma}_o^2 \;=\; \varepsilon^2 \overset{i}{\sigma}_f^2 = \varepsilon^2 \; \nu \; \langle \operatorname{diag}(\overset{i}{\mathbf{S}}{}^{\mathrm{f}}) \rangle \tag{18}$$

Equations (13)- (18) have been written with many details, in a somewhat pedantic way, to emphasize the differences between those two situations. When the ensemble is underdispersive, the sum $\overset{i}{\sigma}_o^2 + \overset{i}{\sigma}_b^2$ is bounded by the upper limit $\overset{i}{\sigma}_{ob}^2$. This is not the case when the ensemble is adequate. It is worth remarking that the test conditions are independent from $\nu$. In fact, for instance the test condition for the first situation can be equivalently written as $\langle \left( \overset{i}{\mathbf{y}}{}^{\mathrm{o}} - \overset{i}{\mathbf{y}}{}^{\mathrm{b}} \right)^2 \rangle > [(1+\varepsilon^2)\operatorname{diag}(\overset{i}{\mathbf{S}}{}^{\mathrm{f}})]$.

The third situation is the special case when the background is deemed as perfect, that is when all the observed values and all the forecasts, at all observation locations, have the same value. In practice, this occurs in case of no precipitation. In this case, $\overset{i}{\sigma}_f^2 = 0$ and $\overset{i}{\sigma}_{b\prime}^2 = 0$. Errors are not Gaussian in this case, then Eq. (6) and Eq. (7) are not needed anymore, as discussed in the next section, Sec. 2.2.3.

    With reference to the working assumptions stated at the beginning of this section, they can now be reformulated in more
precise mathematical terms by referring to the above definitions and equations. P1 led us to Eq. (6) and Eq. (7) and supported our choice of a grid point by grid point implementation of the algorithm. P2 led us to Eq. (11) and subsequent equations including the term $\overset{i}{\sigma}_{ob}^2$. P3 led us to the introduction of $\varepsilon^2$ in Eq. (8). P4 is also a key reason for having an algorithm that can be optimized as a function of the grid point under consideration. Other than that, P4 has not been used explicitly in this section, since it will in general affect the specification of $\overset{i}{\mathbf{\Gamma}}{}^{\mathrm{u}}$ in Eq. (6). In this section, we do not postulate any formulation of $\overset{i}{\mathbf{\Gamma}}{}^{\mathrm{u}}$ as
being preferable to another, this depends on the application. P4 led us to the specification of $\overset{i}{\mathbf{\Gamma}}{}^{\mathrm{u}}$ in Algorithm 1 as a location dependent matrix through $D_i$, that is the length scale determining the decrease rate of the background error unexplained by the forecasts. This length scale is set both in Algorithm 1 and Sec. 3 as a function of the observational network density in the surrounding of the $i$th grid point. In this sense, $D_i$ is dependent on the characteristics of precipitation as they can be observed by our network. This point is discussed further in Sec. 3.1.6. As far as we know, as stated in the Introduction, this is an innovative
part of our interpolation scheme since most of the other schemes do postulate that a single analytical correlation function or semi-variogram is valid for the whole spatial domain considered. P5 led us to the introduction of $\nu$ in Eqs. (10)- (11).

### 2.2.3   Analysis procedure

The expressions for the analysis and its error variance are direct results of the linear filter theory (Jazwinski, 2007) and they are derived in several books based on different formulations (e.g., Tarantola (2005); Kalnay (2003); Carrassi et al. (2018)). The
analysis at the $i$th grid point is equal to the background plus a weighted average of the $\overset{i}{p}$ innovations, while the analysis error

variance is derived from the error covariance matrices:

$$\mathbf{x}_i^{a} = \mathbf{x}_i^{b} + \overset{i}{\mathbf{G}}{}_{i,:}^{b}\left(\overset{i}{\mathbf{S}}{}^{b}+\overset{i}{\mathbf{R}}\right)^{-1}\left(\overset{i}{\mathbf{y}}{}^{o}-\overset{i}{\mathbf{y}}{}^{b}\right) \tag{19}$$

$$(\boldsymbol{\sigma}^{2})_i^{a} = \overset{i}{\mathbf{P}}{}_{ii}^{b} - \overset{i}{\mathbf{G}}{}_{i,:}^{b}\left(\overset{i}{\mathbf{S}}{}^{b}+\overset{i}{\mathbf{R}}\right)^{-1}\left(\overset{i}{\mathbf{G}}{}_{i,:}^{b}\right)^{\mathrm{T}} \tag{20}$$

Eq. (19)- (20) are also typical of optimal interpolation and the formulation used is similar to the one adopted by Uboldi et al.
(2008), which follows from Ide et al. (1997). It is worth remarking that the background used in Eq. (19) is the ensemble
mean, since we have assumed $\mathbf{x}^{b} = \mathbf{x}^{f}$ in Sec. 2.2.1. The ensemble members are used to determine the background error
covariance matrices. The method is a modified version of EnOI (Evensen, 2003) where an ensemble of synchronous realizations
is considered instead of a time-lagged ensemble approach. As an additional difference between EnSI-GAP and other methods,
it should be noted that the grid point by grid point implementation makes it possible to modify the interpolation settings to
adapt them to the different regions in the domain, as discussed in Sec. 3.1.6.

The special case of a perfect background, as introduced in Sec. 2.2.2, leads to a perfect analysis $\mathbf{x}_i^{a} = \mathbf{x}_i^{b}$. Because all the
information available shows an exceptional level of agreement, we have chosen to set the analysis error variance to zero (i.e.
background is the truth), such that for those points the analysis PDFs are Dirac's delta functions and this has consequences for
the inverse transformation, as discussed in the next section, Sec. 2.3.

## 2.3   Data inverse transformation

The inverse transformation $g^{-1}$ of $g$ described in Sec. 2.1 and reported in Eq. (3) for a scalar values $x$ is:

$$g^{-1}(x) = Q_{\mathrm{Gamma}}\left(\mathrm{Norm}(x);\alpha_D,\beta_D\right) - \xi \tag{21}$$

where $\mathrm{Norm}(x)$ is the Gaussian cumulative distribution function. $Q_{\mathrm{Gamma}}(\ldots;\alpha_D,\beta_D)$ is the quantile function for the Gamma
distribution with shape $\alpha_D$ and rate $\beta_D$ that are obtained as described in Sec. 2.1. $\xi$ is a constant. If $x$ is a vector instead of a
scalar value, then we apply Eq. (21) to its components.

The inverse transformation at the $i$th grid point is written as:

$$\tilde{\mathbf{x}}_i^{a} = g^{-1}(\mathbf{x}_i^{a}) \tag{22}$$

however, we need to back-transform a Gaussian PDF and not a scalar value. Eq. (22) returns the median of the gamma distri-
bution associated to the $i$th grid point. Our goal is to obtain the $m$ vectors of gamma shape and rate $\boldsymbol{\alpha}^{a}$ and $\boldsymbol{\beta}^{a}$, respectively.
To achieve that, the inverse transformation $g^{-1}$ is applied to 400 quantiles of the (univariate) Gaussian PDF defined by $\tilde{\mathbf{x}}_i^{a}$ and
$(\boldsymbol{\sigma}^{2})_i^{a}$, a similar approach is used by Erdin et al. (2012). Then, a least-mean-square optimization procedure is used to obtain
the optimal shape and rate that better fits the back-transformed quantiles. In the special case of a perfect analysis, the analysis
PDF in the original space of hourly precipitation values is a Dirac's delta function and the analysis is the scalar obtained as in
Eq. (21) when $x = \mathbf{x}_i^{a}$.

Given $\boldsymbol{\alpha}^{a}$ and $\boldsymbol{\beta}^{a}$, it is possible to obtain the statistics that better represent the distribution for a specific application (e.g.
median, 99th percentile and so on). In Sec. 3, the analysis value chosen is often the mean as it is the value that minimizes

the spread of the variance. However, other choices may be more convenient depending on the applications, as discussed by Fletcher and Zupanski (2006) where, for instance, the mode was chosen as the best estimate. In Sec. 3, we will also consider selected quantiles of the gamma distribution to represent analysis uncertainty.

## 3    Results


The aim of this section is to provide guidance on the implementation of EnSI-GAP for some applications that we consider important or useful to understand how it works.

In Sec. 3.1, EnSI-GAP is applied over a one-dimensional grid and in a "controlled environment", using synthetic data specifically generated for testing EnSI-GAP on precipitation.

In Sec. 3.2, a second more realistic example of application for EnSI-GAP is reported, where the spatial analysis is performed for a case study of convective precipitation over South Norway. The case study can not be strictly considered an evaluation of the method, since all the available observations are used in the spatial analysis and it is not possible to validate the predictions where no observations are available. It is an example intended to show the potential of EnSI-GAP for (automatic) weather forecasting or civil protection purposes.

Section 3.3 describes the results of cross-validation experiments over South Norway. EnSI-GAP performances are evaluated for a period of five months centered over summer 2019, that is from May to September. The verification scores considered are commonly used in forecast verification and described by several books, such as for example Jolliffe and Stephenson (2012). A further useful reference for the scores is the website of the World Meteorological Organization https://www.wmo.int/pages/prog/arep/wwrp/new/jwgfvr.html (last time checked 2020-05-13).

### 3.1    One-dimensional simulations


The aim of this section is to show how EnSI-GAP works and to assess its performances with different configurations under idealized conditions. The impacts of Gaussian anamorphosis and different specifications of background error covariances are also investigated. The functioning of the algorithm is shown with the example application to a single simulation. The conclusions on the EnSI-GAP pros and cons are based on the statistics collected over 100 simulations.

### 3.1.1    Simulation setup


A one-dimensional grid with 400 points and spacing of 1 spatial unit, or $1\,u$, is considered. The domain covers the region from $0.5\,u$ to $400.5\,u$ and the generic $i$th grid point is placed at the coordinate $i\,u$. A simulation begins with the creation of a true state, then observations and ensemble background are derived from it.

The simulation presented here is shown in Fig. 1a. For each grid point, the true value (black line) is generated by a random

extraction from the gamma distribution with shape and rate set to 0.2 and 0.1, respectively. To ensure spatial continuity of the truth, an anamorphosis is used to link a 400-dimensional multivariate normal (MVN) vector with the gamma distribution. The samples from the MVN distribution, with a prescribed continuous spatial structure, are obtained as described by Wilks (2019),

chapter 12.4. The MVN mean is a vector with 400 components all set to zero and the covariance matrix is determined using a Gaussian covariance function with 10 $u$ as the reference length used for scaling distances. The effective resolution (Sec. 2.2.1)

of the truth is then 10 $u$.

The ensemble background (gray lines in Fig. 1a) on the grid, 10 members, is obtained by perturbing the truth. The background values at observation locations are obtained from the members using nearest neighbour interpolations. For each member, the truth is perturbed by shifting it along the grid by a random number between -10 $u$ and +10 $u$, thus simulating misplacement of precipitation events. Then, the effective resolution of the member is set to be coarser than that of the truth. The true

values are multiplied by coefficients derived from a uniform distribution with values between 0.05 and 2 and a spatial structure function given by a MVN with Gaussian covariance function with a reference length extracted from a Gaussian distribution with mean of 50 $u$ and a standard deviation of 5 $u$. Two special regions are considered and they are shown with the shaded bright color in Fig. 1. In region R1, between 50 $u$ and 150 $u$, each background member follows an alternative truth (i.e. it is literally being derived from a different truth) that is everywhere different from 0 mm. In R1 the background is neither accurate

nor precise and this leads to the occurrence of misses and false alarms. In region R2, between 200 $u$ and 300 $u$, none of the ensemble members simulate precipitation while the true state reports precipitation. In this region the background is precise but not accurate, since the ensemble is missing, or poorly representing, an event which is otherwise well covered by observations. Because we had to ensure continuity of the background, we have enforced smooth transitions between the two regions and their surroundings. For example, R2 is actually beginning a bit after 200 $u$ and ending a bit before 300 $u$.

The number of observations (blue dots in Fig. 1a) is set to 40. The observed value at a location is obtained as the true value of the nearest grid point, plus a random noise that is determined as a random number between -0.02 and 0.02 that multiplies the true value. The procedure is consistent with the fact that observation precipitation errors should follow a multiplicative model (Tian et al., 2013). The observation locations are randomly chosen, there are: 5 between 1 $u$ and 100 $u$; 30 between 101 $u$ and 300 $u$; 5 between the 301 $u$ and 400 $u$. The distribution is denser in the central part of the domain and sparser closer to the

borders.

The effect of the Gaussian anamorphosis is shown in Fig. 1b. The transformed precipitation varies within a smaller range than the original precipitation, thus effectively shortening the tail of the distribution, reducing its skewness and making it more similar to a Gaussian distribution.

An example application of EnSI-GAP is presented in Algorithm 1. The choices that are kept fixed and that will not vary for

the whole Sec. 3.1.1 are described in this paragraph. The localization matrix $\overset{i}{\boldsymbol{\Gamma}}$, of Eq. (6), is specified using Gaussian functions, of the form of those used in Algorithm 1 for $\overset{i}{\mathbf{Z}}$ and $\overset{i}{\mathbf{V}}$, with $L_i = 25\,u$ for all the grid points. The sensitivity of the results to variations in the specification of the scale matrix will be investigated in Sec. 3.1.3, nonetheless the strategy determining $D_i$ will always be the same whether we choose to use a Gaussian function, as in Algorithm 1, or an exponential function. $D_i$ is determined adaptively at each grid point as shown in Fig. 1c as the distance between the $i$th grid point and its closest 3rd

observation location. In addition, $D_i$ has been constrained to vary between 5 $u$ and 20 $u$. The tool used to quantify the impact of the spatial distribution of the observations on the analysis is the Integral Data Influence (IDI Uboldi et al., 2008), that is a parameter that stay close to 1 for observation-dense regions, while it is exactly equal to 0 in observation-void regions. In

practice, the IDI at the $i$th grid point is computed here as the analysis in Eq. (19) when all the observations are set to $1$ and the background to $0$. IDI has been adapted to EnSI-GAP in the sense that only the scale matrix is considered in the calculation

of $\overset{i}{\mathbf{P}}\!^{\mathrm{b}}$ in Eq. (6), because the part of $\overset{i}{\mathbf{P}}\!^{\mathrm{b}}$ taking into account the atmospheric dynamics does not depend on the observational network. Where the IDI is close to zero, the analysis is as good as the background. The panel $d$ in Fig. 1 shows the IDI when $D_i$ is set as the distance between the $i$th grid point and its closest 3rd observation location. EnSI-GAP is very sensitive to the tuning of $D_i$ and its estimation is further discussed in Sec. 3.1.6.

### 3.1.2 Evaluation scores

The evaluation of analysis versus truth at grid points are evaluated using two scores, that are applied over precipitation values. The mean-squared error skill score (MSESS) quantifies the agreement between the analysis expected value and the truth. The continuous ranked probability score (CRPS) is a much used measure of performance for probabilistic forecasts. The definitions of both scores can be found e.g. on Wilks (2019). The MSESS has been used for studies on precipitation by e.g. Isotta et al. (2019), while applications of CRPS to precipitation can be found e.g. in the paper by Hersbach (2000). The definitions adapted

to our case are reported here:

$$\mathrm{MSESS} = 1 - \frac{\frac{1}{m}\sum_{i=1}^{m}(\tilde{\mathbf{x}}_i^{\mathrm{a}} - \tilde{\mathbf{x}}_i^{\mathrm{t}})^2}{\frac{1}{m}\sum_{i=1}^{m}(\tilde{\mathbf{x}}_i^{\mathrm{t}} - c)^2} \quad ; \quad c = \frac{1}{m}\sum_{i=1}^{m}\tilde{\mathbf{x}}_i^{\mathrm{t}} \tag{23}$$

$$\overline{\mathrm{CRPS}} = \frac{1}{m}\sum_{i=1}^{m}\int_{0}^{\infty}[F_{\mathrm{a}}(\boldsymbol{\alpha}_i^{\mathrm{a}}, \boldsymbol{\beta}_i^{\mathrm{a}}; y) - F_{\mathrm{t}}(\tilde{\mathbf{x}}_i^{\mathrm{t}}; y)]^2 dy \tag{24}$$

$\overline{\mathrm{CRPS}}$ is the CRPS averaged over all the grid points. The squared difference is between the continuous cumulative distribution

functions (CDFs): $F_{\mathrm{a}}(\boldsymbol{\alpha}_i^{\mathrm{a}}, \boldsymbol{\beta}_i^{\mathrm{a}}; y)$, which is the gamma analysis CDF at point $i$, with the indicated shape and rate parameters, evaluated at the value $y$; $F_{\mathrm{t}}(\mathbf{x}_i^{\mathrm{t}}; y)$ is the Heaviside function, which is equal to $0$ when $y < \mathbf{x}_i^{\mathrm{t}}$ and equal to $1$ when $y \geq \mathbf{x}_i^{\mathrm{t}}$.

### 3.1.3 Sensitivity analysis on the scaling parameters

A sensitivity analysis on variations in the scaling parameters $\nu$, $\varepsilon^2$ and in the correlation function defining $\overset{i}{\boldsymbol{\Gamma}}\!^{\mathrm{u}}$ is presented. At the same time, the operation of EnSI-GAP is shown step-by-step.

The sensitivity study considers three situations, which are used also in Figs. 2- 6. A reference setup is defined with $\nu = 0.5$ and $\varepsilon^2 = 0.5$. Then, we consider a perturbed situation where $\varepsilon^2 = 0.1$ and the observations are assumed to be 10 times more precise than the background. Finally, a situation is considered with $\nu = 0.1$ where only a small part of the ensemble spread determines $\overset{i}{\sigma}_b^2$. In addition, two different functions are used for the specification of the scale matrix $\overset{i}{\boldsymbol{\Gamma}}\!^{\mathrm{u}}$: a Gaussian function and an exponential function. In the scientific literature, both functions have been used to specify correlations for spatial analysis of

precipitation. For instance, the Gaussian function is used by (Lussana et al., 2009; Erdin, 2009) and the exponential function by Mahfouf et al. (2007); Lespinas et al. (2015); Soci et al. (2016).

The scaling of the covariances, which in turn determines the weights used in the analysis, is determined by $\overset{i}{\hat{\sigma}}{}^2_o$ (Eq. (7)) and $\overset{i}{\hat{\sigma}}{}^2_u$ (Eq. (6)), that are related to $\overset{i}{\hat{\sigma}}{}^2_b$ and $\overset{i}{\hat{\sigma}}{}^2_f$. In Fig. 2, the variances are shown and their values do not depend on the correlation functions, they depend only on $\nu$ and $\varepsilon^2$. In the reference situation, Fig. 2a, the ensemble spread is adequate ($\overset{i}{\hat{\sigma}}{}^2_u = 0$) for 58% of the grid points and it is overconfident between points $160\,u$ and $300\,u$, most of these points are in R2. $\overset{i}{\hat{\sigma}}{}^2_b$ and $\overset{i}{\hat{\sigma}}{}^2_o$ are larger in R1 and R2 than outside those two special regions, as expected, and in R2 $\overset{i}{\hat{\sigma}}{}^2_b$ is almost equal to $\overset{i}{\hat{\sigma}}{}^2_u$ because the ensemble is missing the precipitation event. On average, $\overset{i}{\hat{\sigma}}{}^2_b = 0.15$ and $\overset{i}{\hat{\sigma}}{}^2_o = 0.07$. In the case in Fig. 2b of $\varepsilon^2 = 0.1$, the percentage of points where the spread is adequate decreases to 27%, such that the scale matrix is used more than in the reference situation. The mean values become $\overset{i}{\hat{\sigma}}{}^2_b = 0.19$ and $\overset{i}{\hat{\sigma}}{}^2_o = 0.02$. In the case in Fig. 2c of $\nu = 0.1$, the reduction of $\overset{i}{\hat{\sigma}}{}^2_b$ is evident and on average $\overset{i}{\hat{\sigma}}{}^2_b = 0.03$ and $\overset{i}{\hat{\sigma}}{}^2_o = 0.015$. The percentage of points where the spread is adequate is determined by $\varepsilon^2$, then in panel $c$ it is the same as in the reference situation.

The transformed precipitation analysis is shown in Fig. 3 and the analysis in the original precipitation space, after the inverse transformation, is shown in Fig. 4. The layout of the figures is organized such that each row corresponds to the same row in Fig. 2, in the left column a Gaussian function has been used in $\overset{i}{\mathbf{\Gamma}}{}^{\mathrm{u}}$ and in the the right column an exponential function has been used.

By comparing Figs. 3- 4 with Fig. 2, it is possible to study the impact of different choices on the analysis in the transformed space. By increasing (decreasing) the error variances, the analysis spread increases (decreases) too. The comparison Gaussian versus exponential correlation function shows that, given the same values of $\nu$ and $\varepsilon^2$, the exponential function shows larger analysis spread. The analysis expected values do not vary significantly among panels that are on the same row, thus indicating that the expected value is not that sensitive to the correlation function chosen. For instance, the MSESS for Fig. 4c is 0.78, while for Fig. 4d is 0.77. The $\overline{\mathrm{CRPS}}$ for Fig. 4c is 0.43, while for Fig. 4d is 0.44. For the other panels of Fig. 4, the MSESS and $\overline{\mathrm{CRPS}}$ have lower values. The comparison to the reference situations of Fig. 4 panels $a$ and $b$, show that the analysis expected values in panels $c$ and $d$ do better fit the observations and the analysis spread is more likely to include the true values. The situation is the opposite in panels $e$ and $f$, the reference setup performs better.

The analysis of over 100 simulations confirms the considerations we have made above on the basis of a single simulation. If we consider 100 simulations, the results are shown in Tab. 3, in the EnSI-GAP column. The configuration leading to the best results, in terms of both MSESS and $\overline{\mathrm{CRPS}}$, is the one shown in Fig. 4c. The worst results have been obtained when $\nu = 0.1$.

### 3.1.4 Considerations on the data transformation

In Fig. 5 the EnSI-GAP results are shown for the same settings used in Sec. 3.1.3 without applying data transformations, just interpolating the original precipitation values. The layout of Fig. 5 is the same as in Fig. 4. The best results are found in Fig. 5 for the configurations of panels $c$ and $d$, as in Fig. 4. The agreement between analysis expected values and true values is similar to those shown in Fig. 4, the differences are small. For instance, the MSESS of Fig. 5c is 0.76. The most evident difference is in the spike in analysis spread between $50\,u$ and $100\,u$, which is present in Fig. 5 and absent in Fig. 4. This may indicate that without data transformation it is more likely to obtain unrealistically large analysis spread.

The comparison of the analysis spread between Fig. 5 and Fig. 4 shows also that without data transformation it is more likely that the true values fall outside the analysis spread shown in the figure. For example, in Fig. 4d the analysis spread includes the true values for 75% of the grid points when precipitation is higher than 1 mm, with respect to Fig. 5d that percentage is 53%. The $\overline{\text{CRPS}}$ for Fig. 5c is 0.59, while for Fig. 5d is 0.56.

      If we consider 100 simulations, the results are reported in the column "no transformation" of Tab. 3. The MSESS is often
comparable or even slightly higher than using EnSI-GAP, which confirms that the analysis expected value does provide a good fit of the truth even without data transformation. The benefits of the data transformation are in the better representation of the analysis PDF, as it can be seen by comparing the $\overline{\text{CRPS}}$: the analysis with the data transformation performs better for all configurations.

### 3.1.5    Considerations on the use of an ensemble

In Fig. 6 the results are shown when the ensemble background is not considered, instead a single member or the ensemble mean are considered. In this case, in Eq. (6), $\overset{i}{\mathbf{P}}{}^{\text{f}}$ is not considered and $\overset{i}{\mathbf{P}}{}^{\text{b}}$ is determined only by $\overset{i}{\mathbf{\Gamma}}{}^{\text{u}}$. Note that in R2 the differences between Fig. 6 and Fig. 4 are very small, since in R2 $\overset{i}{\mathbf{P}}{}^{\text{b}}$ is almost equal to $\overset{i}{\mathbf{\Gamma}}{}^{\text{u}}$ anyway. In the figure, $\overset{i}{\mathbf{\Gamma}}{}^{\text{u}}$ is specified only through an exponential function, which shows better results than the Gaussian function as in the previous two sections. In the left column, the results are shown when the best ensemble member is chosen as the background. The best member is defined
as the one that better fits the observations, in terms of minimizing the squared deviations between background and observed values. In the right column, the ensemble mean is chosen as the background.

      When comparing the three different configurations, the general considerations are the same as in Secs. 3.1.3- 3.1.4. The best results have been obtained with $\varepsilon^2 = 0.1$ and $\nu = 0.5$, in panels c-d. In particular, the analyses based on the ensemble mean perform better than with the best member, which sometimes may deviate significantly from the truth as it happens between $50\,u$
and $100\,u$. The scores support this conclusion. For Fig. 6c, $\text{MSESS} = 0.64$ and $\overline{\text{CRPS}} = 0.54$. For Fig. 6d, $\text{MSESS} = 0.72$ and $\overline{\text{CRPS}} = 0.50$.

      If we consider 100 simulations, the results are reported in the column "no ensemble" of Tab. 3 and only for the case when the ensemble mean is considered as the background. EnSI-GAP performs better than in the case of a deterministic background for almost all configurations. Only in the case of $\varepsilon^2 = 0.1$, $\nu = 0.5$ and $\overset{i}{\mathbf{\Gamma}}{}^{\text{u}}$ defined through an exponential function, the analysis
performs better without considering the ensemble. In fact, the MSESS and $\overline{\text{CRPS}}$ mark this configuration as the one returning the best results among all configurations.

### 3.1.6    Discussion

If we consider the 100 simulations on the one-dimensional grid, the comparison of results in Tab. 3 between the different implementation modes (EnSI-GAP, no transformation and no ensemble) brings us to the following conclusions on the benefits
of EnSI-GAP. The use of Gaussian anamorphosis ensures a more accurate probabilistic analysis than without any data transformation, as demonstrated by the fact that EnSI-GAP shows the best CRPS for almost all the configurations. The use of an

ensemble in the definition of $\overset{i}{\mathbf{P}}^{\mathrm{b}}$ allows the analysis to be more resistant to misbehaviour in the background, as shown by the better scores obtained by EnSI-GAP for most of the configurations.

The comparison between exponential and Gaussian correlation functions in $\overset{i}{\mathbf{\Gamma}}^{\mathrm{u}}$ favors the exponential function. From geo-statistics, we know that a Gaussian variogram model is infinitely differentiable at the origin (Wackernagel, 2003). This imposes unrealistic smoothness constraints on the analysis and, as a side effect, causes an over-confidence in the analysis: underestimation of the analysis uncertainty and a tendency to produce high and low values outside the range of observations. Those effects are more evident where the observational network is sparse and the spatial analysis scheme is less constrained by the observations. The risks related to the use of a Gaussian covariance are described by Diamond and Armstrong (1984).

The EnSI-GAP implementation in Algorithm 1 requires the specification of four parameters: $D$, $L$, $\nu$ and $\varepsilon^2$. In the previous sections, the last two parameters have been considered in the sensitivity study. In this last paragraph of this section, some general considerations on the setup of $D$ and $L$ are presented. The optimization of $D_i$ is an important part of EnSI-GAP, as remarked in Sec. 3.1.1. There are classical methods for estimating the statistical structure of background errors as a function of observation location separation (Lönnberg and Hollingsworth, 1986) based on minimizing the deviations between theoretical structure functions and empirical estimates from data. When the variation is bounded, the covariance function is equivalent to a variogram, which is used in geostatistics (Wackernagel, 2003). Often, one single value of $D_i$ is considered valid for the whole domain, as for instance by Uboldi et al. (2008). In accordance with P4 of Sec. 2.2.2, we want $D_i$ to be dependent on the spatial location. The blending of different variograms using regional weights has been done for temperature by Frei (2014); Hiebl and Frei (2016). For precipitation, the method described by Hiebl and Frei (2018) adapts the estimation of variograms for daily precipitation anomaly fields to the density of the observational network. In this document, we follow a simple procedure, each time step is considered independently from the others (P5, Sec. 2.2.2) and we take advantage of the choice to implement the algorithm based on a grid point by grid point elaboration. The observations and background are combined into the analysis because we want some observations, not just one observation, to have an impact on the analysis in the surrounding of a point. In Sec. 3.1.1, the IDI has been introduced and it is shown in Fig 1d. We have configured the simulation such that the IDI is almost always larger than 0.8, which can be roughly interpreted as having at least one observation, possibly a few, significantly influencing the analysis everywhere over the domain. The procedure we suggest to set $D_i$ is the following. Have the objective of your investigation clear in your mind. Choose a functional form of the scale matrix that suits your objective. Test different strategies for the determination of $D_i$, based on an inspection of the IDI showing the regions of the domain that would be more influenced by the observations. Select the range of values for $D_i$ that may lead to acceptable results in the spatial analysis. Refine the optimization of $D_i$ by evaluating EnSI-GAP performances on the basis of skill-scores that serve your goals.

$L_i$ depends on the characteristics of the background used and it should reflect the size of typical precipitation events occurring in a region. If we assume that it is reasonable to use the observational network to refine the effective resolution of the background, then we can imagine that $L_i$ should be set to values larger than $D_i$.

## 3.2 Intense precipitation case over South Norway

The data used in this section are those used in the operational daily routine at MET Norway. The forecasts are from the MetCoOp Ensemble Prediction System (MEPS, Frogner et al., 2019). MEPS has been running operationally four times a day (00 UTC, 06 UTC, 12 UTC, 18 UTC) since November 2016 and its ensemble consists of 10 members. The hourly precipitation fields are available over a regular grid of 2.5 km. In the articles by Frogner et al. (2019); Müller et al. (2017) the performances of MEPS in simulating precipitation fields are discussed in detail. MEPS adds more value over deterministic forecasts for summer precipitation events than for winter. The smaller spatial scales (e.g. smaller then $\approx 50\,\mathrm{km}$) have some predictability for up to 6 hour forecast lead time. One of the main findings of the study by Frogner et al. (2019) was that "with limited predictability of small scales, post-processing should be an integrated part of any system". The observational dataset of hourly precipitation is composed of two data sources: precipitation estimates derived from the composite of MET Norway's weather radar and meteorological weather stations equipped with ombrometers, such as rain gauges or other devices. The hourly precipitation in-situ observations have been retrieved from MET Norway's climate database frost.met.no (last time the website was checked is 2020-05-13). In addition to MET Norway's official weather stations, the database includes data collected by several Norwegian public institutions, such as for example: universities (e.g., the Norwegian Institute of Bioeconomy Research - Nibio), the Norwegian Water Resources and Energy Directorate (NVE), the Norwegian Public Roads Administration (Statens vegvesen). As described in the recent paper by Nipen et al. (2020), MET Norway is successfully integrating amateur weather stations temperature data into its operational routine. The method applied is described by Lussana et al. (2019a). Integrating citizen observations into operational systems comes with a number of challenges. The operational systems must be robust and therefore rely on strict quality control procedures, such as those described by Båserud et al. (2020). In this study, hourly precipitation observations from the same network of opportunistic sensors are considered and used both in Sec. 3.2 and in Sec. 3.3. The majority of data measured by stations managed by citizens have been collected thanks to collaboration between MET Norway and Netatmo, a manufacturer of private weather stations. The observations used in Sec. 3.2 and in Sec. 3.3 have been quality controlled by MET Norway, therefore they are considered as correct data.

A mass of moist air from the ocean moving towards the Norwegian mountains was at the origin of several intense showers over western Norway on the 30th of July 2019. South Norway, the domain considered, is shown in Fig. 7; it measures 373 km in the meridional and 500 km in the zonal directions. The measurements from MET Norway's weather stations show values with more than $20\,\mathrm{mm/h}$, which is extremely intense given the climatology of the region. In addition, thousands of lightning strikes have been recorded (not shown here), thus confirming the convective nature of the precipitation. Intense events have been observed in the afternoon along the coast and over the nearby mountains, especially in Sogn og Fjordane. This region is shown as the black box in Fig. 7, it extends for 80 km in both meridional and zonal directions. Point A is well covered by observations and it corresponds to the center of a grid box where a maximum of precipitation has been observed. Point B is the center of a grid box that is not covered by observations and where a maximum of precipitation has been reconstructed by the analysis. The distance between points A and B is 14 km and their elevations a.m.s.l. are: A at 198 m; B at 911 m. In Sogn og

Fjordane damages have been reported (Agersten et al., 2019), they were caused by the heavy rain that also triggered a series of landslides. One of them caused a fatality when a driver was caught in the debris flow.

The two domains of South Norway and Sogn og Fjordane have been chosen to showcase two typical situations that can be found in an operations center. On both domains, the focus is on the representation of hourly precipitation patterns at the mesoscale, as it has been defined by Thunis and Bornstein (1996); Stull (1988), though over different domains we will focus on different parts of the mesoscale. South Norway is used to show that the variability of the fields represented by the forecast ensemble members involves mostly the Meso-$\beta$ part of the mesoscale (i.e. spatial scales from 20 km to 200 km). Weather forecasters are used to making decisions on the basis of information at such scales. Sogn og Fjordane is a domain where high-resolution information is needed to support fine-scale analysis by e.g. civil protection authorities. In this case, we will study precipitation patterns at the Meso-$\gamma$ scale (i.e. from 2 km to 20 km).

### 3.2.1 EnSI-GAP setup

The Algorithm 1 has been used over a grid with 2.5 km of spacing, which is the resolution of the MEPS grid (see Sec. 3). The parameters are $\varepsilon^2 = 0.1$, $\nu = 0.1$, $p_{mx} = 200$, $L_i = 50\,\mathrm{km}$ constant. A Gaussian function has been used in $\overset{i}{\mathbf{\Gamma}^{\mathrm{u}}}$. $D_i$ is estimated adaptively on the grid as the distance between the grid point and the 10th closest observation location with upper and lower bounds of 3 km and 10 km, respectively. The settings are such that the analyses would stay much closer to the observations than to the forecasts, where observations are available. The analysis uncertainty will reflect locally both the forecast ensemble spread and the averaged innovation. The two parameters $p_{mx}$ and $D_i$ are used to limit the number of observations that can influence the analysis at a grid point. The localization parameter $L_i$ is set to a rather large value, such that the dynamics of the forecasts ensemble are evident in the results. The observation error covariance matrix of Eq. (7) is defined with a diagonal $\overset{i}{\mathbf{\Gamma}^{\mathrm{o}}}$, that is a situation where radar-derived and in-situ observations are assumed to have the same precision, moreover we are ignoring the spatial correlation of radar-derived observation errors. An investigation of spatially correlated radar-derived observation errors is outside the scope of this study. Note that those settings are useful for the illustration of the method, while for operational applications other settings may be more appropriate, such as a smaller value of $L_i$ or a more sophisticated characterization of the observation errors, for example.

### 3.2.2 Data transformation

As an example of application, the Gaussian anamorphosis described in Sec. 2.1 is applied here to the transformation of hourly precipitation over Sogn og Fjordane at 2019-07-30 15:00 UTC. The procedure is sketched in Fig. 8. In panel $a$, the distribution of values for an arbitrary ensemble member is shown. In panel $b$, the empirical CDFs of the 10 ensemble members are shown as gray dots and the pink lines represent the gamma CDFs that better approximate each empirical CDF. The values of the gamma shape and rate are then averaged to obtain $\alpha_D$ and $\beta_D$, which are reported on the figure. Panel $c$ displays the CDF for the standard normal, which is the target CDF in our transformation scheme. Finally, panel $d$ shows the distribution of the transformed values for the background ensemble mean, which is used as the background for the analysis in Eq. 19. In panels $a$ and $d$, the distribution of values for the observations are also shown, though they are not used for the estimation of the gamma

parameters. The effects of the Gaussian anamorphosis in adjusting the distribution of values into a bell-shaped distribution are clearly evident.

    The four different steps of the data transformation for an arbitrary value, at approximately 2 mm/h, are also highlighted with circles to guide the reader in the order of application of each step.

### 3.2.3    South Norway

Figure 9 shows the hourly precipitation data for 2019-07-30 15:00 UTC over South Norway. The observational data are shown in panel $a$. For each grid box, the average of radar-derived precipitation and in-situ measurements within that box is shown. Note that the box-averaged observations are used only for illustration because the analysis is using each observation. Grid points that are not covered by observations are marked in gray. In panel $b$, the background ensemble mean derived from a 10-member ensemble forecast is shown, while six of the ten ensemble members are shown in Fig. 10. The 10-member ensemble

shows realistic precipitation fields, moreover they are rather similar, at least in terms of weather situation at the Meso-$\beta$ scale. Weather forecasters can be quite confident in stating that heavy precipitation is likely to occur over western and southern Norway, while is less likely over eastern Norway. The forecast uncertainty is large enough that it is difficult to predict exactly which subregion will be affected by the most intense showers. The observations confirm that showers occur along the coast of western Norway and that the most intense precipitation event is located in Sogn og Fjordane, that is the black box in Fig. 7.

Note that approximately half of the box is not covered by observations. Panel $c$ of Fig. 9 shows the analysis, specifically the analysis mean at each grid point. In this case, the spatial analysis acts almost as a "gap filling" procedure to fill in empty spaces in between observations with the most likely precipitation values. The analysis of precipitation is consistent with the impacts of the intense weather event described in the report by Agersten et al. (2019). As prescribed by our EnSI-GAP settings, the analyses over observation-dense regions are not that different from the observed values.

### 655    3.2.4    Sogn og Fjordane

    One of the main innovation of EnSI-GAP compared to traditional spatial analysis methods (Hofstra et al., 2008) is the specification of anisotropic background error covariances between grid points through non-stationary covariance matrices. Two visual representations of the correlations associated with those covariances are shown in Fig 11 for points A and B. With reference to $\overset{i}{\mathbf{P}}^{\mathrm{b}}$, the background error correlations between the generic $i$th grid point and the other grid points, evaluated at the $i$th grid

point, are the $i$th row (or column) of the correlation matrix $\overset{i}{\mathbf{\Gamma}}^{\mathrm{b}}$, which is obtained as:

$$\overset{i}{\mathbf{\Gamma}}^{\mathrm{b}}{}_{i,:} = \frac{\overset{i}{\mathbf{P}}^{\mathrm{b}}{}_{i,:}}{\sqrt{\overset{i}{\mathbf{P}}^{\mathrm{b}}{}_{i,i}}\sqrt{\mathrm{diag}\left(\overset{i}{\mathbf{P}}^{\mathrm{b}}\right)}} \tag{25}$$

The correlations are shown instead of the covariances because we are interested in the shape of the covariance patterns and correlation is a quantity which is then more correct to compare between the two points. For visualization purposes, in Fig 11 the correlations have been downscaled over a finer resolution grid to highlight asymmetries. The closest two hundreds observations

are shown with different symbols, depending on rain occurrence. The two maps in Fig. 11 are rather different. For point A, the correlation extends more to the west than to the east. The point is located in a valley floor, rather sheltered from the main atmospheric flow, and this seems to be represented in its correlation pattern which rapidly decays as we move upwards. The area where the correlation is higher than 0.6 is confined within approximately 5 km in any direction from point A. At point B, the situation is different and the correlation extends more to the east than to the west. The point is located on a plateau at 911

m and the correlation pattern follows the main atmospheric flow from west to east. The no-precipitation observations 20 km north-east of point B have correlations that are comparable to those of observations at 10 km west of B.

The evolution in time of the hourly precipitation fields is shown in Fig. 12 for observations, background and analysis at three different hours: 14:00, 15:00 and 17:00 UTC. It is worth noticing that the example used to illustrate the data transformation process in Fig. 8 refers to the Sogn og Fjordane domain at 15:00 UTC. The background is smoother than the observed field

and shows scattered showers for 14:00 UTC and 15:00 UTC, then a wider precipitation cell over point B is shown at 17:00 UTC. The observed fields show a large variability over short distances and the difference between two adjacent points can be as large as 30 mm/h. According to P4 of Sec. 2.2.2, in data dense areas we would like the analysis to stay closer to the observed value than in data sparse areas. Point A is in a densely observed area, while point B is almost in the middle of the observation-void region and the closest observations are located at a distance of approximately 10 km. $D_i$ at point A is closer

to 3 km, while at point B is closer to 10 km. This ensures a higher effective resolution at point A than at point B. At 14:00 UTC, the observed value at point A (from radar-derived estimates) is over 30 mm/h and a sharp gradient from south-west to north-east is evident. The gradient is so intense that the nearby points south-west of point A, only 3 km apart, show almost no precipitation. The background indicates that a maximum of the field can occur between point A and B. The analysis matches the observations, though smoothing out their spatial variability, such that at point A the analysis value is less than 10 mm/h. A

precipitation maximum of more than 30 mm/h has been reconstructed in the analysis between points A and B, that is consistent with the gradient in the observations and the pattern in the background. At 15:00 UTC, the radar-estimated precipitation at point A is again over 30 mm/h but there are several points in its surroundings with similar values, such that the local gradient of the field is less steep and it shows a decrease of precipitation east of point A. The background also shows that it is more likely to find intense precipitation immediately to the west of point A than to the east. A second precipitation maximum is

found in the background, north of point B. The analysis ignores this second precipitation maximum, since it is not supported by observations. The analysis around point A matches closely both the observed values and the gradient, such that the field in the observational-void area does not show significant local extremes. The shape of the area with precipitation rate higher than 30 mm/h around point A is similar to the pattern of point A correlations higher than 0.6 in Fig. 11. At 17:00 UTC, all the observations report values smaller than 20 mm/h and the analysis reconstruct a maximum of over 30 mm/h at point B. In this

case, the observations and background precipitation yes/no patterns are similar and they both show a south-east to north-west gradient. The analysis estimates a narrow band of precipitation around point B where values of more than 20 mm/h and up to 30 mm/h are extrapolated. The extrapolated values are consistent with the effects of the extreme event reported by MET Norway (Agersten et al., 2019).

The time series of hourly precipitation at points A and B are shown in Fig. 13. At point A, the graphs show the time series of (aggregated) observation, background and analysis, together with the estimated uncertainties. Note that the observation is used in the analysis at point A. At point B, observations are not available. For the background, the percentiles are derived from the 10-member forecast ensemble through a linear interpolation of the empirical cumulative distribution function. For the analysis, the percentiles are derived from the estimated parameters of the gamma distribution representing the marginal probability density function of the analysis at the points. In general, EnSI-GAP forces the analysis to follow more closely the observations than the background and the analysis uncertainty is smaller than those of the background. As a consequence, the timing of the precipitation onset is also better represented in the analysis. At point A, the PDF of the precipitation analysis between 10:00 UTC and 13:00 UTC indicates with certainty that it is not raining. From 14:00 UTC onward, the analysis PDF is a gamma. From 14:00 UTC to 23:00 UTC, the observed values are within the analysis envelopes shown in Fig. 13 for 50% of the hours, that is a consistent improvement compared to the background. For the other 50% of the hours, the observed values lie outside the envelopes and 14:00 UTC and 19:00 UTC are the two hours when the deviations between observations and analyses are the most evident. For those two hours, the local variability of the precipitation field is extremely large, as shown in Fig. 12 for 14:00 UTC, and the observed values at point A are outliers, if compared to their neighbours. With respect to the precipitation yes/no distinction, from 14:00 UTC to 23:00 UTC, the analysis clearly shows that precipitation is occurring at the point, while the background is more uncertain. At point B, the analysis uncertainties between 10:00 UTC and 12:00 UTC are so small that the analysis is exactly 0 mm/h, despite there are no observations exactly located at that point. From 13:00 UTC onward, the analysis follows a gamma PDF and the spread is wider at point B that at point A. The increased analysis spread reflects the increase in the uncertainty in predicting the tails of the PDF where no observations are available. It is perhaps remarkable that even for observational dense regions, such as at point A, the analysis spread remains quite large.

### 3.2.5 Discussion

EnSI-GAP can support weather forecasters and civil protection by filling in the empty spaces in the observational networks. The analysis seamlessly merges the high-resolution NWP models with observations and it remains closer to the observed values where they are available. The predicted fields are easy to interpret by experienced staff that is aware of the spatial distribution of the observations and the characteristics of the NWP considered. The analysis is more precise and accurate than the background where observations are available, as at point A in Sec. 3.2.4, also for the onset of precipitation. Uncertainty on the estimate at a point increases as the number of nearby observations decreases. The analysis procedure modifies the field also where observations are not available in a credible way, as at point B in Sec. 3.2.4. The uncertainty estimates can be used to have an idea of the extreme values that may occur in a region, which is useful information both for nowcasting of an event and in the subsequent reporting phase.

In Sec. 3.2.4, the observed values show strong gradients over small distances. The spatial analysis finds the best estimates of true values, which are areal averages, as discussed in Sec. 2.2.1 and defined in Eq. (4), with spatial supports determined by the EnSI-GAP settings. In Fig. 13, at 14:00 UTC and 19:00 UTC, the representativeness errors of the observations at point A are particularly large with respect to the spatial supports of the true values, such that the corresponding observations get

"filtered out" as outliers by the analysis and their values are unlikely to occur according to the analysis PDF. If the ensemble is overconfident, according to the definition of Sec. 2.2.2, by reducing $D_i$ it is in principle possible to modify the analysis PDF such that the analysis spread would become larger, which in this case would correspond to a reduction of the spatial support for the true values, and the analysis envelope would be more likely to include the observations. However, when a single observation is an outlier with respect the neighbouring observations, as in Fig. 13 at 14:00 UTC and 19:00, the tuning of $D_i$ to include the observation in the analysis PDF may lead to unrealistic discontinuous patterns in the analysis due to the sudden "jump" in the spatial supports used in the definition of true values. In general, A very dense observational network, that is with observations that are closer than the effective resolution of the background, has two effects on the analysis where precipitation varies significantly over small distances: (i) it forces the analysis expected value to stay close to the areal average of the observations; (ii) it increases the observations and background error variances because of the increased value of the term $\langle \left( \overset{i}{\mathbf{y}}^{\mathrm{o}} - \overset{i}{\mathbf{y}}^{\mathrm{b}} \right)^2 \rangle$ in Eqs. 11- 18, this will in turn increase the analysis uncertainty in Eq. (20). The trade-off between accuracy and precision of the analysis at a point ultimately depends on the objective of an application.

## 3.3 Validation over South Norway through cross-validation experiments

The cross-validation experiments have been conducted over the South Norway domain shown in Fig. 7. The data sources and grid settings of the experiments are the same as for the case study of intense precipitation of Sec. 3.2. The time period considered is from the 1st of May to the 30th of September 2019. The observations from MET Norway's stations have not been used in the spatial analysis. Instead, because of the expected better quality of those measurements, they have been reserved as independent observations for verification. This cross-validation strategy is widely used in atmospheric sciences (Wilks, 2019). The locations of the 57 weather stations directly managed by MET Norway are shown in Fig. 7 as red triangles. They are distributed all over the domain, the station network density is higher along the coast and sparser on the mountains, because of the inherent difficulties of operating weather stations there.

### 3.3.1 EnSI-GAP setup

The EnSI-GAP Algorithm 1 has been used. The spatial analysis predicts values at those station locations used for cross-validation. The fixed parameters in this implementation are: $p_{mx} = 200$, $L_i = 50\,\mathrm{km}$. A Gaussian function has been used in $\overset{i}{\mathbf{\Gamma}}^{\mathrm{u}}$. $D_i$ is estimated adaptively at each location as the distance between that point and the 10th closest observation location with upper and lower bounds of 3 km and 10 km, respectively.

The parameters that are allowed to vary and that are the objective of the sensitivity analysis that follows are: $\varepsilon^2$ and $\nu$.

There is an important difference here with respect to Sec. 3.2, in this example the radar-derived estimates are assumed to be less precise than the in-situ observations but more precise then the background. The in-situ observations are assumed to be ten times more precise than the background, then $\varepsilon^2$ is set to 0.1 as in Sec. 3.2. However, the radar-derived observations are assumed to be only two times more precise than the background, or in other words they are five times less precise than the in-situ observations, and the elements of the diagonal matrix $\overset{i}{\mathbf{\Gamma}}^{\mathrm{o}}$ corresponding to radar observations are set to 5, instead of 1 as

for the in-situ observations. The background ensemble and analysis PDF values considered are those extracted at the locations of stations used for cross-validation.

### 3.3.2   Cross-validation statistics

Figure 14 shows the distribution of values for selected percentiles of the background ensemble and analysis PDF as a function of the independent observations. The distribution of the observed values has been divided into intervals, they are (units mm/h):
0-0.1, 0.1-0.5, 0.5-1, 1-2, 2-3, 3-5, 5-10, 10-35. The number of samples within each interval is shown in panel *a*, note the logarithmic scale on the y-axis. Most of the observations are smaller than 1 mm/h, nonetheless there are still more than one thousand values that are greater than 1 mm/h. Consider an arbitrary observation interval, for each probabilistic prediction in it, either background or analysis, we have computed the percentiles: 10th, 25th, 50th, 75th and 90th. The black line in Fig. 14 shows the average median within each interval, while the regions between the 90th and the 10th percentiles, and the 75th
and the 25th are shown with gray shades. The diagonal (1:1) ideal line is shown as a dashed line. The background is shown in panel *a*. The background envelope deviates significantly from the diagonal, especially for values greater than 2-3 mm/h. The analysis PDFs are shown in the other panels for different EnSI-GAP configurations that are clearly indicated within each panel. The angular coefficients of the regression lines that better fit the analysis medians are reported in panels *b-d*. In all cases, the medians are closer to the diagonal for the analyses than for the background. The analysis biases conditional to
the observations are always smaller than that of the background. As expected, by giving more weight to the observations, with $\nu = 1$ and $\varepsilon^2 = 0.1$, the analysis bias conditional to the observations decreases. If we compare different analysis configurations, the medians vary less than the other percentiles and this indicates that variations in the EnSI-GAP configuration impacts more on the spread of the analysis PDF (i.e. analysis uncertainty) than on its central moment. Panels *b* and *c* show the two extreme situations, while panel *d* displays an intermediate situation. The uncertainty is more sensitive to variations over $\nu$ than over $\varepsilon^2$.
In the case of $\nu = 0.1$ and $\varepsilon^2 = 0.1$, the angular coefficient of the regression line approximating the analysis median reaches its the best value, however the analysis spread is small and the independent observations fall above the 90th percentile. In the case $\nu = 0.1$ and $\varepsilon^2 = 0.1$, the angular coefficient of the regression line has the best value. For the two cases with $\nu = 1$, the independent observations fall into or around the 90th percentile of the analysis. Once again, it is the specific application that would determine the best combination of parameters to use.
Figure 15 shows the Equitable Threat Score (ETS) for the background and analysis means. Four different analysis configurations are shown. The independent observations are used to judge if events have occurred. The condition defining the "yes" event for either observations or predictions is that the corresponding value must be higher than the precipitation threshold specified on the x-axis. For all predictions, it is more likely that a predicted "yes" event corresponds to an observed "yes" event for smaller thresholds than for the higher ones. The added value of the analysis over the background is evident for all config-
urations. The two configurations with $\nu = 1$ present similar ETS curves, though the one with $\varepsilon^2 = 0.1$ performs better. The same holds true when $\nu = 0.1$, though in this case the ETS is more sensitive to variations in $\varepsilon^2$ and the analysis performance decreases faster with the increase of $\varepsilon^2$.

## 4 Conclusions

The ensemble-based statistical interpolation with Gaussian anamorphosis (EnSI-GAP) applies inverse problem theory to the spatial analysis of hourly precipitation. Numerical model output provides the prior information, and specifically we have considered ensemble forecasts, that have been combined with radar-derived estimates and in-situ observations. EnSI-GAP has been applied on datasets that are typically available within national meteorological services. In addition, opportunistic sensing networks based on citizen observations have been considered. The precipitation representation is a synthesis of all the data available. Thanks to the diffusion of open data policies, the same datasets are also nowadays available in real-time to the general public. For instance, MET Norway provides free access to the weather forecasts and the radar data used in this article via thredds.met.no, while in-situ observations, except the citizen observations, are available via frost.met.no.

EnSI-GAP assumes the precipitation fields to be locally stationary, transformed Gaussian random fields. The marginal distribution of precipitation at a point is a gamma distribution, that is different for each point. Gaussian anamorphosis is used to pre-process data in order to better comply with the requirements of linear filtering. A special case is considered where uncertainties are so small that the returned analysis values have delta functions as their marginal distributions.

EnSI-GAP considers each hour independently and it requires the specification of four parameters that can vary across the domain. The implementation is designed to run in parallel on a grid point by grid point basis. Despite the small number of parameters to optimize, the spatial analysis scheme is flexible enough that it can be applied also when the background ensemble is not representing the truth satisfactorily. An important case is when, in a region, all the ensemble members show no precipitation, while the observations report precipitation. By adding a scale matrix to the flow-dependent background error covariance matrix, the analysis can predict precipitation even where the background is sure that it is not occurring.

The examples of applications presented allow for a better understanding of the characteristics of EnSI-GAP and they show how the statistical interpolation can be adapted to meet specific requirements. It can be used to fill in the gaps between observation-rich regions to obtain a continuous precipitation field. The analysis expected value is available everywhere, as it is the background, and in observation-dense regions it can be as accurate as the observations. Thanks to the data transformation, the spread of the analysis PDF is less likely to become unrealistically large because of either large model errors or large variability of observed small-scale precipitation. Within certain limits, determined by the spatial distribution of the observational network, the analysis envelope at a point can be tuned such that it is representative of the distribution of precipitation values determined by atmospheric processes occurring at smaller spatial scales than those resolved by the background. For instance, in an observation-void region, the EnSI-GAP analysis PDF at a point provides a better estimate than the background for the probability of precipitation exceeding a threshold by an observation hypothetically placed at that point. This is an important result, especially when high-impact weather is involved.

*Data availability.* Some of the datasets used in Secs. 3.2- 3.3 are freely available online. MET Norway provides free access to: the weather forecasts at https://thredds.met.no/thredds/catalog/meps25epsarchive/catalog.html; the hourly precipitation derived from the Norwegian composite of weather radars https://thredds.met.no/thredds/catalog/remotesensingradaraccr/catalog.html; the archive of Norwegian historical

weather and climate in-situ observations frost.met.no. Due to distribution restrictions imposed by some of the providers, opportunistic sensing networks, such as citizen observations, are not freely available online.

*Author contributions.* CL developed EnSI-GAP, tested it on the case studies and prepared the manuscript with contributions from all co-authors. TN and IS configured EnSI-GAP to work with MET Norway's datasets, collected in-situ observations from opportunistic sensing
networks and quality controlled them. CE prepared the radar data.

*Competing interests.* The authors declare that no competing interests are present.

*Acknowledgements.* This research was partially supported by: RADPRO (Radar for Improving Precipitation Forecast and Hydropower Energy Production), an innovative industry project funded by the Research Council of Norway (NFR) and partnering hydropower industries; the collaboration between MET Norway and "The Norwegian Water Resources and Energy Directorate" (NVE) within the national project
"Felles aktiviteter NVE-MET 2019-2020 tilknyttet nasjonal flom- og skredvarslingstjeneste".

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

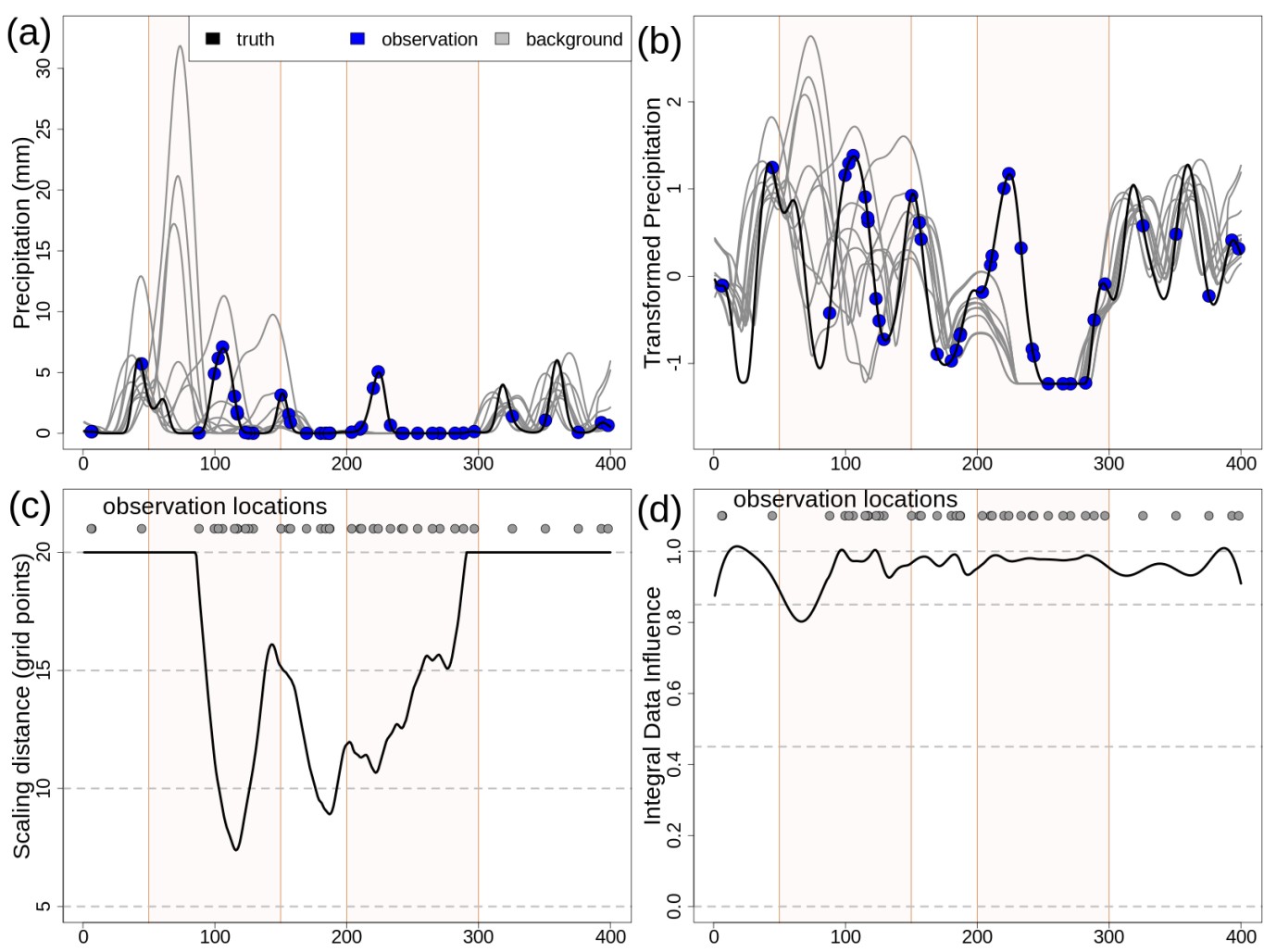

**Figure 1.** One-dimensional simulation. Panel *a*, precipitation (mm): truth (black line), observations (blue dots) and background (gray lines). Panel *b*, transformed values. Panel *c*, reference length scale for the scale matrix $D_i$ (units $u$, as defined in Sec. 3.1), $D_i$ is bounded within 3 $u$ and 20 $u$. Panel *d*, Integral Data Influence (IDI) based on $D_i$ from panel *c*. The two regions R1 and R2 have been highlighted with a shaded color in the background of each panel.

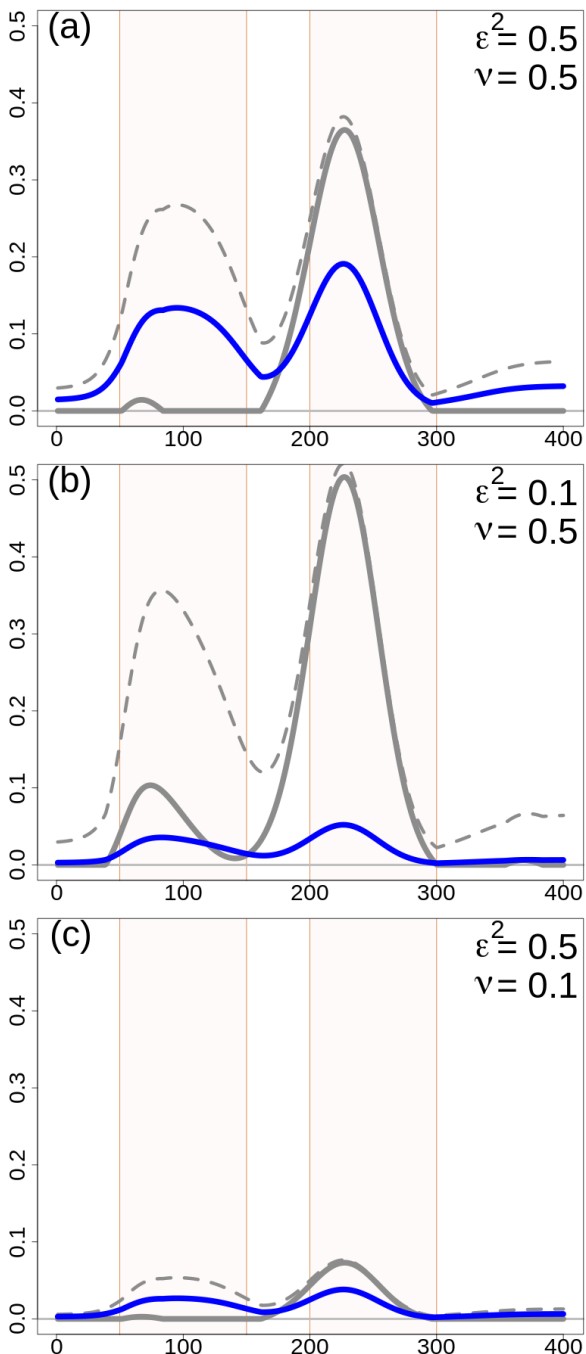

**Figure 2.** One-dimensional simulation. Error variances (dimensionless quantities) for different configurations of the scaling parameters. The variances shown are: $\overset{i}{\sigma}_u^2$ thick gray line; $\overset{i}{\sigma}_b^2$ dashed gray line; $\overset{i}{\sigma}_o^2 (= \varepsilon^2 \overset{i}{\sigma}_b^2)$ blue line. $\overset{i}{\sigma}_f^2$ is the difference between $\overset{i}{\sigma}_b^2$ and $\overset{i}{\sigma}_u^2$. For all panels $L = 25\,u$ in $\overset{i}{\mathbf{\Gamma}}$ and the error variances do not depend on choices on $\overset{i}{\mathbf{\Gamma}}$ or $\overset{i}{\mathbf{\Gamma}}^{\mathrm{u}}$. Panel $a$ $\varepsilon^2 = 0.5$ and $\nu = 0.5$. Panel $b$ $\varepsilon^2 = 0.1$ and $\nu = 0.5$. Panel $c$ $\varepsilon^2 = 0.5$ and $\nu = 0.1$. The two regions R1 and R2 have been highlighted with a shaded color in the background of each panel.

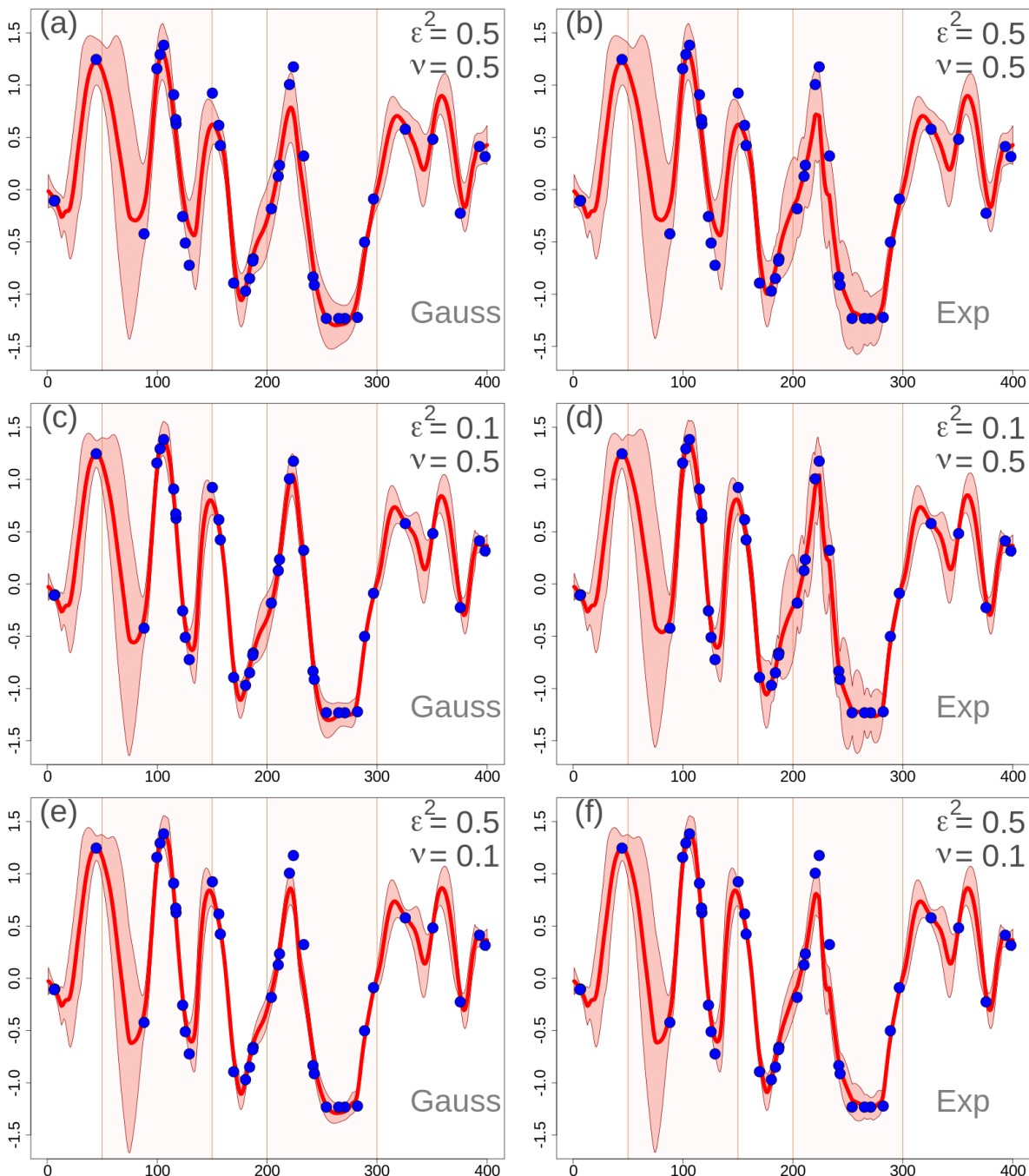

**Figure 3.** One-dimensional simulation in the transformed precipitation space. Analyses at grid points with different EnSI-GAP configurations. For all panels $L = 25\,u$. The values of $\nu$ and $\varepsilon^2$ are reported in the panels. Specification of the scale matrix $\mathbf{\Gamma}^{ui}$: the panels on the left column have been obtained with a Gaussian function, while the panels on the right column with an exponential function. For each panel, the red line is the analysis (expected value), the pink area shows the interval between the 90th and the 10th percentiles, the blue dots are the observations as in Fig. 1*b*. The two regions R1 and R2 have been highlighted with a shaded color in the background of each panel.

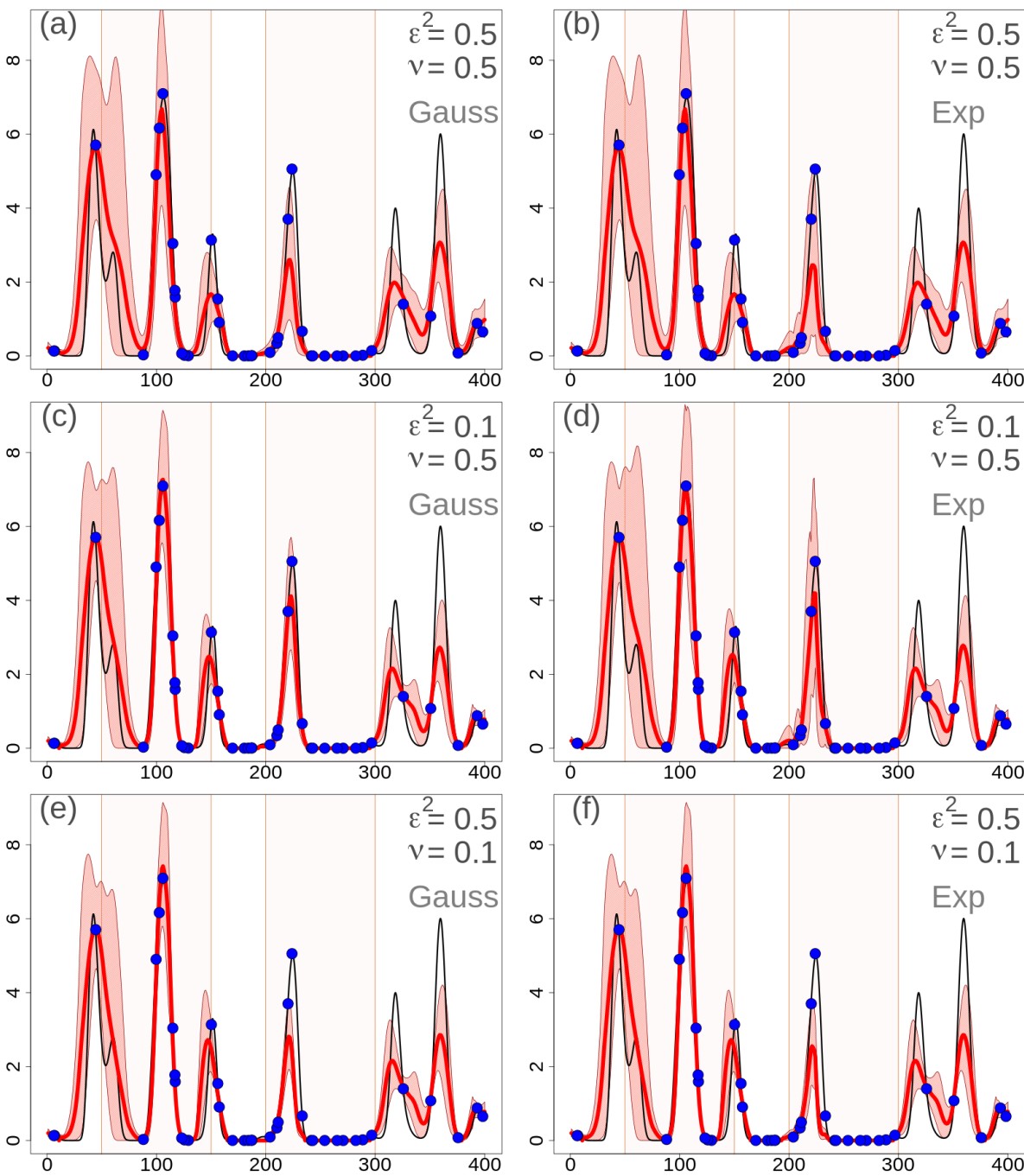

**Figure 4.** One-dimensional simulation in the original precipitation space (mm). Analyses at grid points with different EnSI-GAP configurations. The layout is the same as in Fig. 3.

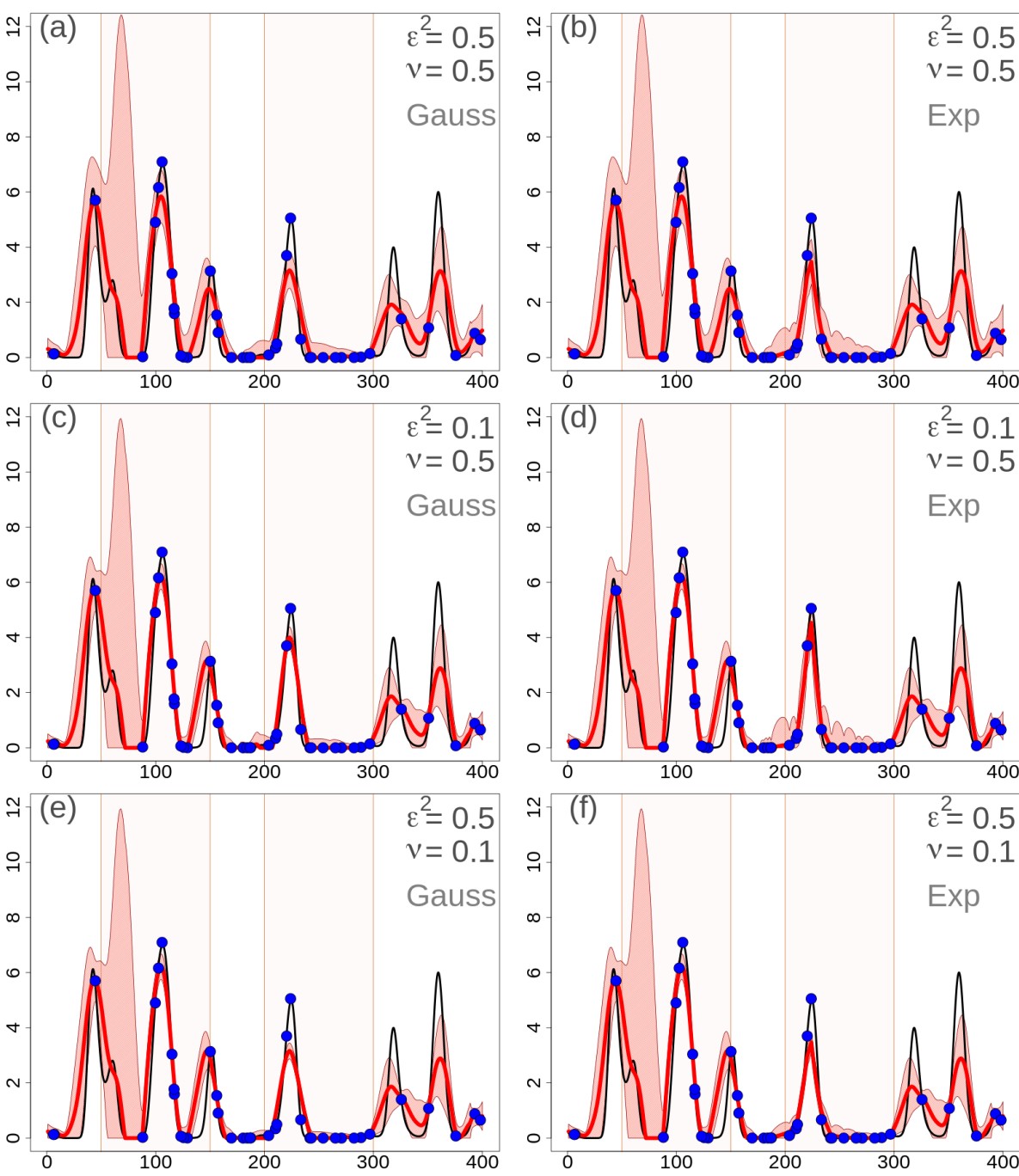

**Figure 5.** One-dimensional simulation in the original precipitation space (mm). Analyses at grid points with different EnSI-GAP configurations without applying the data transformation.. The layout is the same as in Fig. 3.

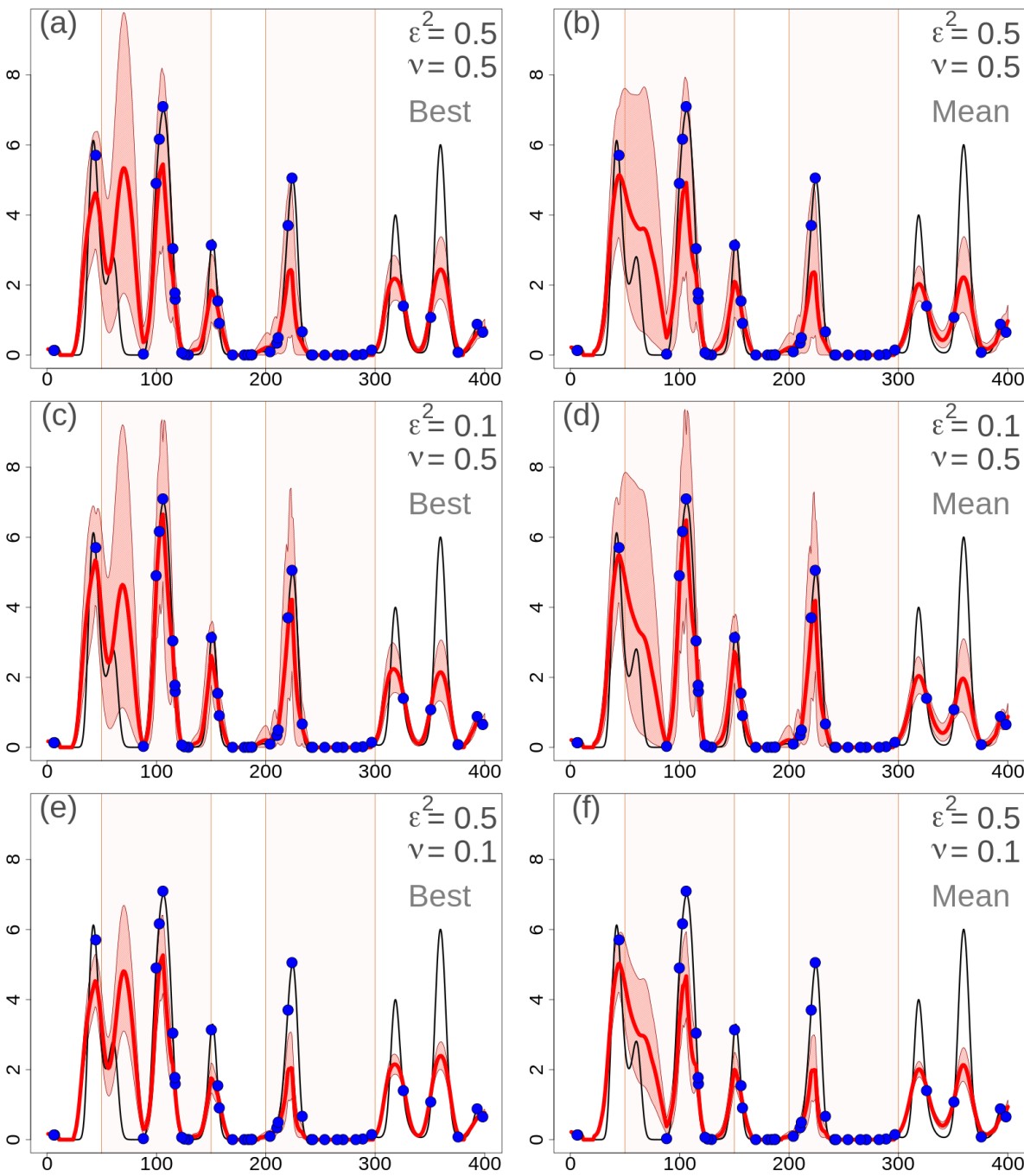

**Figure 6.** One-dimensional simulation in the original precipitation space (mm). Analyses at grid points with different EnSI-GAP configurations without considering the whole ensemble. Specification of the scale matrix $\mathbf{\Gamma}^{iu}$ through an exponential function. $\mathbf{\Gamma}^{if}$ is not used. The layout is similar to Fig. 3 except that here the panels in the left column show the results obtained considering as the background the best member of the ensemble, while in the right column the background is the ensemble mean.

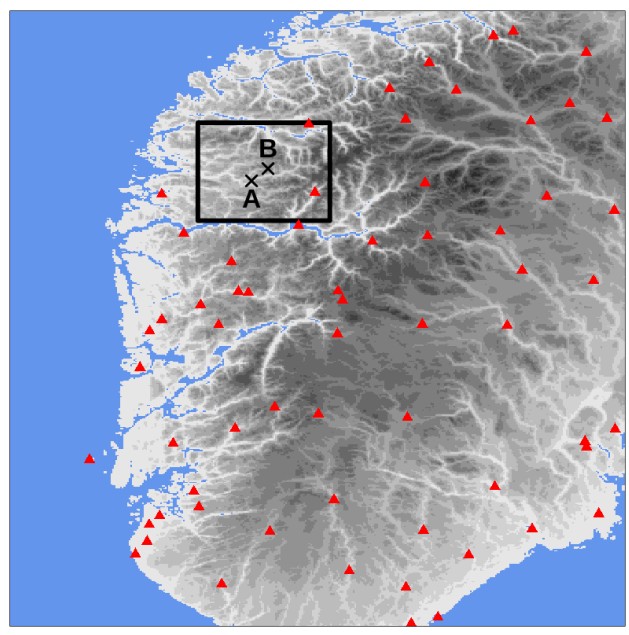

**Figure 7.** "South Norway" domain used in the simulations of Secs. 3.2- 3.3. The red triangles mark station locations used for cross-validation in Sec. 3.3. The gray shades indicate the altitude (from the lighter gray at 0 m to the darker gray at approximately 2400 m a.m.s.l.). The blue shade indicates the sea. The black box delimits the "Sogn og Fjordane" domain shown in Figs. 11- 12, the crosses mark the two points A and B used in the following.

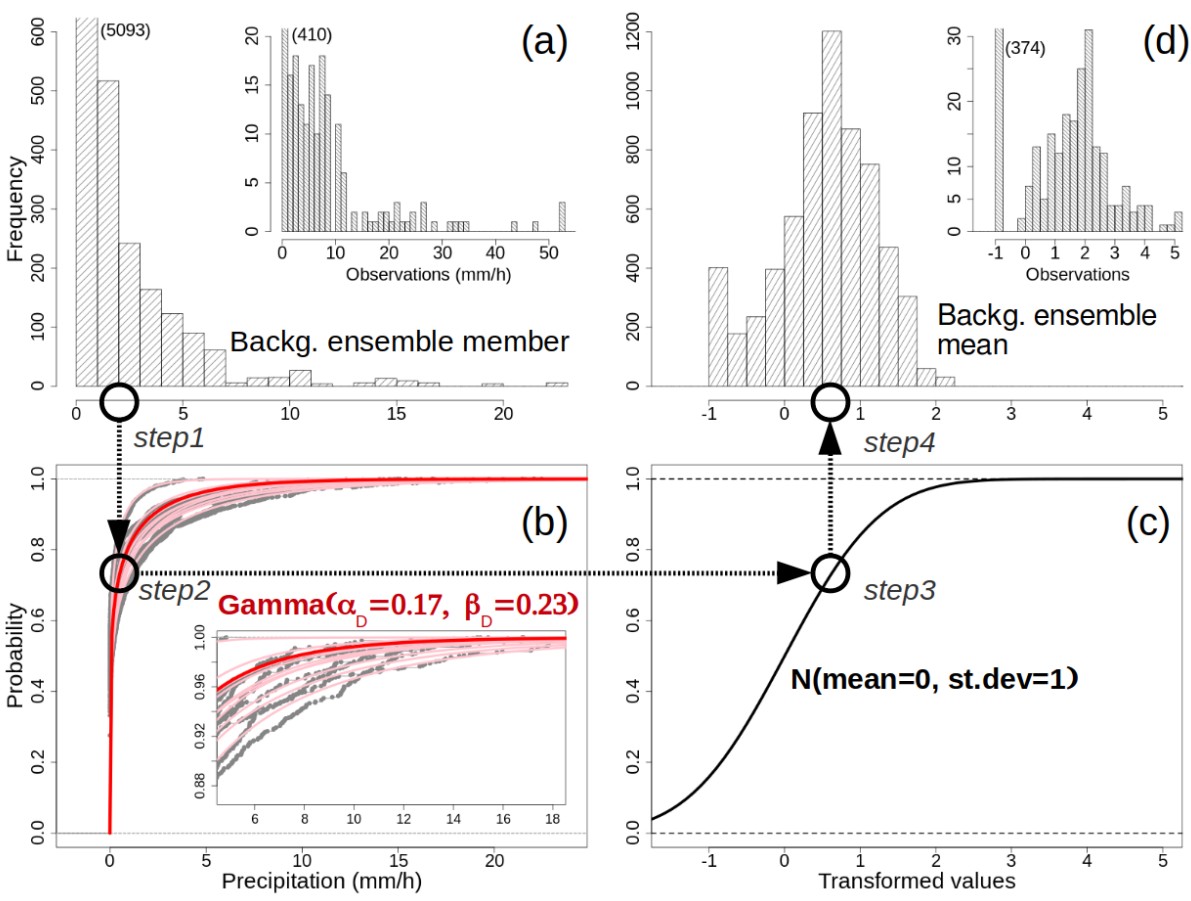

**Figure 8.** Data transformation procedure, example for 2019-07-30 15:00 UTC hourly precipitation totals over Sogn og Fjordane (see Fig. 7). Panel *a* shows the histograms with the frequencies of occurrence for: one member of the ensemble forecast and the observed values. The numbers in round brackets indicate the values of the truncated bins. Panel *b* shows the cumulative distribution functions (CDFs) for the 10 forecast ensemble members: the empirical CDFs are shown with gray dots, the best-fitting Gamma CDFs are shown as pink lines. The final Gamma CDF used in the Gaussian anamorphosis is shown with the red line and the parameters are reported. The inset on the bottom right shows an enlargement of a section of the main graph. Panel *c* shows the standard Gaussian CDF. Panel *d* shows the distributions of transformed values for the background ensemble mean and the observations. The 4 different steps of the data transformation for an arbitrary value of precipitation (approximately 2 mm/h) are indicated by circles and arrows.

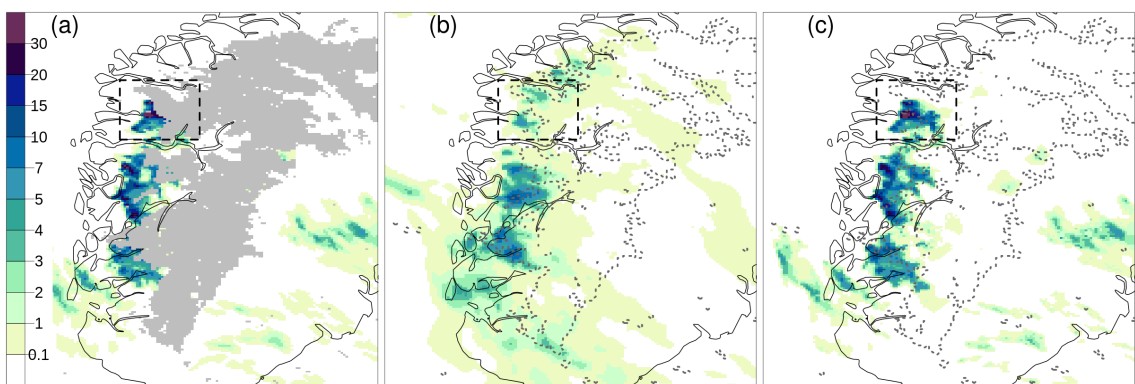

**Figure 9.** 2019-07-30 15:00 UTC, hourly precipitation totals (mm/h) over South Norway (see Fig. 7). Observations are shown in panel *a* over the same grid as the analysis. For each grid cell, the average of the observed values within the cell is shown. Grid points that are not covered by observations are marked in gray in panel *a* and the dashed gray lines in panels *b* and *c* delineate the boundary of the gray area shown in panel *a*. The background ensemble mean is shown in panel *b*. The analysis expected value is shown in panel *c*. The color scale is the same for all panels. The "Sogn og Fjordane" domain of Fig. 7 is shown as the dashed box.

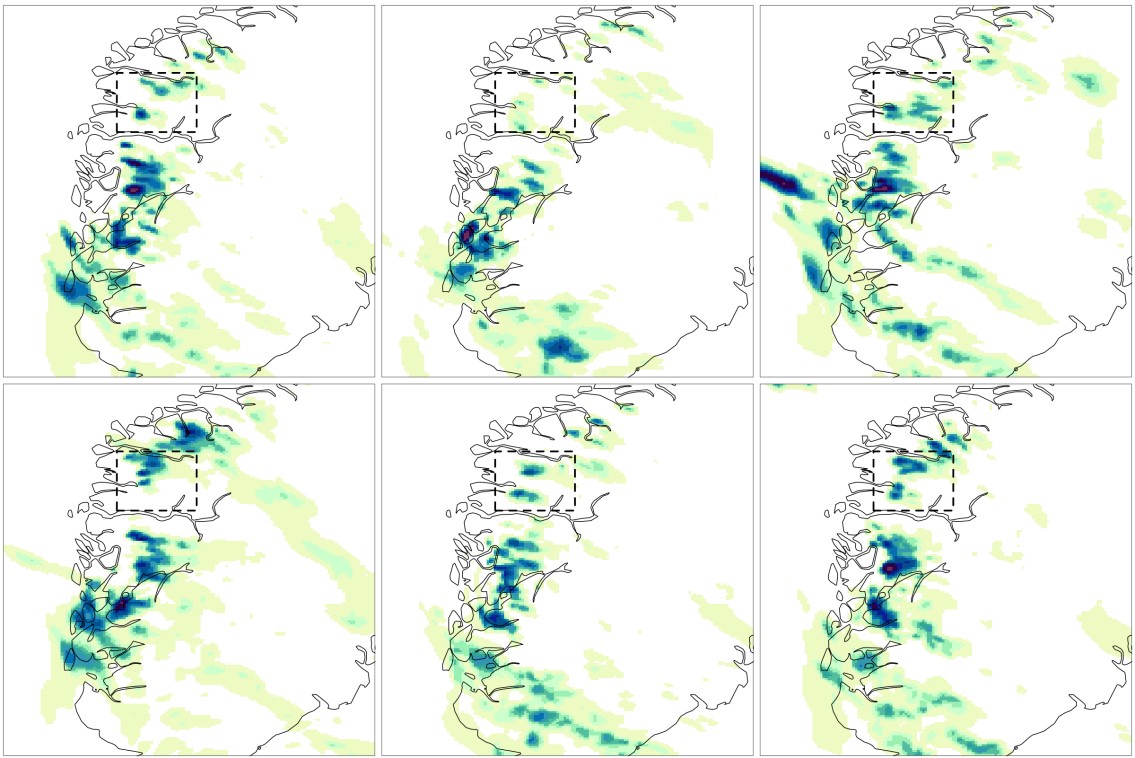

**Figure 10.** 2019-07-30 15:00 UTC, hourly precipitation totals (mm/h) over South Norway (see Fig. 7) for six of the ten background ensemble members. The color scale is the same as in Fig. 9. The "Sogn og Fjordane" domain of Fig. 7 is shown as the dashed box.

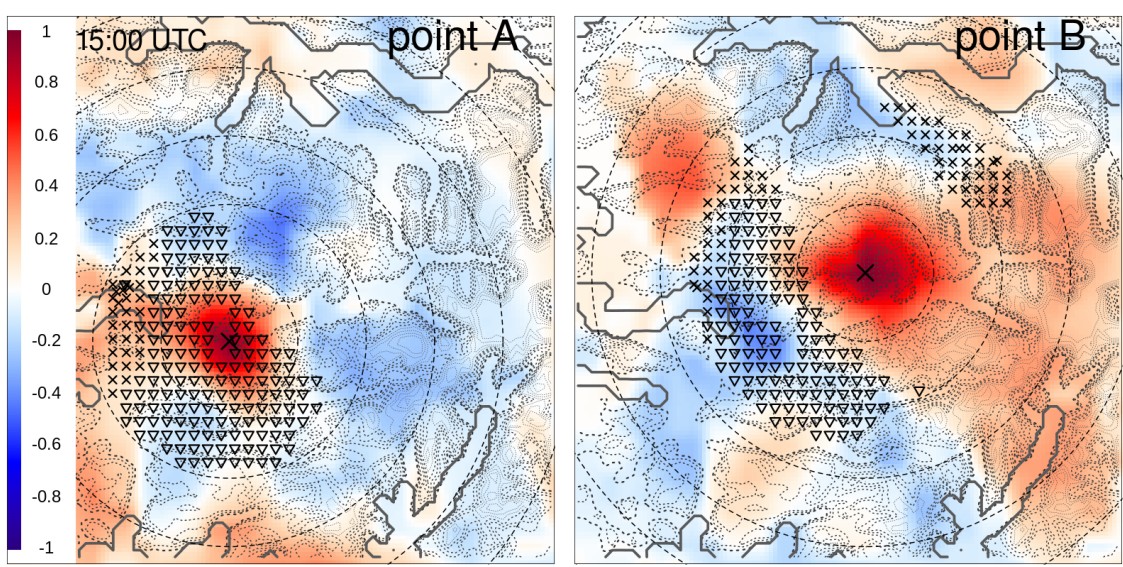

**Figure 11.** 2019-07-30 15:00 UTC, background error correlations $\mathbf{\Gamma}^{\mathrm{b}\,i}_{\phantom{b}i,:}$ of Eq. (25) used for spatial analysis of hourly precipitation totals over Sogn og Fjordane (see Fig. 7). The blue-red color shades show the background error correlations. With reference to Fig. 7, the left panel shows the background error correlations between point A and the grid points. For point B, the correlations are shown in the right panel. The symbols show the closest 200 observations, the triangles are observations of precipitation, while the crosses are observations of no-precipitation. The concentric circles have their common center at either point A or B and they are distance isolines at: 10 km, 20 km, 30 km, 40 km and 50 km. The thick dark gray lines delimit the fjords. The dashed lines are the contour lines for elevation: the thickest mark the 500 m isoline, the others have a gradually smaller thickness for 600 m, 700 m, 900 m, 1000 m, 1100 m, 1200 m, 1300 m, 1400 m, 1500 m.

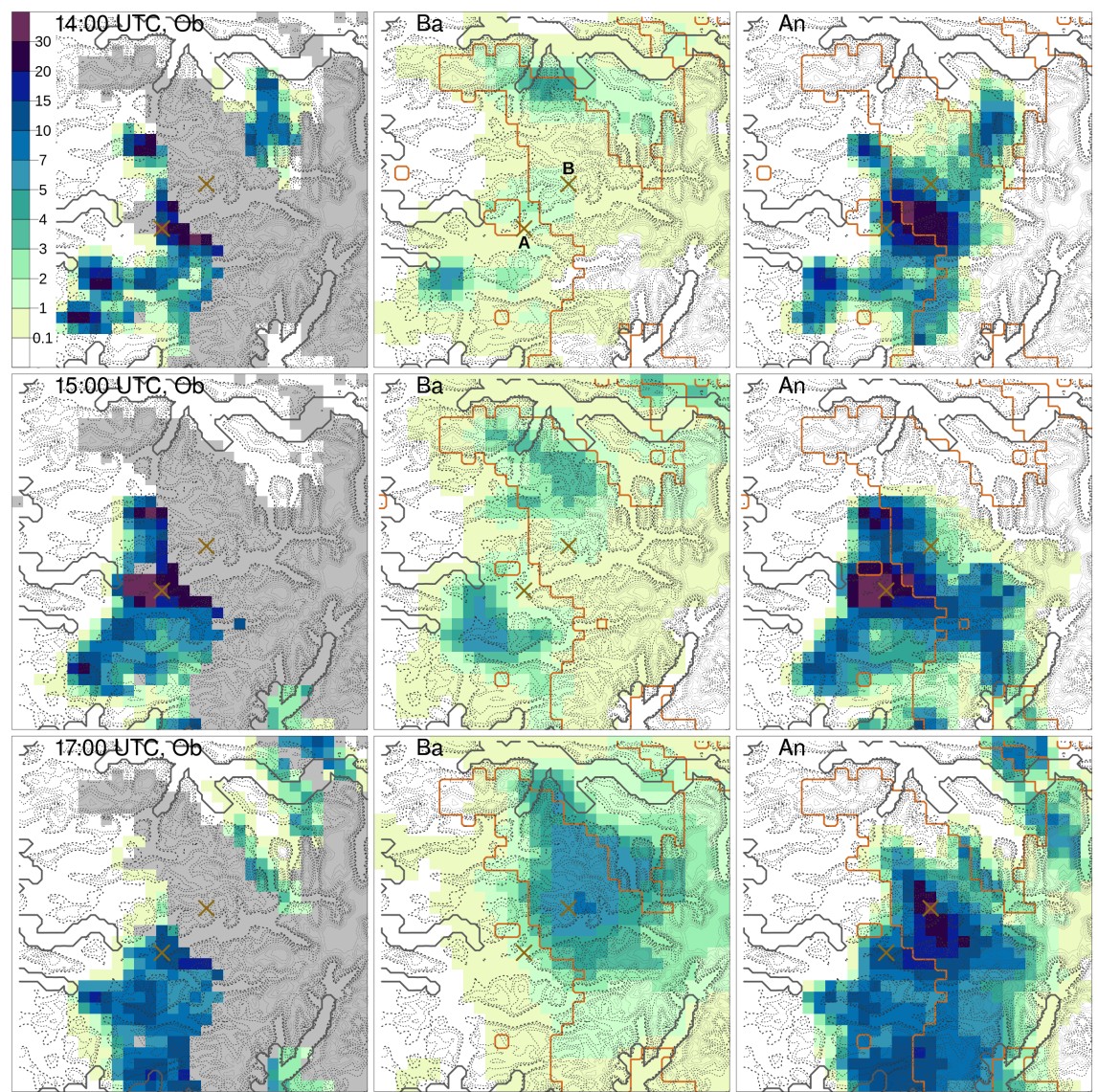

**Figure 12.** 2019-07-30 14:00 UTC (top row), 15:00 UTC (middle row), 17:00 UTC (bottom row) hourly precipitation totals (mm/h) over Sogn og Fjordane (see Fig. 7). The panels labeled with *Ob* (left column) show the aggregated observed values, as in Fig. 9. The panels with *Ba* (middle column) show the background ensemble mean. The panels with *An* (right column) show the analysis expected value. The crosses mark the A and B points of Fig. 7, which are also shown in the middle panel of the top row. The dark orange lines in panels *Ba* and *An* delineate the boundary of the gray area shown in panel *Ob*. The color scale is the same for all panels. The thick lines and the dashed lines have the same meaning as in Fig. 11.

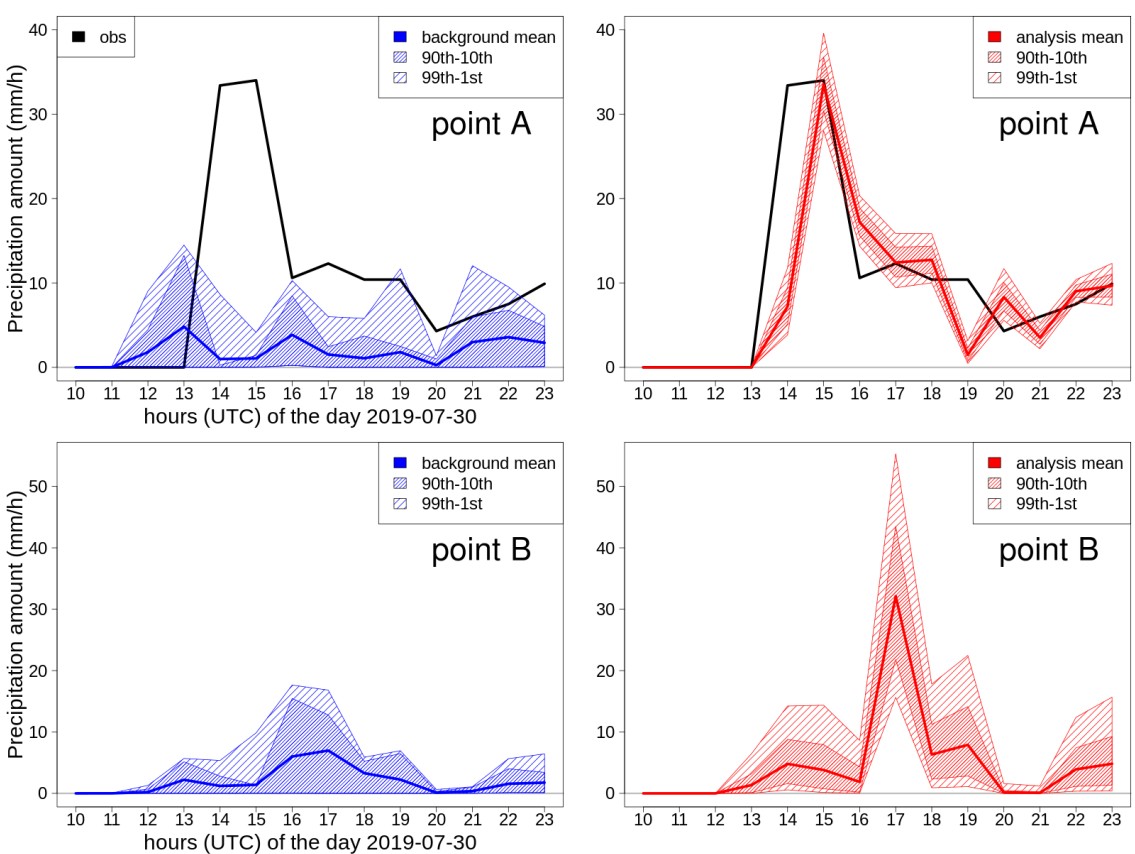

**Figure 13.** Time series of hourly precipitation totals for the period 2019-07-30 10:00 UTC to 23:00 UTC at points A (top row) and B (bottom row) of Fig. 7. The left panels show the background (blue). The right panels show the analysis (red). The blue (red) line shows the background (analysis) mean, the region with denser shading lines is the difference between the 90th and the 10th percentiles, the region with sparser shading lines is the difference between the 99th and the 1st percentiles. For point A, the closest observation, which is a radar-derived estimate, is shown (black line). Point B is in a region where observations are not available.

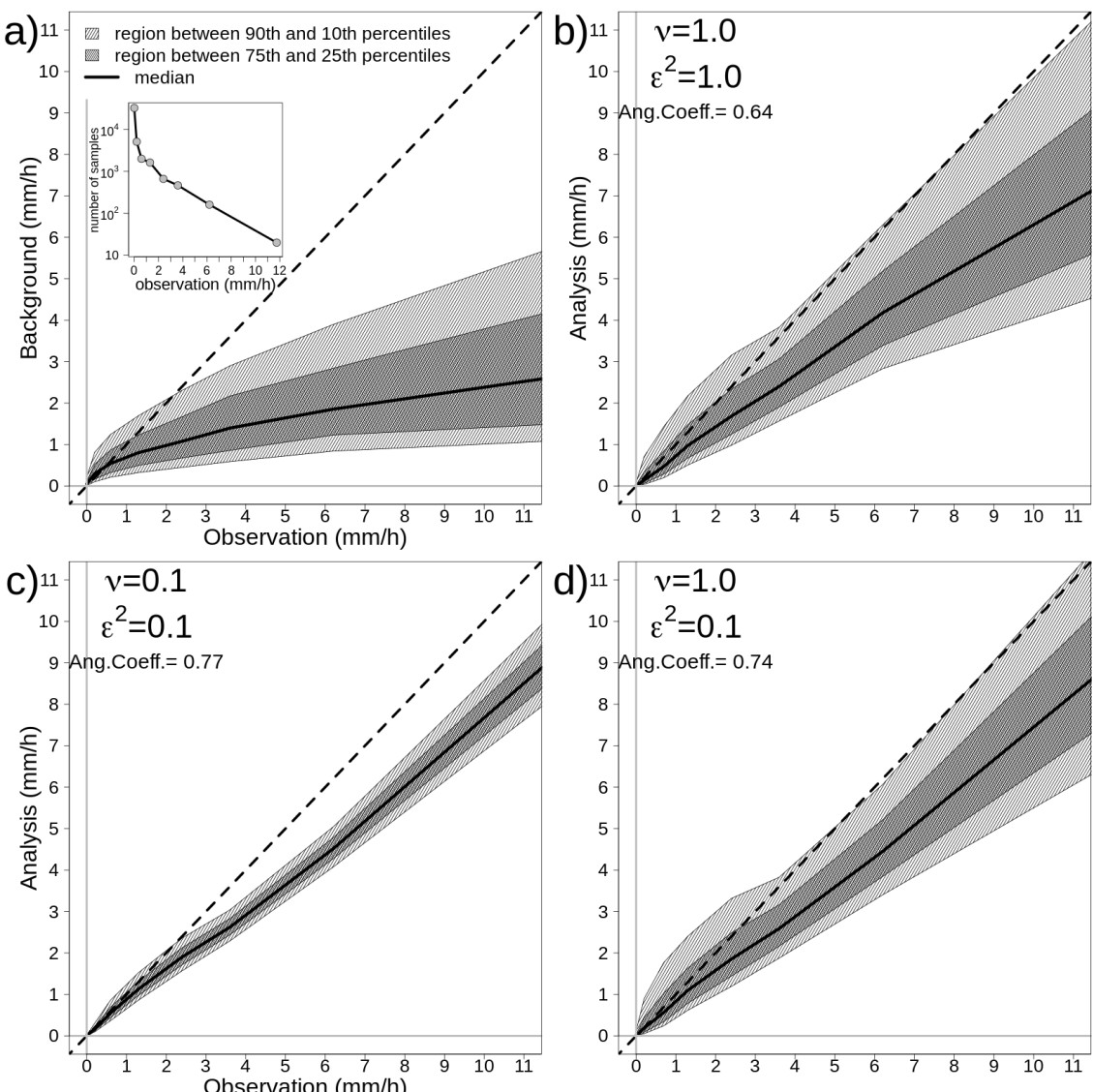

**Figure 14.** Summer 2019 hourly precipitation statistics for the cross-validation experiments. Panels: *a* background versus observations; *b* analysis $\varepsilon^2 = 1$ $\nu = 1$ versus observations; *c* analysis $\varepsilon^2 = 0.1$ $\nu = 0.1$ versus observations; *d* analysis $\varepsilon^2 = 1.0$ $\nu = 0.1$ versus observations. The independent observations have been divided into classes, the number of samples within each class is shown in the inset of panel *a*. Within each class and for each probabilistic prediction, several percentiles have been computed. The regions between the average of the 90th and the 10th percentiles are shown by light gray shades. The regions between the average of the 75th and the 25th percentiles are shown by dark gray shades. The thick black line indicates the average of the medians. The dashed black line is the diagonal (1:1) line. The angular coefficients of the best-fitting lines passing through the origins and better approximating the averages of the medians are shown, for the background in panel *a* it is 0.26 (not shown in the panel).

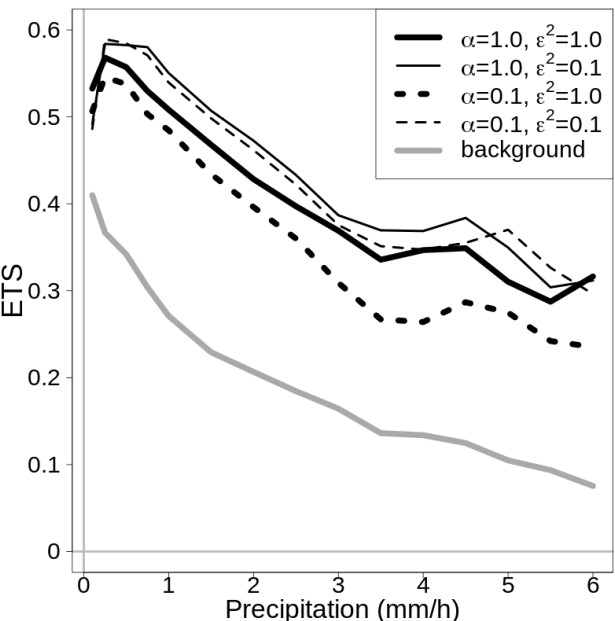

**Figure 15.** Equitable Threat Score (ETS) for summer 2019 hourly precipitation, as obtained through the cross-validation experiments. The black lines are the ETS curves for the analysis mean values, as indicated in the legend. The ETS curve for the background is the gray line. The precipitation thresholds defining the "yes" events are reported on the x-axis.

**Table 1.** Overview of variables and notation for global variables. All the vectors are column vectors if not otherwise specified. If $\mathbf{X}$ is a matrix, $\mathbf{X}_i$ is its $i$th column (column vector) and $\mathbf{X}_{i,:}$ is its $i$th row (row vector).

| symbol | description | space | dimension |
|---|---|---|---|
| $m$ | number of grid points | - | scalar |
| $p$ | number of observations | - | scalar |
| $k$ | number of forecast ensemble members | - | scalar |
| $\tilde{\mathbf{X}}^{\mathrm{f}}$ | forecast ensemble | original | $m$x$k$ matrix |
| $\mathbf{X}^{\mathrm{f}}$ | forecast ensemble | transformed | $m$x$k$ matrix |
| $\mathbf{x}^{\mathrm{f}}$ | forecast ensemble mean | transformed | $p$ vector |
| $\mathbf{A}^{\mathrm{f}}$ | forecast perturbations | transformed | $m$x$k$ matrix |
| $\mathbf{P}^{\mathrm{f}}$ | forecast covariance matrix | transformed | $m$x$m$ matrix |
| $\tilde{\mathbf{y}}^{\mathrm{o}}$ | observations | original | $p$ vector |
| $\mathbf{y}^{\mathrm{o}}$ | observations | transformed | $p$ vector |
| $\tilde{\mathbf{x}}^{\mathrm{t}}$ | truth | original | $m$ vector |
| $\mathbf{x}^{\mathrm{t}}$ | truth | transformed | $m$ vector |
| $\tilde{\mathbf{x}}^{\mathrm{a}}$ | analysis | original | $m$ vector |
| $\mathbf{x}^{\mathrm{a}}$ | analysis | transformed | $m$ vector |
| $\boldsymbol{\eta}^{\mathrm{a}}$ | analysis error | transformed | $m$ vector |
| $\mathbf{P}^{\mathrm{a}}$ | analysis error covariance matrix | transformed | $m$x$m$ matrix |
| $\boldsymbol{\sigma}^{\mathrm{a}}$ | analysis error standard deviation, $\sqrt{\mathrm{diag}\left(\mathbf{P}^{\mathrm{a}}\right)}$ | transformed | $m$ vector |
| $\mathbf{x}^{\mathrm{b}}$ | background | transformed | $m$ vector |
| $\boldsymbol{\eta}^{\mathrm{b}}$ | background error | transformed | $m$ vector |
| $\mathbf{P}^{\mathrm{b}}$ | background error covariance matrix | transformed | $m$x$m$ matrix |
| $\boldsymbol{\varepsilon}^{\mathrm{o}}$ | observation error | transformed | $p$ vector |
| $\mathbf{H}$ | observation operator | transformed | $p$x$m$ matrix |
| $\mathbf{L}$ | reference length scales for localization | transformed | $m$ vector |
| $\mathbf{D}$ | reference length scales of the scale matrix | transformed | $m$ vector |
| $\varepsilon^2$ | relative quality of the background wrt observations | transformed | scalar |
| $\nu$ | inflation factor | transformed | scalar |
| $\xi$ | small constant | original | scalar |
| $\alpha_D$ | shape of the gamma PDF used in the data transformation | original | scalar |
| $\beta_D$ | rate of the gamma PDF used in the data transformation | original | scalar |
| $\boldsymbol{\alpha}^{\mathrm{a}}$ | shape of the analysis gamma PDF | original | $m$ vector |
| $\boldsymbol{\beta}^{\mathrm{a}}$ | rate of the analysis gamma PDF | original | $m$ vector |

**Table 2.** Overview of variables and notation for local variables. All variables are specified in the transformed space. All the vectors are column vectors if not otherwise specified. If $\mathbf{X}$ is a matrix, $\mathbf{X}_i$ is its $i$th column (column vector) and $\mathbf{X}_{i,:}$ is its $i$th row (row vector).

| symbol | description | dimension |
|---|---|---|
| $p_i$ | number of observations in the surroundings of the $i$th grid point | scalar |
| $\overset{i}{\mathbf{H}}$ | observation operator | $p_i \mathrm{x} m$ matrix |
| $\overset{i}{\mathbf{R}}$ | observation error covariance matrix | $p_i \mathrm{x} p_i$ matrix |
| $\overset{i}{\mathbf{\Gamma}}^{\mathrm{o}}$ | observation error correlation matrix | $p_i \mathrm{x} p_i$ matrix |
| $\overset{i}{\mathbf{y}}^{\mathrm{b}}$ | background at observation locations | $p_i$ vector |
| $\overset{i}{\mathbf{P}}^{\mathrm{b}}$ | background error covariance matrix | $m \mathrm{x} m$ matrix |
| $\overset{i}{\mathbf{\Gamma}}$ | localization matrix | $m \mathrm{x} m$ matrix |
| $\overset{i}{\mathbf{V}}$ | localization between grid points and observation locations | $m \mathrm{x} p_i$ matrix |
| $\overset{i}{\mathbf{Z}}$ | localization between observation locations | $p_i \mathrm{x} p_i$ matrix |
| $\overset{i}{\mathbf{\Gamma}}^{\mathrm{u}}$ | scale correlation matrix | $m \mathrm{x} m$ matrix |
| $\overset{i}{\mathbf{G}}^{\mathrm{b}}$ | background error covariances between grid points and observation locations | $m \mathrm{x} p_i$ matrix |
| $\overset{i}{\mathbf{S}}^{\mathrm{b}}$ | background error covariances between observation locations | $p_i \mathrm{x} p_i$ matrix |
| $\overset{i}{\mathbf{G}}^{\mathrm{f}}$ | forecast error covariances between grid points and observation locations | $m \mathrm{x} p_i$ matrix |
| $\overset{i}{\mathbf{S}}^{\mathrm{f}}$ | forecast error covariances between observation locations | $p_i \mathrm{x} p_i$ matrix |
| $\overset{i}{\sigma}_o^2$ | observation error variance | scalar |
| $\overset{i}{\sigma}_b^2$ | average background error variance | scalar |
| $\overset{i}{\sigma}_{b\prime}^2$ | empirical estimate of $\overset{i}{\sigma}_b^2$ | scalar |
| $\overset{i}{\sigma}_f^2$ | average forecast error variance | scalar |
| $\overset{i}{\sigma}_u^2$ | error variance for the scale matrix | scalar |
| $\overset{i}{\sigma}_{ob}^2$ | sum of error variances (Eq. (11)) | scalar |

**Table 3.** Summary statistics on the evaluation of the 100 one-dimensional simulations. Results are presented for three modes: EnSI-GAP; "no transformation", which is EnSI-GAP without applying the Gaussian anamorphosis; "no ensemble", which is EnSI-GAP where the background is the ensemble mean and the background error covariance matrix is determined solely by the scale matrix. The configurations listed are the same that have been used in Figs. 3- 6 and the abbreviations have the same meanings (e.g., with reference to Fig. 3, the first row corresponds to panel *a*, the second to panel *b* and so on). The mean-squared error skill score (MSESS, Eq. (23)) is positively oriented with a perfect score being 1. The continuous ranked probability score ($\overline{\text{CRPS}}$, Eq. (24)) is negatively oriented with a perfect score being 0. For each configuration and score, the best values are marked with bold fonts.

| Mode | EnSI-GAP | | no transformation | | no ensemble | |
|---|---|---|---|---|---|---|
| Configuration | MSESS | $\overline{\text{CRPS}}$ | MSESS | $\overline{\text{CRPS}}$ | MSESS | $\overline{\text{CRPS}}$ |
| $\varepsilon^2 = 0.5$, $\nu = 0.5$, Gauss | **0.66** | **0.80** | **0.66** | 0.91 | 0.63 | 0.95 |
| $\varepsilon^2 = 0.5$, $\nu = 0.5$, Exp | 0.65 | **0.78** | **0.68** | 0.85 | 0.65 | 0.81 |
| $\varepsilon^2 = 0.1$, $\nu = 0.5$, Gauss | **0.70** | **0.79** | 0.68 | 0.95 | 0.65 | 1.01 |
| $\varepsilon^2 = 0.1$, $\nu = 0.5$, Exp | 0.71 | 0.72 | 0.71 | 0.80 | **0.73** | **0.71** |
| $\varepsilon^2 = 0.5$, $\nu = 0.1$, Gauss | 0.66 | **0.92** | **0.67** | 1.04 | 0.61 | 1.33 |
| $\varepsilon^2 = 0.5$, $\nu = 0.1$, Exp | 0.63 | **0.92** | **0.68** | 0.98 | 0.61 | 1.14 |

---

**Algorithm 1** EnSI-GAP: Ensemble-based Statistical Interpolation scheme with Gaussian AnamorPhosis. In this implementation, Gaussian functions have been used for correlation (and correlation-like) matrices. The symbols are listed in Tables 1- 2.

---

**Require:** forecast ensemble $\tilde{\mathbf{X}}^{\mathrm{f}}$ and ensemble mean $\mathbf{x}^{\mathrm{f}}$; observations $\tilde{\mathbf{y}}^{\circ}$; parameters $\varepsilon^2$, $\nu$, $\mathbf{L}$, $\mathbf{D}$

    Gaussian anamorphosis: estimation of $\alpha_D$ and $\beta_D$, then $\mathbf{X}^{\mathrm{f}} = g(\tilde{\mathbf{X}}^{\mathrm{f}})$; $\mathbf{y}^{\circ} = g(\tilde{\mathbf{y}}^{\circ})$

    Define additional global variables: $\mathbf{x}^{\mathrm{b}} = \mathbf{x}^{\mathrm{f}}$; $\mathbf{A}^{\mathrm{f}}$ : $\mathbf{A}_i^{\mathrm{f}} = \mathbf{X}_i^{\mathrm{f}} - \mathbf{x}^{\mathrm{f}}$

    **for all** grid points $\{i = 1, \ldots, m\}$ **do**

        select the closest $p_i$ observations $\overset{i}{\mathbf{y}}{}^{\circ}$ and obtain $\overset{i}{\mathbf{y}}{}^{\mathrm{b}} = \overset{i}{\mathbf{H}}\mathbf{x}^{\mathrm{b}}$

        {Dynamical background error covariance matrices, $\overset{i}{\mathbf{S}}{}^{\mathrm{f}} \approx (k-1)^{-1}(\overset{i}{\mathbf{H}}\overset{i}{\mathbf{\Gamma}}\overset{i}{\mathbf{H}}{}^{\mathrm{T}}) \circ [(\overset{i}{\mathbf{H}}\mathbf{A}^{\mathrm{f}})(\overset{i}{\mathbf{H}}\mathbf{A}^{\mathrm{f}})^{\mathrm{T}}]$}

        $\overset{i}{\mathbf{Z}}$ : $\overset{i}{\mathbf{Z}}_{jl} = \exp\left\{-0.5\left[d(\mathbf{r}_j, \mathbf{r}_l)/L_i\right]^2\right\}, j = 1, \ldots, p_i$ and $l = 1, \ldots, p_i$; $d()$ horizontal distance

        $\overset{i}{\mathbf{S}}{}^{\mathrm{f}}_{jl} \approx (k-1)^{-1}\overset{i}{\mathbf{Z}}_{jl}[(\overset{i}{\mathbf{H}}\mathbf{A}^{\mathrm{f}})_{j,:}(\overset{i}{\mathbf{H}}\mathbf{A}^{\mathrm{f}})_{l,:}], j = 1, \ldots, p_i$ and $l = 1, \ldots, p_i$

        $\{ \overset{i}{\mathbf{G}}{}^{\mathrm{f}}_{i,:} \approx (k-1)^{-1}(\overset{i}{\mathbf{\Gamma}}\mathbf{H}^{\mathrm{T}})_{i,:} \circ [\mathbf{A}_{i,:}^{\mathrm{f}}(\overset{i}{\mathbf{H}}\mathbf{A}^{\mathrm{f}})^{\mathrm{T}}] \}$

        $\overset{i}{\mathbf{V}}_{i,:}$ : $\overset{i}{\mathbf{V}}_{il} = \exp\left\{-0.5\left[d(\mathbf{r}_i, \mathbf{r}_l)/L_i\right]^2\right\}, l = 1, \ldots, p_i$

        $\overset{i}{\mathbf{G}}{}^{\mathrm{f}}_{i,l} \approx (k-1)^{-1}\overset{i}{\mathbf{V}}_{il}[\mathbf{A}_{i,:}^{\mathrm{f}}(\overset{i}{\mathbf{H}}\mathbf{A}^{\mathrm{f}})_{l,:}], l = 1, \ldots, p_i$

        {Background error covariance matrices}

        definition of $\langle \ldots \rangle$: $\langle \mathbf{c} \rangle = \sum_{l=1}^{p_i}(\overset{i}{\mathbf{V}}_{il}\mathbf{c}_l)/\sum_{l=1}^{p_i}(\overset{i}{\mathbf{V}}_{il})$, where $\mathbf{c}$ is a generic $p_i$ vector

        $\overset{i}{\sigma}{}_f^2 = \nu\langle\mathrm{diag}(\overset{i}{\mathbf{S}}{}^{\mathrm{f}})\rangle$; $\overset{i}{\sigma}{}_{ob}^2 = \nu\langle(\overset{i}{\mathbf{y}}{}^{\circ} - \overset{i}{\mathbf{y}}{}^{\mathrm{b}})^2\rangle$

        **if** $(\overset{i}{\sigma}{}_{ob}^2 = 0)$and$(\overset{i}{\sigma}{}_f^2 = 0)$ **then**

            $\mathbf{x}_i^{\mathrm{a}} = \mathbf{x}_i^{\mathrm{b}}$, then apply the inverse data transformation $\tilde{\mathbf{x}}_i^{\mathrm{a}} = g^{-1}(\mathbf{x}_i^{\mathrm{a}})$ and STOP

        **else if** $[\overset{i}{\sigma}{}_{ob}^2/(1+\varepsilon^2)] \le \overset{i}{\sigma}{}_f^2$ **then**

            $\overset{i}{\sigma}{}_u^2 = 0$; $\overset{i}{\mathbf{S}}{}^{\mathrm{b}} = \overset{i}{\mathbf{S}}{}^{\mathrm{f}}$; $\overset{i}{\mathbf{G}}{}^{\mathrm{b}}_{i,:} = \overset{i}{\mathbf{G}}{}^{\mathrm{f}}_{i,:}$

        **else**

            $\overset{i}{\sigma}{}_u^2 = \overset{i}{\sigma}{}_{ob}^2/(1+\varepsilon^2) - \overset{i}{\sigma}{}_f^2$

            {add the scale matrix $\overset{i}{\sigma}{}_u^2\overset{i}{\mathbf{\Gamma}}{}^{\mathrm{u}}$ to the background error covariance matrices}

            $\overset{i}{\mathbf{S}}{}^{\mathrm{b}}$ : $\overset{i}{\mathbf{S}}{}^{\mathrm{b}}_{jl} = \overset{i}{\mathbf{S}}{}^{\mathrm{f}}_{jl} + \overset{i}{\sigma}{}_u^2\exp\left\{-0.5\left[d(\mathbf{r}_j, \mathbf{r}_l)/D_i\right]^2\right\}, j = 1, \ldots, p_i$ and $l = 1, \ldots, p_i$

            $\overset{i}{\mathbf{G}}{}^{\mathrm{b}}_{i,:}$ : $\overset{i}{\mathbf{G}}{}^{\mathrm{b}}_{il} = \overset{i}{\mathbf{G}}{}^{\mathrm{f}}_{il} + \overset{i}{\sigma}{}_u^2\exp\left\{-0.5\left[d(\mathbf{r}_i, \mathbf{r}_l)/D_i\right]^2\right\}, l = 1, \ldots, p_i$

        **end if**

        Observation error covariance matrix: first $\overset{i}{\sigma}{}_b^2 = \overset{i}{\sigma}{}_f^2 + \overset{i}{\sigma}{}_u^2$, then $\mathrm{diag}(\overset{i}{\mathbf{R}}) = \varepsilon^2\overset{i}{\sigma}{}_b^2$

        {Analysis}

        $\mathbf{x}_i^{\mathrm{a}} = \mathbf{x}_i^{\mathrm{b}} + \overset{i}{\mathbf{G}}{}^{\mathrm{b}}_{i,:}(\overset{i}{\mathbf{S}}{}^{\mathrm{b}} + \overset{i}{\mathbf{R}})^{-1}\left(\overset{i}{\mathbf{y}}{}^{\circ} - \overset{i}{\mathbf{y}}{}^{\mathrm{b}}\right)$

        $(\boldsymbol{\sigma}^2)_i^{\mathrm{a}} = \mathbf{P}_{ii}^{\mathrm{f}} + \overset{i}{\sigma}{}_u^2 - \overset{i}{\mathbf{G}}{}^{\mathrm{b}}_{i,:}(\overset{i}{\mathbf{S}}{}^{\mathrm{b}} + \overset{i}{\mathbf{R}})^{-1}(\overset{i}{\mathbf{G}}{}^{\mathrm{b}}_{i,:})^{\mathrm{T}}$

        {Data back transformation}

        inverse transformation $g^{-1}$ of 400 quantiles of the distribution $N\left(\mathbf{x}_i^{\mathrm{a}}, (\boldsymbol{\sigma}^2)_i^{\mathrm{a}}\right)$

        $\boldsymbol{\alpha}_i^{\mathrm{a}}$ and $\boldsymbol{\beta}_i^{\mathrm{a}}$ are obtained by optimizing the fitting of a gamma distribution to the 400 quantiles through a least squares fitting method

    **end for**

---