# Peer review of "Ensemble-based statistical interpolation with Gaussian anamorphosis for the spatial analysis of precipitation"

_Nonlinear Processes in Geophysics, 2020_

## Referee Comment (RC1) · Anonymous Referee #1 · 3 Aug 2020

**Review of "Ensemble-based statistical interpolation with Gaussian anamorphosis for the spatial analysis of precipitation" by Lussana et al.**

August 3, 2020

The authors propose a statistical interpolation approach, EnSI-GAP, to provide hourly precipitation fields based on the optimal combination of surface observations, radar QPE and NWP. They use standard hybrid data assimilation (DA) methods to achieve this objective, with a static and a dynamic component in the background error covariance matrix. The study differs from standard DA on several points. First, the precipitation are transformed using Gaussian anamorphosis, instead of the classical log or cubic transforms. Second, they fit a theoritical distribution to hourly precipitation using the two-parameter Gamma distribution. Finally, the static component of the background error covariance matrix is estimated using different hypotheses on the background and observation uncertainties. The authors evaluate the EnSI-GAP on a synthetic dataset, on a specific extreme event and over a time period with a cross-validation framework.

**General Comments:**

The methodology is well organized and is clearly explained in a very educational way. However, the fitting of a gamma distribution and the back transform of the precipitation could be more explicit (see comments below). We can guess what is done when reading the applications but it is preferable if it appears earlier in the paper.

How do you think this approach perform during the cold season with solid precipitation ? The reporting gauges are less numerous and the radars are difficult to use but NWPs have better performances for large scale precipitation do I was curious about the performances. Moreover, how do you think your tunable parameters will behave ?

A broader question is how do you think your approach is better than a standard hybrid approach that uses as a static covariance based on 2DVar or other DA approaches (see for example, Hamill et al. 2000), that has the advantage of not tuning parameters based on some assumptions ?

**Specific comments:**

L5-6, page 1: Could you be more specific about the additional source of uncertainty that is used ?

L35, page 2: The word "prediction" may be inappropriate as not forecasts are provided by the approach.

L34-36, page 2: When you say "The objective of our study is the precipitation reconstruction through the combination of numerical model output with observations

from multiple data sources... data sources", you define the general objective of data assimilation approaches. Could you be more specific on the objectives of the study ?

L47-49, page 2: The Lundquist et al., 2019 study does not demonstrate that, generally, the performances of NWP models in representing orographic precipitation are better than observations. It does for annual precipitation amounts, it also depends on the model, its resolution and configuration. Plus, this statement is in contradiction the working assumption 3.

L79, page 3: A reference supporting the gamma distribution assumption for the hourly precipitation must be added. You should be careful when you mention the extremes and the IDF curves in L42-44, as the Gamma distribution is known to be inappropriate for such events.

L99-100, p4: As it is an innovative part, maybe you could give more details on "opportunistic sensing networks".

L110, page 4: Maybe you could mention that the temporal autocorrelation of hourly precipitation field exist and is insured by the background fields.

L137-140, page 5: This part is a bit confusing, do you fit a Gamma distribution on hourly precipitation and for each grid-cell independantly over a time period? Indeed, we can see in the introduction in L70 that "the aim at estimating its shape and rate parameters for each grid point", suggesting different Gamma parameters at each grid cell. But, in the conclusion, we can read in L654 that "the Gaussian anamorphosis [...] is based on the same gamma distribution parameters for the whole domain.".

L630, page 20: Rephrase the following sentence "The independent observations are used to judge if an event does in fact obtain."

L649-651, page 20: We can see that performances decrease for the BSS and the ETS for precipitation strictly above 1mm/h. Are you seeing similar results when investigating the reliability for higher precipitation thresholds ? Consistency bars as suggested by Brocker and Smith (2007) in the reliability diagram could help better assess the reliability.

Figure 6, page 33: Please, add the unit to the figure and explain the gray color in the caption (as in the text).

---

## Referee Comment (RC2) · Anonymous Referee #2 · 26 Aug 2020

My field of expertise covers data assimilation and statistical analysis, but not meteorology, so my review will not cover the meteorological aspects of the work neither the realism of the fields nor bibliographic references within weather forecast post-processing.

The manuscript adresses the important problem of probabilistic analysis of precipitations and presents a new method (at least new to my knowledge). The study has borrowed methods and terminology from the ensemble data assimilation literature to perform post-processing of EPS outputs. The main originality of the manuscript seems to lie on the use of EPS output as background covariance for the statistical interpolation, which is precisely the modus operandi of the Ensemble Kalman Filter, except

without the need to re-use the analysis as initial condition for the following forecast. Since the ensemble data assimilation framework is also using some concepts from the statistical interpolation literature, the terminology can be confusing at times. I have tried to correct that below. Rather than assimilating data, method is more intended as a "conditional simulation" in the geostatistical literature, it is performing a Kriging of the mean of the Gaussian transformed values and then retrieves the probability distribution of precipitation at every location using a predefined Gamma distribution. I wish such a summary of the method had been spelled out more clearly in the article, however.

The manuscript is otherwise well articulated, moving logically from the algebraic formalism to an idealised case and a real application over Southern Norway. The method seems to work very well according to cross-validation results against independent weather stations data, which is reassuring, but not a proof that the method is flawless.

My main weaknesses of the manuscript are the following (more details below):

- The paper is introducing a new method, but does not give a point of comparison to any pre-existing methods other than references to the data assimilation literature. The literature review of post-processing methods seems to be missing (and I am not an expert of it). As the paper stands, the most natural comparison should be static versus flow-dependent covariances, but that is not done in the paper. One extra experiment with alpha equals to zero would seem necessary to compare the new method to a reference method, albeit very basic.

- The benefits of flow-dependent covariances are not exhibited other than showing their existence. The ensemble does hold information about the physics of precipitation that is absent from the stationary static covariance and I would expect some meteorological interpretation of the ensemble correlation.

- The algorithm description is at times unclear, in particular the adjustment of the Gaussian anamorphosis and the need for the alpha stabilisation parameter. - The testing focuses excessively on two values of the alpha and epsilon parameters but leaves other
choices - however critical - under diffuse justifications (the D radius, the dilemma between Gaussian and exponential covariances). The authors have generally a hard time in comparing their results, possibly assuming that excessively solid evidence is needed to choose between two options. This causes an imbalance in the article between the testing of alpha and epsilon versus all of the other choices that felt more interesting to me.

- The choice of the Gamma distribution could have been supported by a histogram of observations.

- The choice of error covariances could have been settled by plotting the experimental variogram of the errors against the densest observation source (the radar). The shape of the variogram at its origin is the critical choice for kriging and should be adjusted manually against whichever of the Gaussian or exponential function that fits best. Long ranges of the variogram can be ignored for the sake of this study. The adjustment of the variogram will not only reveal which of the two functions is most representative of precipitation but will also deliver the range parameter which can justify the choice of the reference length scales L and D.

- The Working Assumptions are expressed as vague principles but do not determine the algorithmic choices made thereafter. Reformulating them as statistical hypotheses (temporal independence, spatial scales, monotonicity) would make an easier read.

- The paper can be shortened in several places.

Using the Gaussian covariance function for the unpredicted scales Gamma\_u is risky in cases where Pf is equal to zero (no rain the background EPS, but rain in observations). This is not visible in the toy experiment (Figure 2b) because the test is only considering interpolation cases but not extrapolation (observations on one side only). The Gaussian covariance has indefinitely many zero derivatives at the origin, which imposes unrealistic smoothness constraints on the analysis and, as a side effect, causes an "over-confidence" in the analysis: underestimation of the analysis uncertainty and a
tendency to produce high and low values outside the range of observations. Figure 2 (b and d) show that the uncertainties are indeed much smaller with the Gaussian than the exponential covariance (in the "missed" case between points 250 and 300) but is not decisive because both experiments seem to cover well the observations. Looking at these results, I believe that the Gaussian covariance makes less sense qualitatively because the uncertainties are lower when the model predicts no rain at all than only a little rain. As if one should be more confident when the model is completely wrong (no rain) than when it does not predict the right quantity of rain.

If in addition the observation errors R had been zero, the authors would also have experienced problems with the inversion of the (S+R) matrix. The risks related to the use of a Gaussian covariance are described in Diamond and Armstrong (1984).

Overall I have the impression that the method presented in the manuscript is useful and will make an interesting contribution to the field of meteorology and possibly beyond.

Detailed comments:

- The abstract does not mention that the realistic examples are using data over Norway. This could be important to know for the readers.

- I49: The statement that the forecast is better than observations in the absence of observations seems odd to me.

- I81: I would refer to the Ensemble Kalman Filter rather than the Ensemble Optimal Interpolation because the latter uses ensemble members taken at different times, while ensemble members in your method are synchronous.

- Bertino et al. (2003) introduced the concept of Gaussian anamorphosis from geostatistics to data assimilation, but since you are not performing any data assimilation I would think that a general reference on geostatistics is more appropriate: see Chilès and Delfiner (2012), Chapter 6,

- I82: trans-Gaussian would deserve a definition, or to be called explicitly "transformed

**NPGD**
Gaussian".

- I85: The authors seem to refer to the "static" covariance matrix rather than "scale". I suggest you call it "static" by opposition to the flow-dependent ensemble covariance term.

- I114: anisotropy and non-stationarity are two different topics. I agree that precipitation is non-stationary but invoking the non-stationarity of weather phenomena or simply the orography would have seemed a better explanation.

- I135: Specify the direction of the function: X is the Gaussian variable, not X tilde.

- I139: The fitting of the Gaussian anamorphosis is unclear: it is adjusted to "each ensemble member" but does not state which time period nor which geographical region. This is made further puzzling in Algorithm 1, as I read that the adjustment was part of the loop "for all grid points", which implies a local adjustment at each analysis time.

- I160: Include a reference to Table 1 somewhere in the first paragraph.

- Section 2.2.2: The working assumption are not mathematically well posed. They read more as general principles than working assumptions. WA3 could be rephrased as "R and Pf are proportional" for example. An assumption of locality is also missing together with WA5 on the temporal independence.

- I204: As above, "static" seems more appropriate than "scale" when you have other meanings for the word "scales" in the same paper.

- I248: The "stabilisation factor" alpha is poorly introduced (what is it suppose to stabilize?) and has no other effect than to increase the variance of the ensemble. I would therefore call it an inflation factor although it does not have the same effect as in an EnKF due to the absence of recursions. If the term "inflation" is also used in the literature related to ensemble post-processing, then I would suggest you call it "inflation factor". NPGD
- I258: The reasons for introducing alpha have not been enunciated before.

- Eq (13): please insist that the analysis is applied to the ensemble mean and not each member of the background ensemble. I have lost track of the definitions given a few pages earlier.

- I277: As above, refer to the EnKF rather than the EnOI here.

- I284: The authors should state more explicitly that the method "does nothing" in this case.

- I285: remove "direct" before "inverse" in the whole paragraph.

- Section 2.3 comes back to the anamorphosis and the Gamma distribution and it is not clear whether the optimization of the Gamma parameters are a repetition of what was done in 1139 or if the back-transformation is using a different fit than the direct transformation in Eqs (1) and (2). Please rewrite this paragraph.

- Section 3.1.1: The section is very long and it would help if simple concepts as "false positive" and "a miss" were used instead of "an alternative truth" which does not convey appropriately that the authors are dealing with ensemble bias.

- I347: This procedure seems to be a convolution between the signal and a smoother random signal. This does not low-pass the signal as a moving average would do. Please clarify.

- I352: Between "200 u and 300 u": Figure 1b rather indicates 220 than 200.

- I360: Is "the multiplicative model" compatible with the assumption of a Gamma distribution?

- I370: It seems that the length scale of the unresolved covariance is dependent on the measurement network but not the physics of precipitation. This comes as a surprise and should have been clarified upfront as a working assumption in Section 2.2.

**NPGD**
- I387: These three references seem to imply that the exponential covariance is more common than the Gaussian for precipitation.

- I394: Variations in the scale matrix -> the choice of the static covariance matrix.

- I401: remove "almost" since Pf is equal to zero.

- I402-404: Keep this remark until after Figure 4 has been introduced.

- I428: Why restraining yourselves with a qualitative judgement when you could have computed a CPRS or a simple ratio of errors versus estimator uncertainty?

- I436-442: I believe this part can be shortened.

- Figures 6 and 7 should include the Sogn og Fjordane box.

- I.495 Where observations are absent (gray shaded areas), the analysis differs from the background. Please comment on whether this change is realistic or not.

- I508: Kriging does induce some systematic smoothing, the authors could have made a note of that when discussing Eq (13). Smoothing can be avoided by other methods in the family of conditional simulations.

- I 522: The extrapolated values are not commented, are these expected or not?

- I531: This is a very long sentence to say "certainly no rain".

- I532-534: This passage does not seem to be connected with the rest of the text, please make a point or remove it.

- I543: Which "EnSI-GAP settings"? There are several of them.

- I. 549: Is it ok that the analysis rules out the precipitation at point B where no observations are present?

- I551-555: As above, the take-home message of these numbers is not obvious, please shorten the passage or remove.

**NPGD**
- I560: I have been confused by the contradiction between (i) and (ii). Please rephrase.

- Section 3.2: the case "alpha=0.1" can be called a case of "deflation" by opposition to "inflation".

- Figures 12 to 14 seem to give the same message than Figure 11. I believe that they can be removed together with the related text without loss to the argument.

- I679: The justification of the choice of D value should come upfront in Section 2 rather than in the discussion.

- I685: Why "assuming" that the background ensemble is more likely to overestimate the spread wile you could have verified it with the data at hand?

- Algorithm 1, under "Require": include the calculation of the ensemble mean and the predefined anamorphosis function.

- Figure 1: vertically shade areas of "false alarms" and "miss".

- Figure 6: delineate the boundary of the grey area in panel (c) to highlight what the analysis does in the absence of observations. Do that as well in Figure 8.

- Figure 8: annotate the points "A" and "B" on the figure.

- Figure 10: The flow-dependence of the correlations is visible, but does not speak much for itself. The ensemble may contain some information about the orography for example and it could be interesting to include the elevations with a few isolines.

- Figure 11: An integrated measure of the goodness-of-fit can be included in the legend of each panel.

Typos and minor issues

- I197: tenths -> tens

- I395: remove "the" before "higher uncertainties".

NPGD
- Use "anamorphosis" instead of "anamorphism". I don't think "anamorphism" is grammatically wrong but it is less usual in the literature.

- I499: "clearly" can be removed. Let the reader judge the quality of the figure.
- I505: "it is evident a sharp gradient..." -> "a sharp gradient [...] is evident".
- I539: "sort of outliers" -> "outliers".
- I617: "much accurate" -> "much more accurate"
- I618: "those" -> "that"

References:

Chilès, J.P. and P. Delfiner (2012) Geostatistics: Modeling Spatial Uncertainty. Wiley and sons. https://onlinelibrary.wiley.com/doi/book/10.1002/9781118136188

Diamond, P., and Armstrong, M. (1984), "Robustness of Variograms and Conditioning of Kriging Matrices," Mathematical Geology, 16, 809–822.

**NPGD**

---

## Editor Comment (EC1) · Alberto Carrassi (Editor) · 27 Aug 2020

Dear Authors,

I think the reviewers have done an excellent job in identifying some aspects that, if addressed properly, can lead to a further better version of your manuscript.

Please provide your answers to their comments and the description on how you would modified the manuscript if needed. I look forward to receiving your comments.

With best regards,

Alberto

---

## Author Comment (AC1) · 21 Sep 2020

Dear Editor, Dear Reviewers,

Thanks for your thoughtful and thorough work in revising our manuscript. We are grateful to the reviewers for their kind comments and insightful criticisms that will help us in improving the quality of our work.

In some of their comments, both reviewers share the same concerns. We will take extra care in addressing those issues, such as: imprecisions in the descriptions of some of the references reported in the Introductions (e.g. Lundquist et al., 2019); the

algorithm description, that is at times unclear, especially with respect to the gamma transformation; the lack of a comparison between EnSI-GAP and other spatial interpolation methods, such as those based on static covariance matrices. This last issue is particularly relevant since the inclusion of such a comparison will allow us to show the benefit of using ensemble-based covariance matrices (fully time dependent and spatially inhomogeneous) with respect to static covariance matrices.

About the general comments of reviewers 1 and 2:

In the first paragraph of her/his communication, reviewer 2 offers a summary of the spatial analysis method from the point of view of a geostatistician. We are grateful to the reviewer for providing this summary and we will include it in our Introduction.

Reviewer 2 writes "The method seems to work very well according to cross-validation results against independent weather stations data, which is reassuring, but not proof that the method is flawless.". Similarly, in her/his general comments, reviewer 1 suggests the application of EnSI-GAP to a winter case study. Then, both reviewers express doubts about some of our choices about parameter settings.

The main focus of the manuscript is in the method description. For this reason we have included Section 3.1.1 on a one-dimensional simulation, which gives the readers a better idea (a) on how the method works (b) different configurations of the method are possible and they should be explored -and exploited- when applying EnSI-GAP. The method is based on inverse problem theory. In this sense, we think EnSI-GAP theory is solid enough that the method can be considered flawless and it can be used to combine different data sources in a well defined theoretical framework. However, a distinction needs to be done between the general theory and its application to specific problems, where implementation choices must be made. It is possible that some implementation choices, for some very specific situations, may result in EnSI-GAP delivering unexpected results (e.g. a null R matrix may lead to problems in the applications, as discussed below). This issue is common to all spatial analysis methods, we
believe, think for example to the application of Kriging with a "wrong" semi-variogram. We will try to add warnings in the text on the implementation choices that should be avoided. In addition to describing the method, we wanted to show that the method has been validated under conditions that are relevant for operational applications, such as the case study of intense precipitation presented in Section 3.1.2. At the same time, it was important for us to show that EnSI-GAP was able to provide reasonable results not just for some specific days but for longer time periods and for this reason we have considered the summer season of 2019, when the method can be tested under ideal conditions because of the availability of a dense observational network. We have chosen summer, instead of winter, because convective precipitation -if compared to stratiform precipitation- is characterized by a larger variability over smaller spatial scales. The NWP models perform generally worse in representing precipitation. As a consequence, data collected over (the same) summer season represents a consistent dataset (no significant variations in the observational network or NWP settings) and at the same time a collection of a number of challenging cases for our spatial analysis. The application of EnSI-GAP over the winter season would be an interesting application indeed. However, one that cannot be compared straightforwardly to the results obtained for the summer season because of the huge differences in the observational network (e.g. almost all citizen observations are not available since the instrumentations they use do not provide for sensor heating). Besides, raingauge observations of wintertime precipitation in Scandinavia are often adjusted for wind-induced under-catch, which is especially significant for solid precipitation. This constitutes an additional source of uncertainty that is difficult to quantify. The issues mentioned above do not impact that much on the methodology but they would require different choices for the parameter settings. A complex topic such as the application of EnSI-GAP to different climatic conditions, when at the same time the observational network and the model performances vary significantly, deserves an article of its own. In the case of our research team, we are currently working exactly on these tasks and our ultimate goal is to create a gridded dataset of hourly precipitation covering a large domain over

Scandinavia.

the list of the main changes we will make to the manuscript follows: The comparison of results against a spatial analysis method based on a static covariance matrix for the background error will be included (Section 3.2) Revision of Section 2.1 to improve the description of the estimation of the Gamma parameters Add an histogram of the observations to support our choice of a Gamma distribution Reformulate the working assumptions (Section 2.2.2) We will refer to a "static matrix" instead of using the term "scale matrix" We will refer to an "inflation factor" alpha instead of using the term "stabilization factor" Some of the Figures will be modified according to the reviewers suggestions (see point-by-point response)

Point-by-point response to reviewers' comments follows.

Best Regards,

Cristian Lussana on behalf of the Authors

======================================================================
Excerpt from reviewer1's letter (the reviewer's comments are reported in Italic)

[...] Specific comments:

*L5-6, page 1: Could you be more specific about the additional source of uncertainty that is used ?*
**Answer:** We will add "additional source of uncertainty based on the deviations between background and observations..."

*L35, page 2: The word "prediction" may be inappropriate as not forecasts are provided by the approach.*
**Answer:** We will replace "prediction" with "representation".

*L34-36, page 2: When you say "The objective of our study is the precipitation reconstruction through the combination of numerical model output with observations from*

*multiple data sources... data sources", you define the general objective of data assimilation approaches. Could you be more specific on the objectives of the study?*
**Answer:** One of the main differences between spatial analysis and data assimilation is that data assimilation aims at providing the initial conditions for a model run. Lines 34-44 state the objectives of our study.

*L47-49, page 2: The Lundquist et al., 2019 study does not demonstrate that, generally, the performances of NWP models in representing orographic precipitation are better than observations. It does for annual precipitation amounts, it also depends on the model, its resolution and configuration. Plus, this statement is in contradiction to the working assumption 3.*
**Answer:** We will rephrase our statement.

*L79, page 3: A reference supporting the gamma distribution assumption for the hourly precipitation must be added. You should be careful when you mention the extremes and the IDF curves in L42-44, as the Gamma distribution is known to be inappropriate for such events.*
**Answer:** We will add here a reference to Wilks's book "Statistical methods in the atmospheric sciences" (now it is reported at line 108). In addition, we will make clear at lines 42-44 that we are not suggesting to use Gamma distributions to model extremes. Sometimes, people use gridded datasets -such as those derived by using statistical methods- to infer extreme precipitation values for a region, then apply extreme value theory to compute IDF curves.

*L99-100, p4: As it is an innovative part, maybe you could give more details on "opportunistic sensing networks".*
**Answer:** We will include here specific references to the Sections in the manuscript where the use of citizen observations is described and we will expand that paragraph.

*L110, page 4: Maybe you could mention that the temporal autocorrelation of hourly precipitation fields exist and is insured by the background fields.*

**Answer:** Will do.

*L137-140, page 5: This part is a bit confusing, do you fit a Gamma distribution on hourly precipitation and for each grid-cell independently over a time period? Indeed, we can see in the introduction in L70 that "the aim at estimating its shape and rate parameters for each grid point", suggesting different Gamma parameters at each grid cell. But, in the conclusion, we can read in L654 that "the Gaussian anamorphosis [...] is based on the same gamma distribution parameters for the whole domain.".*

**Answer:** Both reviewers found this part confusing. This is a clear signal that it needs to be rewritten. We should better communicate when we are referring to either one of the following three procedures: (i) Gaussian anamorphosis as an entire process; (ii) the transformation g(): "original precipitation values" -> "transformed values" (Section 2.1); (iii) its inverse: "transformed values" -> "original precipitation values" (Section 2.3). It appears that the inverse transformation described in Section 2.3 is well described since the reviewers do not raise concerns about it. The confusion arises from the description of the data transformation in Section 2.1. In particular, the procedure adopted during the data transformation for the determination of the two parameters of the Gamma distribution -shape and rate- is not clear. It is poorly described. The procedure we (try to) describe in l137-140 is the following. First of all, two different solutions have been adopted, depending on the weather conditions occurring for the hour under consideration. Two different conditions have been considered: case of a prevalence of "dry" weather; case of a prevalence of "rainy" weather. We are in the presence of "dry" weather conditions when at least one of the ensemble members reports precipitation in less than 10In case of "rainy" conditions, for each ensemble member we derive a single value of shape and a single value of rate, which is valid for the whole domain. The estimated gamma shape and rate are the maximum likelihood estimators. Then, we average all the values of shape (1 value for each ensemble member) and we get the final shape value we use for g (and its inverse). Analogously, the rate is obtained by averaging all the values of rate we found (1 value for each ensemble member). In case of "dry" weather, we fall back to the climatological values of shape and rate, which have

been estimated as the averages of shape and rate, respectively, over all the available hours/ensembles.

*L630, page 20: Rephrase the following sentence "The independent observations are used to judge if an event does in fact obtain."*
**Answer:** We will rephrase the statement.

*L649-651, page 20: We can see that performances decrease for the BSS and the ETS for precipitation strictly above 1mm/h. Are you seeing similar results when investigating the reliability for higher precipitation thresholds ? Consistency bars as suggested by Brocker and Smith (2007) in the reliability diagram could help better assess the reliability.*
**Answer:** We are considering just one summer and this limits the number of intense precipitation cases that we can use to assess the performance of EnSI-GAP, especially for higher precipitation thresholds.

*Figure 6, page 33: Please, add the unit to the figure and explain the gray color in the caption (as in the text).*
**Answer:** Will do.

==================================================================
Excerpt from reviewer2's letter (the reviewer's comments are reported in Italic)

[...] *My main weaknesses of the manuscript are the following (more details below):*

*- The paper is introducing a new method, but does not give a point of comparison to any pre-existing methods other than references to the data assimilation literature. The literature review of post-processing methods seems to be missing (and I am not an expert of it). As the paper stands, the most natural comparison should be static versus flow-dependent covariances, but that is not done in the paper. One extra experiment with alpha equals to zero would seem necessary to compare the new method to a*

*reference method, albeit very basic.*

**Answer:** The comparison of results against a spatial analysis method based on a static covariance matrix for the background error will be included in Section 3.2.

*- The benefits of flow-dependent covariances are not exhibited other than showing their existence. The ensemble does hold information about the physics of precipitation that is absent from the stationary static covariance and I would expect some meteorological interpretation of the ensemble correlation.*

**Answer:** we will include meteorological interpretation of the ensemble for the presented case study (Sec. 3.1.2)

*- The algorithm description is at times unclear, in particular the adjustment of the Gaussian anamorphosis and the need for the alpha stabilisation parameter.*

**Answer:** We will clarify those two points in the algorithm description (see answer to reviewer 1, comment on L137-140).

*- The testing focuses excessively on two values of the alpha and epsilon parameters but leaves other choices - however critical - under diffuse justifications (the D radius, the dilemma between Gaussian and exponential covariances). The authors have generally a hard time in comparing their results, possibly assuming that excessively solid evidence is needed to choose between two options. This causes an imbalance in the article between the testing of alpha and epsilon versus all of the other choices that felt more interesting to me.*

**Answer:** The objective of the paper is to present the spatial analysis method EnSI-GAP, then we propose possible solutions for the choice of parameter values. Parameter estimation is not a central topic of this manuscript, instead we just propose different configurations of EnSI-GAP and we present the methodologies used to implement them. The strategies we used in our choice of parameter values are described in the text. We will better describe the reasons that led to the choices we made. The choice of the D radius is discussed in one of the following answers to the reviewer comments (see comment referring to l370).

*- The choice of the Gamma distribution could have been supported by a histogram of observations.*
**Answer:** We will add the histogram to support our choice of a Gamma distribution.

*- The choice of error covariances could have been settled by plotting the experimental variogram of the errors against the densest observation source (the radar). The shape of the variogram at its origin is the critical choice for kriging and should be adjusted manually against whichever of the Gaussian or exponential function that fits best. Long ranges of the variogram can be ignored for the sake of this study. The adjustment of the variogram will not only reveal which of the two functions is most representative of precipitation but will also deliver the range parameter which can justify the choice of the reference length scales L and D.*
**Answer:** The reviewer is outlying here a possible procedure to estimate characteristic lengths for the functional form chosen to represent the variograms (or the spatial correlations). We will include these instructions in the manuscript as a viable alternative to our choice, which is described below (see our answer to comment referring to l370). Once again, we are extremely grateful to the reviewer for his brilliant contribution.

*- The Working Assumptions are expressed as vague principles but do not determine the algorithmic choices made thereafter. Reformulating them as statistical hypotheses (temporal independence, spatial scales, monotonicity) would make an easier read.*
**Answer:** The objective of the definitions of our working assumptions was to make explicit the motivations behind our choices, in this sense we agree that some of them are stated as general principles. We will revise the paragraph on the definition of the working assumptions, such that they will read more as statistical hypotheses.

*- The paper can be shortened in several places.*
**Answer:** We will revise the text.

*Using the Gaussian covariance function for the unpredicted scales Gamma_u is risky in cases where Pf is equal to zero (no rain the background EPS, but rain in observa-*

*tions). This is not visible in the toy experiment (Figure 2b) because the test is only considering interpolation cases but not extrapolation (observations on one side only). The Gaussian covariance has indefinitely many zero derivatives at the origin, which imposes unrealistic smoothness constraints on the analysis and, as a side effect, causes an "over-confidence" in the analysis: underestimation of the analysis uncertainty and a tendency to produce high and low values outside the range of observations. Figure 2 (b and d) shows that the uncertainties are indeed much smaller with the Gaussian than the exponential covariance (in the "missed" case between points 250 and 300) but is not decisive because both experiments seem to cover well the observations. Looking at these results, I believe that the Gaussian covariance makes less sense qualitatively because the uncertainties are lower when the model predicts no rain at all than only a little rain. As if one should be more confident when the model is completely wrong (no rain) than when it does not predict the right quantity of rain. If in addition the observation errors R had been zero, the authors would also have experienced problems with the inversion of the (S+R) matrix. The risks related to the use of a Gaussian covariance are described in Diamond and Armstrong (1984). Overall I have the impression that the method presented in the manuscript is useful and will make an interesting contribution to the field of meteorology and possibly beyond.*

**Answer:** We agree that by presenting only one toy experiment several interesting situations are not considered, such as the extrapolation. In the case of observations on one side only, by design EnSI-GAP would fall back onto the background. The warning on the use of the Gaussian covariance functions that the reviewer has shared with us will be included in the manuscript. The (S+R) matrix should be chosen such that matrix inversion does not constitute a problem. We agree that choosing a null R matrix may create a lot of problems with respect to matrix inversion. In the presented experiments/applications of EnSI-GAP (Section 3), (S+R) is an invertible matrix, because the diagonal R matrix makes the columns of (S+R) linearly independent.

*Detailed comments:*

*- The abstract does not mention that the realistic examples are using data over Norway. This could be important to know for the readers.*
**Answer:** We will modify the abstract as suggested

*- l49: The statement that the forecast is better than observations in the absence of observations seems odd to me.*
**Answer:** We will revise the statement. We meant that numerical models can provide predictions of comparable, or even better, quality then observational gridded datasets where the observational network is sparse and the terrain complex.

*- l81: I would refer to the Ensemble Kalman Filter rather than the Ensemble Optimal Interpolation because the latter uses ensemble members taken at different times, while ensemble members in your method are synchronous.*
**Answer:** Good point. We will make it clear in the manuscript.

*- Bertino et al. (2003) introduced the concept of Gaussian anamorphosis from geostatistics to data assimilation, but since you are not performing any data assimilation I would think that a general reference on geostatistics is more appropriate: see Chilès and Delfiner (2012), Chapter 6,*
**Answer:** We will modify the text as suggested.

*- l82: trans-Gaussian would deserve a definition, or to be called explicitly "transformed Gaussian".*
**Answer:** We will modify the text as suggested.

*- l85: The authors seem to refer to the "static" covariance matrix rather than "scale". I suggest you call it "static" by opposition to the flow-dependent ensemble covariance term.*
**Answer:** In fact, we were undecided on which name to use for the second term on the right-hand side of Eq. (5). As explained in l.202-203, we opted for a "scale matrix"

to emphasize the fact that we use this static matrix to scale the noise component of the model error. However, we have decided to accept the reviewer's suggestion to emphasize instead the opposition between static and time dependent (and spatially inhomogeneous) components.

*- l114: anisotropy and non-stationarity are two different topics. I agree that precipitation is non-stationary but invoking the non-stationarity of weather phenomena or simply the orography would have seemed a better explanation.*
**Answer:** We will specify it in the text.

*- l135: Specify the direction of the function: X is the Gaussian variable, not X tilde.*
**Answer:** We will specify in the text that the function g() transforms the non-Gaussian variable X tilde into the Gaussian variable X.

*- l139: The fitting of the Gaussian anamorphosis is unclear: it is adjusted to "each ensemble member" but does not state which time period nor which geographical region. This is made further puzzling in Algorithm 1, as I read that the adjustment was part of the loop "for all grid points", which implies a local adjustment at each analysis time.*
**Answer:** We will revise the description of the fitting of the Gaussian anamorphosis (see answer to reviewer 1, comment on L137-140).

*- l160: Include a reference to Table 1 somewhere in the first paragraph.*
**Answer:** Will do.

*- Section 2.2.2: The working assumptions are not mathematically well posed. They read more as general principles than working assumptions. WA3 could be rephrased as "R and Pf are proportional" for example. An assumption of locality is also missing together with WA5 on the temporal independence.*
**Answer:** We will follow the reviewer's suggestion.

*- l204: As above, "static" seems more appropriate than "scale" when you have other meanings for the word "scales" in the same paper.*

**Answer:** We will modify the text as stated above.

*- l248: The "stabilisation factor" alpha is poorly introduced (what is it supposed to stabilize?) and has no other effect than to increase the variance of the ensemble. I would therefore call it an inflation factor although it does not have the same effect as in an EnKF due to the absence of recursions. If the term "inflation" is also used in the literature related to ensemble post-processing, then I would suggest you call it "inflation Factor".*

*- l258: The reasons for introducing alpha have not been enunciated before.*
**Answer to the two comments above:** We accept the suggestion on the name and we will call alpha an "inflation factor" and we will introduce this parameter in a better way. Alpha is introduced because the procedure used to estimate sigma_2_f is not "resistant" when applied using one single timestep. On the other hand, operational applications value the fact that each timestep is independent from the others. Resistance refers to an insensitivity to misbehaviour of the data (e.g. Lanzante (1996)). Then, a few bad outliers in Eq. (9) may have a strong effect on the estimation of sigma_2_f. The introduction of alpha makes the estimation procedure more resilient in the presence of outliers and other non-standard behaviour. In this sense, it is a "stabilization" factor.

Ref:
Lanzante, J.R. (1996), Resistant, robust and non-parametric techniques for the analysis of climate data: theory and examples, including applications to historical radiosonde station data. Int. J. Climatol., 16: 1197-1226. doi:10.1002/(SICI)1097-0088(199611)16:11<1197::AID-JOC89>3.0.CO;2-L

*- Eq (13): please insist that the analysis is applied to the ensemble mean and not each member of the background ensemble. I have lost track of the definitions given a few*

*pages earlier.*
**Answer:** We will add a comment as suggested by the reviewer in Sec. 2.2.3.

*- l277: As above, refer to the EnKF rather than the EnOI here.*
**Answer:** We will follow the reviewer suggestion.

*- l284: The authors should state more explicitly that the method "does nothing" in this case.*
**Answer:** We will reformulate the statement.

*- l285: remove "direct" before "inverse" in the whole paragraph.*
**Answer:** We will do it.

*- Section 2.3 comes back to the anamorphosis and the Gamma distribution and it is not clear whether the optimization of the Gamma parameters are a repetition of what was done in l139 or if the back-transformation is using a different fit than the direct transformation in Eqs (1) and (2). Please rewrite this paragraph.*
**Answer:** We will rewrite the paragraph as suggested by both reviewers.

*- Section 3.1.1: The section is very long and it would help if simple concepts as "false positive" and "a miss" were used instead of "an alternative truth" which does not convey appropriately that the authors are dealing with ensemble bias.*
**Answer:** We will shorten the section and make use of the formulations suggested by the reviewer.

*- l347: This procedure seems to be a convolution between the signal and a smoother random signal. This does not low-pass the signal as a moving average would do. Please clarify.*
**Answer:** The reviewer is correct, the procedure is a convolution between the signal and a smoother random signal. We are implementing a convolution-based smoothing, but there is more than that and we fail in communicating properly the meaning of the implemented procedure. In fact, we are not only smoothing the signal but we are

also introducing a random noise component that inflates or deflates the signal given a prescribed spatial structure. We will revise the text accordingly.

*- l352: Between "200 u and 300 u": Figure 1b rather indicates 220 than 200.*
**Answer:** We will modify the text as suggested.

*- l360: Is "the multiplicative model" compatible with the assumption of a Gamma distribution?*
**Answer:** In line 360, the reference to the multiplicative model is connected to the procedure of the creation of synthetic precipitation data. This might not be clear enough in the text and we will revise it to make it more explicit. However, the reviewer's question is a different one and we think that the assumption of a Gamma distribution is compatible with the multiplicative error model for hourly precipitation. According to Wilks (2019), the Gamma distribution is a common choice for representing variables that are distinctly asymmetric and skewed to the right, such as precipitation amounts or wind speed. In the manuscript, we refer to the paper by Tian et al. (2013) where they discuss the model error suitable for daily precipitation. They make a distinction between the choice of an optimal error model that involves the aggregation time chosen. In fact, they state that (paragraph 25) "However, the selection of an error model is dictated by the data. We speculate that at coarser spatial and temporal resolutions (seasonal or longer), the magnitude of precipitation variability is much suppressed and both precipitation and the measurement errors are closer to the normal distribution [e.g., Sardeshmukh et al., 2000], and the additive model may become more viable. On the other hand, at finer spatial and temporal resolutions, the probability distributions of precipitation and the errors are highly skewed and closer to the Gamma or lognormal distribution, and the multiplicative model may prevail. How the error modeling transitions with the spatial and temporal scales requires further study. Nevertheless, the three criteria proposed in this paper are general and rational enough to be applicable to other data sets and models as well."

Ref:

Tian, Y., Huffman, G. J., Adler, R. F., Tang, L., Sapiano, M., Maggioni, V., and Wu, H.: Modeling errors in daily precipitation measurements: Additive or multiplicative?, Geophysical Research Letters, 40, 2060–2065, 2013. Wilks, D. S. (2019): Statistical methods in the atmospheric sciences. (Fourth Edition)

*- l370: It seems that the length scale of the unresolved covariance is dependent on the measurement network but not the physics of precipitation. This comes as a surprise and should have been clarified upfront as a working assumption in Section 2.2.*
**Answer:** The solution we propose at line 370 for the determination of the length scale D is "D is estimated as the distance between the i-th grid point and its closest 3rd observation location". The more general statement we use for EnSI-GAP is that there is a part of the covariance matrix Pb (Eq.(5)) that is represented by the ensemble, then there is a part of Pb that is not explained by the ensemble and here we introduce Gamma_u. In EnSI-GAP, we do not postulate any shape of Gamma_u as being preferable to another, this depends on the application. The reference reported at line 219 (Gaspari and Cohn, (2019)) should be useful for the readers in helping them choose the best option among several correlation functions. For this reason, the working assumptions in Section 2.2 do not deal with the settings of Gamma_u. We should better specify in the text this distinction between the general method we propose and the "customization" of the method we adopt in the result and evaluation sections. As remarked by the reviewer, we decide that the length scale D of that part of the covariance matrix which is not possible to explain by means of the ensemble is determined by the characteristics of the observational network. The topic raised by the reviewer is discussed in the text between lines 677-681 (Section 3.3, Discussion). We will revise the text to ensure that different parts of the manuscript discussing the same topics are linked, such that the readers can be better oriented in the text. Note that our choice of D makes the length scale variable across the domain, while to our knowledge most of the spatial analysis schemes assume fixed D over the whole domain. As stated in the text (l667-670), the settings of location-dependent parameters have to avoid sharp

variations in space.

*- l387: These three references seem to imply that the exponential covariance is more common than the Gaussian for precipitation.*
**Answer:** We will add other references where the Gaussian covariance is used.

*- l394: Variations in the scale matrix -> the choice of the static covariance matrix.*
**Answer:** Will do.

*- l401: remove "almost" since Pf is equal to zero.*
**Answer:** We will modify the text as suggested.

*- l402-404: Keep this remark until after Figure 4 has been introduced.*
**Answer:** We will move the remark at a different point in the text.

*- l428: Why restraining yourselves with a qualitative judgement when you could have computed a CPRS or a simple ratio of errors versus estimator uncertainty?*
**Answer:** As suggested, we will quantify the agreement to make our statement objective.

*- l436-442: I believe this part can be shortened.*
**Answer:** We will revise the text.

*- Figures 6 and 7 should include the Sogn og Fjordane box.*
**Answer:** The Sogn og Fjordane box will be included in the figures.

*- l.495 Where observations are absent (gray shaded areas), the analysis differs from the background. Please comment on whether this change is realistic or not.*
**Answer:** We consider the analysis to be realistic where observations are absent, since they are in line with the type of event under examination. We will comment on this issue in the text.

*- l508: Kriging does induce some systematic smoothing, the authors could have made a note of that when discussing Eq (13). Smoothing can be avoided by other methods*

*in the family of conditional simulations.*
**Answer:** We will make a note, as suggested.

*- l 522: The extrapolated values are not commented, are these expected or not?*
**Answer:** The values are realistic, considering the characteristics of the event and the climatology of the region.

*- l531: This is a very long sentence to say "certainly no rain".*
**Answer:** OK. We will modify it.

*- l532-534: This passage does not seem to be connected with the rest of the text, please make a point or remove it.*
**Answer:** We will rephrase it.

*- l543: Which "EnSI-GAP settings"? There are several of them.*
**Answer:** We will reformulate the statement to avoid confusion.

*- l. 549: Is it ok that the analysis rules out the precipitation at point B where no observations are present?*
**Answer:** it seems that the analysis is making the correct choice for those hours, because the surrounding observations do not report precipitation. We are inclined to believe that the onset of precipitation is well represented at point B in this case. However, since we do not have a measurement station located exactly at point B we have no way to know this for sure. We have included section 3.2 with the validation over a longer time period to integrate the case-specific results of section 3.1.

*- l551-555: As above, the take-home message of these numbers is not obvious, please shorten the passage or remove.*
**Answer:** We will rephrase it.

*- l560: I have been confused by the contradiction between (i) and (ii). Please rephrase.*
**Answer:** We will rephrase it.

*- Section 3.2: the case "alpha=0.1" can be called a case of "deflation" by opposition to*

*"inflation".*
**Answer:** yes, correct. We will include this remark in the text.

*- Figures 12 to 14 seem to give the same message as Figure 11. I believe that they can be removed together with the related text without loss to the argument.*
**Answer:** We believe that Fig. 12 on the ETS is adding information with respect to the performances of the method for different precipitation thresholds. Figs. 13-14 can be regarded as "supplementary material" and added to that specific section that we will add to the manuscript.

*- l679: The justification of the choice of D value should come upfront in Section 2 rather than in the discussion.*
**Answer:** At this point of the manuscript we are describing our motivations in the choice of D. Other choices are equally valid. The method does not constrain people to use the same choices we have made. However, we will specify our choice of D in Sec. 2.

*- l685: Why "assuming" that the background ensemble is more likely to overestimate the spread while you could have verified it with the data at hand?*
**Answer:** We will reformulate the statement, since we have verified it.

*- Algorithm 1, under "Require": include the calculation of the ensemble mean and the predefined anamorphosis function.*
**Answer:** Will modify as suggested.

*- Figure 1: vertically shade areas of "false alarms" and "miss".*
**Answer:** Will modify as suggested.

*- Figure 6: delineate the boundary of the grey area in panel (c) to highlight what the analysis does in the absence of observations. Do that as well in Figure 8.*
**Answer:** Will modify as suggested.

*- Figure 8: annotate the points "A" and "B" on the figure.*
**Answer:** Will modify as suggested.

*- Figure 10: The flow-dependence of the correlations is visible, but does not speak much for itself. The ensemble may contain some information about the orography for example and it could be interesting to include the elevations with a few isolines.*
**Answer:** Will modify as suggested.

*- Figure 11: An integrated measure of the goodness-of-fit can be included in the legend of each panel.*
**Answer:** Will modify as suggested.

*Typos and minor issues*
**Answer:** Thanks a lot for this. We will correct the typos.

*- l197: tenths -> tens*

*- l395: remove "the" before "higher uncertainties".*

*- Use "anamorphosis" instead of "anamorphism". I don't think "anamorphism" is grammatically wrong but it is less usual in the literature.*

*- l499: "clearly" can be removed. Let the reader judge the quality of the figure.*

*- l505: "it is evident a sharp gradient..." -> "a sharp gradient [...] is evident".*

*- l539: "sort of outliers" -> "outliers".*

*- l617: "much accurate" -> "much more accurate"*

*- l618: "those" -> "that"*

*References:*

*Chilès, J.P. and P. Delfiner (2012) Geostatistics: Modeling Spatial Uncertainty. Wiley and sons. https://onlinelibrary.wiley.com/doi/book/10.1002/9781118136188*

*Diamond, P., and Armstrong, M. (1984), "Robustness of Variograms and Conditioning of Kriging Matrices," Mathematical Geology, 16, 809–822.*
* * *

---

## Editor Comment (EC2) · Alberto Carrassi (Editor) · 25 Sep 2020

Dear Authors,

I found your answers to the Reviewers exhaustive and I invite you to submit a revised version of the manuscript with a "diff-version" included.

Best Regards

Alberto Carrassi
* * *
2020-20, 2020.

---

## Author Response (AR1)

Dear Editor, Journal of Nonlinear Processes in Geophysics

We would like to thank you for the letter dated 25/09/2020, and the opportunity to resubmit a revised copy of this manuscript. We would also like to take this opportunity to express our thanks to the reviewers for the positive feedback and helpful comments for correction or modification.

We believe it has resulted in an improved revised manuscript. The manuscript has been revised to address the reviewer comments, which are appended alongside our responses to this letter.

We very much hope the revised manuscript is accepted for publication in the Journal of Nonlinear Processes in Geophysics.

Sincerely yours,
Cristian Lussana on behalf of the authors

In the interactive discussion, our answers to the reviewers comments include point-by-point responses to the reviews. A list of the relevant changes follows:

- Better description of the data transformation. We have revised Secs. 2.1 - 2.2 and 3.3. Adjusted the algorithm and the tables with the mathematical notations. We have Introduced the scalar variables $\alpha_D$ and $\beta_D$ defining the gamma distribution used in the data transformation. We have Introduced the vectors $\alpha^A$ and $\beta^A$ defining the gamma distributions of hourly precipitation at grid points. The coefficient previously denoted with $\alpha$ is now $\nu$. Fig 8 is new and it serves two purposes: It shows an example of data transformation and it supports our choice of using a Gamma cumulative distribution function in the transformation.

- The "stabilization factor" has been renamed as "inflation factor" and it has been introduced and described in a better way than before.

- Reviewer 2 suggested to refer to EnKF instead of EnOI, because EnOI makes use of a time-lagged ensemble. We have modified the statement when we are referring to EnOI to point out the differences. However, we still make a connection between EnSI-GAP and EnOI, since the EnSI-GAP equations are more similar to EnOI than EnKF.

- We have tried to implement the suggestion of Reviewer2 on renaming the "scale matrix" as the "static covariance matrix". However, the term "static covariance matrix" is rather general and it may generate confusion. For instance, the observation error covariance matrix is also a static covariance matrix. In addition, the scale matrix may change every hour and in this sense it is not static. To avoid confusion, we should have replaced the "scale matrix" with "static background error covariance matrix", which is too long. We keep the term "scale matrix" and we have added a disambiguation on the fact that we are not referring to "spatial scales".

- Working assumptions introduced in Sec. 2.2.2. We have almost entirely rewritten

Sec. 2.2.2 and adapted Sec. 2.2.3. The working assumptions are first formulated as general principles, then at the end of the section they are rewritten in more precise mathematical terms with links to the corresponding equations. We have also linked the different parts in the text where they are used.

- Section 3. Results. This section has been almost completely rewritten. It has been also better organized in subsections. The figures referring to this section have been modified and new figures have been added. Two figures have been removed (Figs. 13 and 14).

- We discuss other ways to determine the background error covariance matrices, in addition to those used in the manuscript, in Sec 3.1.6. The reader could also find the other methods in the literature we cite.

- Section 3.1 has been completely rewritten, new results have been added, Figures have been re-plotted. The same simulation as before has been considered, moreover we have extended the considerations to 100 simulations similar to the one presented. This way, our conclusions are more general, since they are not related to a single case. The interpretation of results is more quantitative and less qualitative, thanks to the introduction of two scores MSESS and CRPS. The session is now better organized, because of the subdivision into subsections. Figs. 2-6 are either new or re-done. Table 3 summarizes the statistics of results over 100 simulations.

- Figures 8 and 10 (ex-figures 6 and 8) have been modified to emphasize the region where the observations are not available. In Fig. 8 we have also added the Sogn og Fjordane box. Fig 10 has isolines.

**Ensemble-based statistical interpolation with Gaussian anamorphosis for the spatial analysis of precipitation**

Cristian Lussana[1], Thomas N. Nipen[1], Ivar A. Seierstad[1], and Christoffer A. Elo[1]

[1]Norwegian Meteorological Institute, Oslo, Norway

**Correspondence:** Cristian Lussana (cristianl@met.no)

**Abstract.** Hourly precipitation over a region is often simultaneously simulated by numerical models and observed by multiple data sources. An accurate precipitation representation based on all available information is a valuable result for numerous applications and a critical aspect of climate monitoring. Inverse problem theory offers an ideal framework for the combination of observations with a numerical model background. In particular, we have considered a modified ensemble optimal interpolation scheme, that takes into account deficiencies of the background. An additional source of uncertainty for the ensemble background has been included. The deviations between background and observations are used to adjust for deficiencies of the ensemble. A data transformation based on Gaussian anamorphosis has been used to optimally exploit the potential of the spatial analysis, given that precipitation is approximated with a gamma distribution and the spatial analysis requires normally distributed variables. For each point, the spatial analysis returns the shape and rate parameters of its gamma distribution. The Ensemble-based Statistical Interpolation scheme with Gaussian AnamorPhosis (EnSI-GAP) is implemented in a way that the covariance matrices are locally stationary and the background error covariance matrix undergoes a localization process. Concepts and methods that are usually found in data assimilation are here applied to spatial analysis, where they have been adapted in an original way to represent precipitation at finer spatial scales than those resolved by the background, at least where the observational network is dense enough. The EnSI-GAP setup requires the specification of a restricted number of parameters and specifically the explicit values of the error variances are not needed, since they are inferred from the available data. The examples of applications presented over Norway provide a better understanding of the characteristics of EnSI-GAP. The data sources considered are those typically used at national meteorological services, such as local area models, weather radars and in-situ observations. For this last data source, measurements from both traditional and opportunistic sensors have been considered.

*Copyright statement.* Usage rights are regulated through the Creative Commons Attribution 3.0 License (https://creativecommons.org/licenses/by/3.0).

[revised manuscript text omitted]

$$ {}^i\hat{\sigma}_b^2 \;\widetilde{\equiv}\; {}^i\hat{\sigma}_f^2 + {}^i\hat{\sigma}_u^2 \tag{9}$$

$\varepsilon^2$ is  a global variable and it is the relative precision of the observations with respect to the background. Eq. (7) implements P3 and $\varepsilon^2$ should be set to a value smaller than  1. For example, $\varepsilon^2 = 0.1$ means that we believe the observations to be ten times more precise an estimate of the true value than the background.

 Eq. (8) is an adaptation from Eq. (5).

$$ {}^i\hat{\sigma}_b^2 = {}^i\hat{\sigma}_f^2 + {}^i\hat{\sigma}_u^2 $$

$$ {}^i\hat{\sigma}_f^2 = \alpha \; \langle \mathrm{diag}\left( {}^i\mathbf{S}^{\mathrm{f}} \right) \rangle = \alpha \; \frac{1}{p_i} \sum_{j=1}^{p_i} {}^i\mathbf{S}^{\mathrm{f}}{}_{jj} $$

 . The next two relationships we introduce have the objective to estimate ${}^i\hat{\sigma}_f^2$  and the empirical (i.e. based on data, not on theories) estimate of ${}^i\hat{\sigma}_{ob}^2$, which is the sum of ${}^i\hat{\sigma}_o^2$ plus ${}^i\hat{\sigma}_b^2$, directly from the forecasts and the observed values. ${}^i\hat{\sigma}_{ob}^2$ is used to get a reference value to judge if the ensemble spread is adequate. The equations are (the averaging operator $\langle \dots \rangle$ is defined as in Algorithm 4):

$$ {}^i\hat{\sigma}_f^2 \;\widetilde{\equiv}\; \nu \; \langle \mathrm{diag}\left( {}^i\mathbf{S}^{\mathrm{f}} \right) \rangle \tag{10}$$

$$ {}^i\hat{\sigma}_{ob}^2 \;\widetilde{\equiv}\; \nu \; \langle \left( {}^i\hat{\mathbf{y}}^{\mathrm{o}} - {}^i\hat{\
[revised manuscript text omitted]

Data back transformation

505  inverse transformation $g^{-1}$ of 400 quantiles of the distribution $N\left(\mathbf{x}^{\mathrm{a}}_i, (\boldsymbol{\sigma}^2)^{\mathrm{a}}_i\right)$ $\boldsymbol{\alpha}^{\mathrm{a}}_i$ and $\boldsymbol{\beta}^{\mathrm{a}}_i$ are obtained by optimizing the

fitting of a gamma distribution to the 400 quantiles through a least squares fitting method

---

## Author Response (AR2)

01-12-2020

Dear Editor Alberto Carrassi, Journal of Nonlinear Processes in Geophysics

Further to our correspondence a couple of days ago, I'm attaching the revised version of the article entitled "Ensemble-based statistical interpolation with Gaussian anamorphosis for the spatial analysis of precipitation" I have now completed all of the changes you requested.

I hope that the changes I've made resolve all your concerns about the article. I'm more than happy to make any further changes that will improve the paper and/or facilitate successful publication.

Thank you once again for your time and interest. I look forward to hearing from you.

Sincerely,

Cristian Lussana on behalf of the authors

**Ensemble-based statistical interpolation with Gaussian anamorphosis for the spatial analysis of precipitation**

Cristian Lussana[1], Thomas N. Nipen[1], Ivar A. Seierstad[1], and Christoffer A. Elo[1]

[1]Norwegian Meteorological Institute, Oslo, Norway

**Correspondence:** Cristian Lussana (cristianl@met.no)

**Abstract.** Hourly precipitation over a region is often simultaneously simulated by numerical models and observed by multiple data sources. An accurate precipitation representation based on all available information is a valuable result for numerous applications and a critical aspect of climate monitoring. Inverse problem theory offers an ideal framework for the combination of observations with a numerical model background. In particular, we have considered a modified ensemble optimal interpolation scheme. The deviations between background and observations are used to adjust for deficiencies of the ensemble. A data transformation based on Gaussian anamorphosis has been used to optimally exploit the potential of the spatial analysis, given that precipitation is approximated with a gamma distribution and the spatial analysis requires normally distributed variables. For each point, the spatial analysis returns the shape and rate parameters of its gamma distribution. The Ensemble-based Statistical Interpolation scheme with Gaussian AnamorPhosis (EnSI-GAP) is implemented in a way that the covariance matrices are locally stationary and the background error covariance matrix undergoes a localization process. Concepts and methods that are usually found in data assimilation are here applied to spatial analysis, where they have been adapted in an original way to represent precipitation at finer spatial scales than those resolved by the background, at least where the observational network is dense enough. The EnSI-GAP setup requires the specification of a restricted number of parameters and specifically the explicit values of the error variances are not needed, since they are inferred from the available data. The examples of applications presented over Norway provide a better understanding of EnSI-GAP. The data sources considered are those typically used at national meteorological services, such as local area models, weather radars and in-situ observations. For this last data source, measurements from both traditional and opportunistic sensors have been considered.

*Copyright statement.* Usage rights are regulated through the Creative Commons Attribution 3.0 License (https://creativecommons.org/licenses/by/3.0).

[revised manuscript text omitted]